# ECLipsE-Gen-Local: Efficient Compositional Local Lipschitz Estimates for Deep Neural Networks

**Yuezhu Xu** *xu1732@purdue.edu*
*Edwardson School of Industrial Engineering*
*Purdue University*

**S. Sivaranjani** *sseetha@purdue.edu*
*Edwardson School of Industrial Engineering*
*Purdue University*

**Reviewed on OpenReview:** *https://openreview.net/forum?id=CuqnFjeu5a*

## Abstract

The Lipschitz constant is a key measure for certifying the robustness of neural networks to input perturbations. However, computing the exact constant is NP-hard, and standard approaches to estimate the Lipschitz constant involve solving a large matrix semidefinite program (SDP) that scales poorly with network size. Further, there is a potential to efficiently leverage local information on the input region to provide tighter Lipschitz estimates. We address this problem here by proposing a compositional framework that yields tight yet scalable Lipschitz estimates for deep feedforward neural networks. Specifically, we begin by developing a generalized SDP framework for Lipschitz estimation that is highly flexible, accommodating heterogeneous activation function slope bounds for each neuron on each layer, and allowing Lipschitz estimates with respect to arbitrary input-output pairs in the neural network and arbitrary choices of sub-networks of consecutive layers. We then decompose this generalized SDP into a equivalent small sub-problems that can be solved sequentially, yielding the ECLipsE-Gen series of algorithms, with computational complexity that scales linearly with respect to the network depth. We also develop a variant that achieves near-instantaneous computation through closed-form solutions to each sub-problem. All our algorithms are accompanied by theoretical guarantees on feasibility and validity, serving as strict upper bounds on the true Lipschitz constant. Next, we develop a series of algorithms, termed as ECLipsE-Gen-Local *, that explicitly incorporate local information on the input region to provide tighter Lipschitz constant estimates. Our experiments demonstrate that our algorithms achieve substantial speedups over a multitude of benchmarks while producing significantly tighter Lipschitz bounds than global approaches. Moreover, we demonstrate that our algorithms provide strict upper bounds for the Lipschitz constant with values approaching the exact Jacobian from autodiff when the input region is small enough. Finally, we demonstrate the practical utility of our approach by showing that our Lipschitz estimates closely align with network robustness. In summary, our approach considerably advances the scalability and efficiency of certifying neural network robustness, while capturing local input–output behavior to deliver provably tighter bounds, making it particularly suitable for safety-critical and adaptive learning tasks.

---

This work was partially supported by the Air Force Office of Scientific Research grant, FA9550-23-1-0492.

*Our code is available at `https://github.com/YuezhuXu/ECLipsE`, and the associated Python package can be installed via PyPI at `https://pypi.org/project/eclipse-nn/`.

# 1 Introduction

Neural networks (NNs) are extensively deployed in a wide range of domains (LeCun et al. (2015)), from autonomous systems (Antsaklis et al. (1990); Tang et al. (2022)), power system (Haque & Kashtiban (2000)) to medical diagnostics (Amato et al. (2013)). While NN-based models have achieved remarkable performance, it remains a major challenge to provide rigorous guarantees on the behavior of NNs, especially in safety-critical applications. Specifically, it is desirable to provide robustness certificates (Carlini & Wagner (2017); Zhang et al. (2018); Fazlyab et al. (2020); Tan & Wu (2024); Fazlyab et al. (2023)), achieve resilience against adversarial attacks (Tsuzuku et al. (2018); Amini & Ghaemmaghami (2020); Finlay et al. (2018); Zühlke & Kudenko (2025)), and ensure stability in NN-based control, (Aswani et al. (2013); Brunke et al. (2022); Yin et al. (2021); Xu & Sivaranjani (2023); Sun et al. (2019). In these applications, it is essential to characterize the behavior of model outputs under input perturbations to ensure safety and robustness.

One widely adopted metric is the Lipschitz constant, which quantifies the worst-case output deviation per unit input change. Despite its fundamental role in certifying robustness, computing the exact Lipschitz constant of a neural network is NP-hard (Virmaux & Scaman (2018)). Consequently, significant efforts have been made to obtain tight and provable upper bounds for feedforward networks (FNNs) and a variety of network architectures beyond, such as convolutional neural networks (CNNs), and residual networks (Pauli et al. (2023); Wang et al. (2024); Pauli et al. (2024); Fazlyab et al. (2023)). For FNNs, both global and local Lipschitz bounds are addressed in these studies, considering networks with different type of activation functions, such as piecewise linear (most commonly ReLU) (Virmaux & Scaman (2018)), differentiable (Latorre et al. (2020)), or general ones (Xu & Sivaranjani (2024)). Also, Lipschitz constants defined for various norm choices, such as $\ell_1$- (Jordan & Dimakis (2020)), $\ell_p$- (Virmaux & Scaman (2018)), $\ell_\infty$- (Shi et al. (2022)), and cross-norms (Wang et al. (2022)), are investigated. A detailed compilation of these works is presented later in this section.

In this paper, we focus on the problem of estimating both the global and local $\ell_2$-norm Lipschitz constants for FNNs. The $\ell_2$-norm is a standard robustness metric in signal processing, control theory, and scientific modeling domains (Fazlyab et al. (2019); Tsuzuku et al. (2018)). It also plays a central role in reachability analysis for NN-based models, which is crucial in safety-critical control applications like autonomous driving, robotics, and power systems (Ruan et al. (2018); Everett et al. (2021); Xiang et al. (2020); Huang et al. (2019)). In machine learning, many theoretical generalization bounds for NNs are directly linked to their $\ell_2$-Lipschitz constant (Bartlett et al. (2017); Neyshabur et al. (2017)). These applications have also motivated the development of methods to design NNs with certifiable robustness guarantees (Huang et al. (2021); Fazlyab et al. (2023); Wang & Manchester (2023); Araujo et al. (2023); Havens et al. (2024)).

Typical approaches for $\ell_2$-norm Lipschitz estimation involve semidefinite program (SDP), as in the LipSDP framework (Fazlyab et al. (2019)), where the slope-restrictedness of the NN activation functions is leveraged to formulate the problem of Lipschitz estimation as a large linear matrix inequality (LMI). Despite their accuracy, the computational complexity of SDP-based methods grows exponentially with network depth. Approaches to enhance scalability of SDP-based Lipschitz estimation methods include neglecting specific neuron coupling constraints at the expense of bound tightness (Fazlyab et al. (2019)), exploiting matrix sparsity through chordal decomposition to generate smaller, more tractable LMIs (Newton & Papachristodoulou (2021)), dissipativity-based approaches (Pauli et al. (2023; 2024)), eigenvalue optimization and memory-efficient computations through `autodiff` (Wang et al. (2024)), and compositional methods that leverage the geometric properties of the underlying SDP to decompose it into a series of sequential sub-problems (Xu & Sivaranjani (2024)), significantly advancing the practical utility of SDP-based Lipschitz estimation methods for deep neural networks. However, all these works remain limited to estimating the global Lipschitz constant over the entire Euclidean space. In contrast, exploiting local information about the input domain can yield more precise Lipschitz bounds, which is a key contribution of this paper.

**Theoretical Approach.** We start with generalizing the certificate of Fazlyab et al. (2019) to allow heterogeneous, nontrivial slope-restrictedness bounds (briefly, slope bounds) for the activation functions at each neuron, as well as Lipschitz constant estimates for subsets of NN layers and arbitrary selections of input-output indices. We then build on the compositional decomposition framework in the ECLipsE series of algorithms proposed in Xu & Sivaranjani (2024) to decompose the resulting large LMI into a series

of small-subproblems that are solved sequentially. It is important to note that the algorithms in Xu & Sivaranjani (2024) assume that the lower slope bound of each activation function is zero and are no longer directly applicable when we generalize to heterogeneous and general slope bounds. The result is a series of algorithms, termed ECLipsE-Gen, to determine the decision variables at each stage. Further, in contrast to ECLipsE Xu & Sivaranjani (2024), we incorporate local information on the input region to derive tighter slope bounds for the activation functions of each neuron, yielding more accurate local Lipschitz estimates. In our algorithms, we iteratively refine the slope bounds for each neuron at each stage, and compute a messenger matrix that passes local information from one stage to the next. This sequential algorithm achieves computational complexity that scales linearly with the network depth, while yielding tight Lipschitz bounds. We further relax the sub-problems to derive a variant, ECLipsE-Gen-Local-CF, that provides closed-form solutions at each stage, completely eliminating the need to solve any SDP, while achieving tighter Lipschitz bounds compared to ECLipsE-Fast Xu & Sivaranjani (2024).

**Contribution.** In this work, we propose a scalable compositional framework that leverages *local information* on the input region to yield *tighter, certified* Lipschitz estimates for deep FNNs. Our main contributions are as follows:

1. We generalize the Lipschitz constant certificates in Fazlyab et al. (2019) by allowing ***heterogeneous, nontrivial slope bounds for each neuron***, and subsequently decompose the resulting large SDP into a series of small, computationally tractable sub-problems inspired by Xu & Sivaranjani (2024). The resulting algorithms are termed the ECLipsE-Gen series.

2. We develop ECLipsE-Gen-Local, a series of algorithms that incorporate local information of the input region to iteratively refine slope bounds while ***propagating information layer by layer, enabling tighter local Lipschitz upper bounds*** compared to the global bounds, while providing strict ***theoretical guarantees*** on feasibility, validity, and tightness.

3. We provide extensive experiments demonstrating that our algorithms consistently produce ***more accurate Lipschitz estimates*** compared to global methods, while achieving ***computational speedups of several orders of magnitude*** over traditional SDP-based approaches. Furthermore, we empirically show that when the input region is considerably small (the neighborhood of a specific point), our ***local estimates approach the exact Lipschitz constant*** represented by the norm of the Jacobian at the region's center, ***reaching the tightness level of the values obtained by `autodiff`*** and thus validating the exceptional tightness of our proposed method, ECLipsE-Gen-Local.

4. We establish a generalized certificate that supports ***arbitrary input-output index selections and consecutive-layer subsets*** (Theorem 3). This flexibility is not only used internally for our layer-wise slope-bound refinement, but also serves as a versatile foundation for targeted downstream certification tasks. As one representative example, we leverage the certificate to perform reachability analysis by computing a certified over-approximation of the reachable output region (Theorem 6).

Beyond standard robustness certification, the flexibility offered by our framework to select arbitrary indices and layer subsets enables specific certified analyses in modular pipelines. For instance, arbitrary index selection allows for coordinate-level sensitivity bounds, which are crucial when outputs represent distinct physical quantities in control systems (e.g., Wei & Liu (2022)) or physical surrogates (e.g., NN surrogates for power flow physics Zhao et al. (2020); Pan (2021)). Similarly, consecutive-layer selection supports the verification of intermediate representations in architectures like early-exit networks (Teerapittayanon et al. (2016)), encoder-decoder structure (Hinton & Salakhutdinov (2006)), or multi-task models (Misra et al. (2016)). Additionally, our reachability analysis (Theorem 6) provides certified output set over-approximations, explicitly illustrating the practical benefit of reduced conservatism for safety verification.

**Related Work.** Estimating the Lipschitz constant of neural networks is NP-hard (Virmaux & Scaman (2018)). The most basic method is the naive upper bound based on the product of induced weight norms (Szegedy et al. (2013)), which is highly conservative. Other practical methods include automatic differentiation-based approximations, which have practical utility but do not provide strict upper bounds on the true Lipschitz constant (Virmaux & Scaman (2018)). More advanced analyses leverage the composition of non-expansive

and affine operators (Chen et al. (2020)), and scalable alternatives using bound propagation to derive local Lipschitz estimates (Zhang et al. (2019); Shi et al. (2022)). Exact layer-wise analytic estimates have been developed for specific architectures Avant & Morgansen (2023) on each layer. Jacobian composition analyses (Zhang et al. (2019)) provide tighter bounds by analyzing compositions of activation functions directly, yielding both upper and lower bounds for the Jacobian. Besides, optimization-based methods have made substantial progress in tightening Lipschitz bounds. For example, taking advantage of the piecewise linear nature of the ReLU activation function, Weng et al. (2018) and Jordan & Dimakis (2020) formulate the Lipschitz constant estimation problem into a linear program (LP) or mixed-integer program (MIP) respectively. Another approach is to encode Lipschitz estimation as a sparse polynomial optimization and further relax the problems into more tractable forms such as quadratically constrained quadratic program (QCQP), second-order cone program (SOCP), and SDP Latorre et al. (2020). Tight estimates can also be obtained using branch-and-bound methods via partitioning Bhowmick et al. (2021), which can be further integrated with other approaches (Shi et al. (2022)). Various methods also differ in their norm specificity, including the general $\ell_p$-norm (Virmaux & Scaman (2018); Bhowmick et al. (2021); Weng et al. (2018)), the $\ell_2$-norm (Fazlyab et al. (2019); Xue et al. (2022); Avant & Morgansen (2023); Wang et al. (2022); Pauli et al. (2023; 2024); Wang et al. (2024); Xu & Sivaranjani (2024)), $\ell_1$-norm (Jordan & Dimakis (2020)), $\ell_\infty$-norm (Latorre et al. (2020); Jordan & Dimakis (2020); Shi et al. (2022)), and methods accommodating arbitrary norms (Combettes & Pesquet (2020); Chen et al. (2020); Zhang et al. (2019)).

Typical $\ell_2$-norm Lipschitz estimation methods such as LipSDP (Fazlyab et al., 2019) rely on SDPs, which while offering accurate Lipschitz estimates often suffer from poor scalability with increasing neural network depth. Strategies to enhance scalability of SDP-based Lipschitz estimation methods include relaxing neuron coupling constraints (Fazlyab et al. (2019)), dissipativity-based formulations (Pauli et al. (2023; 2024)), sparsity exploitation via chordal decomposition (Newton & Papachristodoulou (2021)), eigenvalue optimization and memory-efficient implementations (Wang et al. (2024)) via `autodiff` (Rumelhart et al. (1986)), and compositional decompositions of the SDP into smaller sub-problems (Xu & Sivaranjani (2024)). While these advances enhance scalability, they are still confined to global Lipschitz estimation, whereas this work leverages local input information to obtain sharper bounds on the $\ell_2$ norm Lipschitz constant of deep NNs.

## 2 Problem Formulation and Background

**Notation.** We define $\mathbb{Z}_N = \{1, \ldots, N\}$, where $N$ is a natural number excluding zero. For set $X$ and its subset $Y$, $X \backslash Y$ is the complement of set $Y$ in $X$. $|X|$ represents the number of elements in set $X$. Identity matrix of dimension $n$ is denoted as $I_n$ or briefly $I$ with dimension clear from the context. A symmetric positive-definite matrix $P \in \mathbb{R}^{n \times n}$ is represented as $P > 0$ (and as $P \geq 0$, if it is positive semi-definite). For symmetric matrix $A$ and $B$, $A \geq (>)B$ means $A - B \geq (>)0$. For two vectors $x$ and $y$, $x \leq (\geq)y$ means $x$ is no smaller (larger) than $y$ elementwise. We denote the largest singular value or the spectral norm of matrix $A \geq 0$ by $\sigma_{max}(A)$. The set of positive semi-definite diagonal matrices is written as $\mathbb{D}_+$. For any vector $x$, $D_x$ represents the matrix with the entries of $x$ along its diagonal. We use calligraphic style such as $\mathcal{K}$ for a set of indices. $(x)_{\mathcal{K}}$ denotes the subvector of $x$ indexed by the set $\mathcal{K}$. Briefly, We denote $(x)_k$ if $\mathcal{K} = \{k\}$. $(A)_{(\mathcal{K},\mathcal{L})}$ selects rows $\mathcal{K}$ and columns $\mathcal{L}$; $(A)_{(\bullet,\mathcal{L})}$ and $(A)_{(\mathcal{K},\bullet)}$ select all rows or columns with columns or rows indexed by $\mathcal{L}$ or $\mathcal{K}$, respectively. $\odot$ is the point-wise multiplication. We represent the ball with center $c$ and radius $r$ by $\mathcal{B}(c, r)$.

### 2.1 Problem Formulation

Consider a standard feedforward neural network (FNN) with $N$ layers, where the input is $z \in \mathcal{Z} \subseteq \mathbb{R}^{d_0}$ and the output is $y \in \mathcal{Y} \subseteq \mathbb{R}^{d_N}$, with the input output mapping of the FNN given by $y = f(z)$. Here, $\mathcal{Z}$ denotes the local domain of interest, and $\mathcal{Y}$ is the corresponding codomain. The function $f$ is defined recursively through layers $L_i$, for $i \in \mathbb{Z}_N$, as follows:

$$L_i : z^{(i)} = \phi(v^{(i)}) \quad \forall i \in \mathbb{Z}_{N-1}, \qquad L_N : y = f(z) = z^{(N)} = v^{(N)}, \qquad z^{(0)} = v^{(0)} = z, \qquad (1)$$

where $v^{(i)} = W_i z^{(i-1)} + b_i$, with $W_i$ and $b_i$ denoting the weight matrix and bias vector for layer $L_i$, respectively. The activation function $\phi : \mathbb{R}^{d_i} \to \mathbb{R}^{d_i}$ is applied element-wise to its input. The final layer, $L_N$, is referred to

as the *output layer*. We use $d_i$ to denote the number of neurons in layer $L_i$ for each $i \in \mathbb{Z}_N$. We also define function $f^{(i)} : \mathbb{R}^{d_0} \to \mathbb{R}^{d_i}$ to be the mapping from $z^{(0)}$ to $v^{(i)}$, so that $f = f^{(N)}$. For notational consistency, we let $f^{(0)} : \mathbb{R}^{d_0} \to \mathbb{R}^{d_0}$ to be the identity mapping, i.e., $f^{(0)}(z) = z$ for any $z \in \mathbb{R}^{d_0}$.

**Definition 1.** *For any given region $\mathcal{Z} \subseteq \mathbb{R}^{d_0}$, the function $f : \mathbb{R}^{d_0} \to \mathbb{R}^{d_N}$ is locally Lipschitz continuous on $\mathcal{Z}$ if there exists a constant $\boldsymbol{L} > 0$ such that $\|f(z_1) - f(z_2)\|_2 \leq \boldsymbol{L}\|z_1 - z_2\|_2, \forall z_1, z_2 \in \mathcal{Z}$. The smallest positive $\boldsymbol{L}$ satisfying this inequality is termed the local Lipschitz constant of the function $f$ on domain $\mathcal{Z}$.*

**Problem.** We aim to estimate a tight and strict upper bound for the local Lipschitz constant of the FNN in (1) over the local region $\mathcal{Z} = \mathcal{B}(z_c, \delta_z)$, with $\delta_z > 0$.

Without loss of generality, we assume $W_i \neq 0$, $i \in \mathbf{Z}_N$, as any zero weights will lead to the trivial case where the output corresponding to any input will remain the same after that layer. The goal is to utilize local information from domain $\mathcal{Z}$ and provide a scalable approach to efficiently calculate a tight upper bound for the local Lipschitz constant $\boldsymbol{L} > 0$. Note that the proofs of all the theoretical results in this paper are included in Appendix A.1.

## 2.2 Preliminaries

We begin with a slope-restrictedness property satisfied by most activation functions, which is typically leveraged to to derive SDPs for Lipschitz certificates (Fazlyab et al. (2019); Xu & Sivaranjani (2024)).

**Assumption 1** (Slope-restrictedness). *For the neural network defined in (1), the activation function $\phi$ is slope-restricted in $[\alpha, \beta]$, $\alpha \leq \beta$ in the sense that $\forall v_1, v_2 \in \mathbb{R}^n$, we have $\alpha(v_1 - v_2) \leq \phi(v_1) - \phi(v_2) \leq \beta(v_1 - v_2)$ element-wise. Consequently, we have that for $\forall \Lambda \in \mathbb{D}_+$,*

$$\begin{bmatrix} v_1 - v_2 \\ \phi(v_1) - \phi(v_2) \end{bmatrix}^T \begin{bmatrix} p\Lambda & -m\Lambda \\ -m\Lambda & \Lambda \end{bmatrix} \begin{bmatrix} v_1 - v_2 \\ \phi(v_1) - \phi(v_2) \end{bmatrix} \leq 0, \quad p = \alpha\beta, \quad m = (\alpha + \beta)/2. \tag{2}$$

The assumption holds for all commonly used activation functions. For example, it holds with $\alpha = 0$, $\beta = 1$, that is, $p = 0, m = 1/2$ for the ReLU, sigmoid, tanh, ELU (exponential linear functions). For Leaky ReLU activation function, defined as $\phi(x) = \max(\gamma x, x)$ for some fixed $\gamma \in (0, 1)$, the assumption is satisfied with $\alpha = \gamma$ and $\beta = 1$, i.e., $p = (\gamma + 1)/2$ and $m = (1 + \gamma^2)/2$.

We first introduce LipSDP framework Fazlyab et al. (2019), which provides an accurate upper bound for the global Lipschitz constant, as follows. Note that the matrix inequality presented here has a slightly different, but mathematically equivalent to to the original formulation in Fazlyab et al. (2019).

**Theorem 1** (LipSDP). *For the FNN (1) satisfying Assumption 1, if there exists $F > 0$ and nonnegative diagonal matrices $\Lambda_i \in \mathbb{D}_+$, $i \in \mathbb{Z}_{N-1}$ such that with $p = \alpha\beta$ and $m = \frac{\alpha+\beta}{2}$,*

$$\begin{bmatrix} I + pW_1^T\Lambda_1 W_1 & -mW_1^T\Lambda_1 & 0 & \dots & \dots & 0 \\ -m\Lambda_1 W_1 & \Lambda_1 + pW_2^T\Lambda_2 W_2 & -mW_2^T\Lambda_2 & 0 & \dots & 0 \\ 0 & -m\Lambda_2 W_2 & \Lambda_2 + pW_3^T\Lambda_3 W_3 & \dots & \dots & 0 \\ & & \vdots & & & \\ 0 & \dots & 0 & -m\Lambda_{N-2}W_{N-2} & \Lambda_{N-2} + pW_{N-1}^T\Lambda_{N-1}W_{N-1} & -mW_{N-1}^T\Lambda_{N-1} \\ 0 & \dots & \dots & 0 & -m\Lambda_{N-1}W_{N-1} & \Lambda_{N-1} - FW_N^T W_N \end{bmatrix} > 0, \tag{3}$$

*then $\left\|z_2^{(N)} - z_1^{(N)}\right\|_2 \leq \sqrt{1/F}\left\|z_2^{(0)} - z_1^{(0)}\right\|_2$, which provides a sufficient condition for the Lipschitz constant $\boldsymbol{L}$ to be upper bounded by $\sqrt{1/F}$.*

Practically, LipSDP maximizes $F$ satisfying (3) to obtain a strict Lipschitz upper bound by $\sqrt{1/F}$. LipSDP also provides two variants (Fazlyab et al. (2019); Pauli et al. (2021)): LipSDP-Neuron with $\Lambda_i \in \mathbb{D}_+$ and LipSDP-Layer with $\Lambda_i = \lambda_i I$ ($\lambda_i \geq 0$), $i \in \mathbb{Z}_{N-1}$, which decrease computational complexity by reducing the number of decision variables at the cost of some accuracy. Nevertheless, solving (3) becomes exponentially costly as the number of layers increases. Recently, Xu & Sivaranjani (2024) introduced the ECLipsE framework, which provides a scalable approach by decomposing the large matrix inequality (3) into smaller sub-problems, resulting in linear computational cost with respect to the number of layers. Specifically, the ECLipsE framework provides two algorithms that tradeoff accuracy and computational efficiency: ECLipsE

solves small matrix inequalities that scale with the size of the weight matrices in consecutive layers, and EClipsE-Fast provides closed-form solutions for $\lambda_i$, $i \in \mathbb{Z}_{N-1}$, achieving further scalability at the cost of some accuracy in the Lipschitz bounds.

Despite its scalability, ECLipsE has several limitations. First, it is constrained to activation functions whose slope bounds satisfy (2) with $p = 0$, and thus can cannot accommodate several activation functions such as Leaky ReLU, PReLU (Parametric ReLU), SiLU (Sigmoid Linear Unit), and ELU (Exponential Linear Unit), which are also commonly used in practice. More importantly, ECLipsE does not incorporate any local information from the input domain and is thus limited to computing the global Lipschitz constant, which is typically less tight that a local one. Moreover, ECLipsE cannot provide bounds for the Jacobian in an element-wise or partially indexed fashion. In the following sections, we address these issues by developing more general scalable algorithms to obtain tight local Lipschitz estimates for NNs with a wide variety of activation functions.

## 3 Methodology

### 3.1 SDP-based Lipschitz Estimates with General Slope Bounds and Arbitrary Input-Output Indices

We start with allowing heterogeneous slope bounds for the activation functions for each neuron in each layer. Specifically, the slope bounds are given by two sets of vectors $\{\boldsymbol{\alpha}^i\}_{i=1}^{N-1}$ (lower) and $\{\boldsymbol{\beta}^i\}_{i=1}^{N-1}$ (upper). Then, Theorem 1 from the LipSDP framework can be generalized as follows.

**Theorem 2** (Hetereogeneous Slope Bounds). *For the FNN (1) satisfying Assumption 1, we denote the lower and upper slope bounds for the activation functions on the $l^{th}$ neuron in the $i^{th}$ layer to be $\alpha_l^i$ and $\beta_l^i$, respectively, with $l \in \mathbb{Z}_{d_i}$, $i \in \mathbb{Z}_N$. Let $\boldsymbol{\alpha}^i = \left[\alpha_1^i, \alpha_2^i, \cdots, \alpha_{d_i}^i\right]^T$ and $\boldsymbol{\beta}^i = \left[\beta_1^i, \beta_2^i, \cdots, \beta_{d_i}^i\right]^T$, $i \in \mathbb{Z}_{N-1}$. If there exists $F > 0$ and non-negative diagonal matrices $\Lambda_i \in \mathbb{D}_+$, $i \in \mathbb{Z}_{N-1}$ such that*

$$\begin{bmatrix} \mathcal{P}_1 & \mathcal{R}_2 & 0 & \cdots & 0 \\ \mathcal{R}_2^T & \mathcal{P}_2 & \mathcal{R}_3 & \cdots & 0 \\ 0 & \mathcal{R}_3^T & \mathcal{P}_3 & \cdots & 0 \\ \vdots & \vdots & \vdots & \ddots & \mathcal{R}_N \\ 0 & 0 & 0 & \mathcal{R}_N^T & \mathcal{P}_N \end{bmatrix} > 0, \tag{4}$$

*where, for $i = 1, \ldots, N$,*

$$\mathcal{P}_i = \begin{cases} I + W_1^T D_{\boldsymbol{\alpha}^1} \Lambda_1 D_{\boldsymbol{\beta}^1} W_1, & \text{if } i = 1 \\ \Lambda_{i-1} + W_i^T D_{\boldsymbol{\alpha}^i} \Lambda_i D_{\boldsymbol{\beta}^i} W_i, & 2 \leq i < N \\ \Lambda_{N-1} - F W_N^T W_N, & i = N \end{cases} \tag{5}$$

$$\mathcal{R}_i = -\frac{1}{2} W_{i-1}^T (D_{\boldsymbol{\alpha}^{i-1}} + D_{\boldsymbol{\beta}^{i-1}}) \Lambda_{i-1}, \quad i = 2, \ldots, N,$$

*then $\left\| z_2^{(l)} - z_1^{(l)} \right\|_2 \leq \sqrt{1/F} \left\| z_2^{(0)} - z_1^{(0)} \right\|_2$. This serves as a sufficient condition for the Lipschitz constant $\boldsymbol{L}$ to be upper bounded by $\sqrt{1/F}$.*

We now extend the above result to allow Lipschitz constant estimates with respect to arbitrary input-output pairs and arbitrary subsets of consecutive layers. For uniformity of notation, we denote $\Lambda_0 \triangleq I_{d_0}$. Concretely, we provide the certificates for $\mathbf{L}_{\mathcal{K},\mathcal{L}}^{(p,i)}$ such that for any $\mathcal{K} \subseteq \mathbb{Z}_{d_p}$ and $\mathcal{L} \subseteq \mathbb{Z}_{d_i}$, $0 \leq p < i \leq N$,

$$\left\| \left(v_1^{(i)}\right)_{\mathcal{L}} - \left(v_2^{(i)}\right)_{\mathcal{L}} \right\|_2 \leq \mathbf{L}_{\mathcal{K},\mathcal{L}}^{(p,i)} \left\| \left(v_1^{(p)}\right)_{\mathcal{K}} - \left(v_2^{(p)}\right)_{\mathcal{K}} \right\|_2 \tag{6}$$

holds for any possible $\left(v_1^{(p)}\right)_{\mathcal{K}}$, $\left(v_2^{(p)}\right)_{\mathcal{K}}$ and $\left(v_1^{(i)}\right)_{\mathcal{L}}$, $\left(v_2^{(i)}\right)_{\mathcal{L}}$, where $z_1^{(0)} = v_1^{(0)}$, $z_2^{(0)} = v_2^{(0)} \in \mathcal{Z} \subseteq \mathbb{R}^{d_0}$.

**Theorem 3.** *For any* $\mathcal{K} \subseteq \mathbb{Z}_{d_p}$ *and* $\mathcal{L} \subseteq \mathbb{Z}_{d_i}$, $0 \le p < i \le N$, *inequality (6) holds if there exists* $\boldsymbol{L}_{\mathcal{K},\mathcal{L}}^{(p,i)} = \sqrt{1/F} > 0$ *such that*

$$
\begin{bmatrix}
\mathcal{P}_{p+1} & \mathcal{R}_{p+2} & 0 & \cdots & 0 \\
\mathcal{R}_{p+2}^T & \mathcal{P}_{p+2} & \mathcal{R}_{p+3} & \cdots & 0 \\
0 & \mathcal{R}_{p+3}^T & \mathcal{P}_{p+3} & \cdots & 0 \\
\vdots & \vdots & \vdots & \ddots & \mathcal{R}_i \\
0 & 0 & 0 & \mathcal{R}_i^T & \mathcal{P}_i
\end{bmatrix} > 0,
\tag{7}
$$

*where*

$$
\mathcal{P}_m = \begin{cases}
(\Lambda_p)_{(\mathcal{K},\mathcal{K})} + \left((W_{p+1})_{(\bullet,\mathcal{K})}\right)^T D_{\boldsymbol{\alpha}^{p+1}} \Lambda_{p+1} D_{\boldsymbol{\beta}^{p+1}} (W_{p+1})_{(\bullet,\mathcal{K})}, & m = p+1 \\[2mm]
\Lambda_{m-1} + W_m^T D_{\boldsymbol{\alpha}^m} \Lambda_m D_{\boldsymbol{\beta}^m} W_m, & p+2 \le m < i \\[2mm]
\Lambda_{i-1} - F\left((W_i)_{(\mathcal{L},\bullet)}\right)^T (W_i)_{(\mathcal{L},\bullet)}, & m = i
\end{cases}
\tag{8}
$$

$$
\mathcal{R}_m = \begin{cases}
-\dfrac{1}{2} \left((W_{p+1})_{(\bullet,\mathcal{K})}\right)^T \left(D_{\boldsymbol{\alpha}^{p+1}} + D_{\boldsymbol{\beta}^{p+1}}\right) \Lambda_{p+1}, & m = p+2 \\[3mm]
-\dfrac{1}{2} W_{m-1}^T (D_{\boldsymbol{\alpha}^{m-1}} + D_{\boldsymbol{\beta}^{m-1}}) \Lambda_{m-1}, & p+3 \le m \le i
\end{cases}
$$

Here, arbitrary layers and indices are accommodated by retaining only the weights between the selected layers, and appropriately selecting the sub-matrices of weights in the first and last selected layers corresponding to selected input-output indices. All other components remain unchanged. This operation can be applied whenever bounds for arbitrary layers and indices are required. Note that we present the following theory for the entire NN; however, the results can be analogously extended to arbitrary subsets of layers and input-output pairs in the same manner.

To develop scalable algorithms based on this generalized SDP formulation, we build on the exact decomposition from Xu & Sivaranjani (2024); Agarwal et al. (2019), to derive sufficient and necessary conditions for the matrix inequality (4).

**Theorem 4.** *Matrix inequality (4) holds if and only if the following sequence of matrix inequalities is satisfied:*

$$
X_i > 0, \quad \forall i \in \mathbb{Z}_{N-2}, \qquad X_{N-1} - F W_N^T W_N > 0,
\tag{9}
$$

*where*

$$
X_i = \begin{cases}
I + W_1^T D_{\boldsymbol{\alpha}^1} \Lambda_i D_{\boldsymbol{\beta}^1} W_1 & i = 0 \\[2mm]
\Lambda_i - \frac{1}{4} \Lambda_i (D_{\boldsymbol{\alpha}^i} + D_{\boldsymbol{\beta}^i}) W_i (X_{i-1})^{-1} W_i^T (D_{\boldsymbol{\alpha}^i} + D_{\boldsymbol{\beta}^i}) \Lambda_i + W_{i+1}^T D_{\boldsymbol{\alpha}^{i+1}} \Lambda_{i+1} D_{\boldsymbol{\beta}^{i+1}} W_{i+1} & i \in \mathbb{Z}_{N-2} \\[2mm]
\Lambda_{N-1} - \frac{1}{4} \Lambda_{N-1} (D_{\boldsymbol{\alpha}^{N-1}} + D_{\boldsymbol{\beta}^{N-1}}) W_{N-1} (X_{N-2})^{-1} W_{N-1}^T (D_{\boldsymbol{\alpha}^{N-1}} + D_{\boldsymbol{\beta}^{N-1}}) \Lambda_{N-1} & i = N-1
\end{cases}
\tag{10}
$$

This result extends Theorem 3 of Xu & Sivaranjani (2024) to allow heterogeneous slope bounds for the activation function on every single neuron, where slope bounds $\alpha_l^i$ and $\beta_l^i$, $l \in \mathbb{Z}_{d_i}$, $i \in \mathbb{Z}_N$, can be non-zero.

To facilitate the algorithms, we further define for all $i \in \mathbb{Z}_{N-1}, i \ge 2$, a *messenger matrix*

$$
M_{i-1} \triangleq \Lambda_{i-1} - \frac{1}{4} \Lambda_{i-1} (D_{\boldsymbol{\alpha}^{i-1}} + D_{\boldsymbol{\beta}^{i-1}}) W_{i-1} (X_{i-2})^{-1} W_{i-1}^T (D_{\boldsymbol{\alpha}^{i-1}} + D_{\boldsymbol{\beta}^{i-1}}) \Lambda_{i-1}
\tag{11}
$$

For notational consistency, we set $M_0 = I$. In other words,

$$
X_{i-1} = \begin{cases}
M_{i-1} + W_i^T D_{\boldsymbol{\alpha}^i} \Lambda_i D_{\boldsymbol{\beta}^i} W_i, & i \in \mathbb{Z}_{N-1} \\
M_{i-1} & i = N.
\end{cases}
\tag{12}
$$

With the extensions mentioned above, we develop a series of algorithms, termed the ECLipsE-Gen series as follows. From (10), we observe that $X_i$ is obtained in a recursive manner and depends on $\Lambda_i$ and $X_{i-1}$,

$i \in \mathbb{Z}_{N-1}$. We propose three algorithms where we derive $\Lambda_i$ and simultaneously compute $M_i$, $i \in \mathbb{Z}_{N-1}$ in a sequential manner, laying the foundation for compositional Lipschitz estimation methods whose computational cost grows only linearly with respect to the depth of FNN. We directly present the algorithms here and deliberately defer the supporting rationale and theory in Section 3.5 for clarity.

**ECLipsE-Gen-Acc.** For the most general case where $\Lambda_i$ can have heterogeneous elements on the diagonal, we can obtain $\Lambda_i$, $i \in \mathbb{Z}_{N-1}$ at each stage $i$ using the information from the next layer, i.e. $W_{i+1}$, by solving the following **small SDP**:

$$\max_{c_i, \Lambda_i} \ c_i \quad \text{s.t.} \quad \begin{bmatrix} \Lambda_i - c_i W_{i+1}^T W_{i+1} & \frac{1}{2}\Lambda_i(D_{\boldsymbol{\alpha}^i} + D_{\boldsymbol{\beta}^i})W_i \\ \frac{1}{2}W_i^T(D_{\boldsymbol{\alpha}^i} + D_{\boldsymbol{\beta}^i})\Lambda_i & X_{i-1} \end{bmatrix} > 0, \ \Lambda_i \in \mathbb{D}_+ \ c_i > 0. \tag{13}$$

Recall from (10), with $W_i$, $D_{\boldsymbol{\alpha}^i}$, $D_{\boldsymbol{\beta}^i}$, $\Lambda_{i-1}$, and $X_{i-2}$ known at stage $i$, each block matrix above is linear in $\Lambda_i$ and $c_i$ in each block matrix. Thus, (13) is a semidefinite program (SDP) of small size, involving only the weights from two consecutive layers. With the solution $\Lambda_i$, we can directly compute $M_i$ as (11).

**ECLipsE-Gen-Fast.** In the special case where $\Lambda_i$ is relaxed to $\Lambda_i = \lambda_i I$, $i \in \mathbb{Z}_{l-1}$, and is calculated by solving

$$\max_{c_i, \lambda_i} \ c_i \quad \text{s.t.} \quad \begin{bmatrix} \lambda_i I - c_i W_{i+1}^T W_{i+1} & \frac{1}{2}\lambda_i(D_{\boldsymbol{\alpha}^i} + D_{\boldsymbol{\beta}^i})W_i \\ \frac{1}{2}\lambda_i W_i^T(D_{\boldsymbol{\alpha}^i} + D_{\boldsymbol{\beta}^i}) & \tilde{X}_{i-1} \end{bmatrix} > 0, \ \lambda_i \geq 0, \ c_i > 0, \tag{14}$$

where $\tilde{X}_{i-1}$ denotes the matrix $X_{i-1}$ with the substitution $\Lambda_i = \lambda_i I$.

In this variant, the number of decision variables at each stage is reduced from $(d_i + 1)$ to just two variables ($\lambda_i$ and $c_i$) compared to ECLipsE-Gen-Acc at each stage, resulting in decreased computational complexity, albeit at the expense of some accuracy. We use the solution $\Lambda_i = \lambda_i I$ to compute $M_i$ as (11).

**ECLipsE-Gen-CF.** If the slope bounds $\boldsymbol{\alpha}^i$ and $\boldsymbol{\beta}^i$ for each neuron do not have different signs (i.e., $\boldsymbol{\alpha}^i \odot \boldsymbol{\beta}^i \geq 0$), we have $X_{i-1} \geq M_{i-1}$, enabling further relaxation. Specifically, if $\boldsymbol{\alpha}^i \odot \boldsymbol{\beta}^i = 0$, we have $X_{i-1} = M_{i-1}$, which enables an **optimal closed-form solution** for (14). Under the assumption that $\boldsymbol{\alpha}^i \odot \boldsymbol{\beta}^i \geq 0$, we can adjust $\boldsymbol{\alpha}^i, \boldsymbol{\beta}^i$ as follows. For each $j \in \mathbb{Z}_{d_i}$, if $0 \leq (\boldsymbol{\alpha}^i)_j \leq (\boldsymbol{\beta}^i)_j$, set $(\boldsymbol{\alpha}_j^i, \boldsymbol{\beta}_j^i) \to (0, \boldsymbol{\beta}_j^i)$; if $(\boldsymbol{\alpha}^i)_j \leq (\boldsymbol{\beta}^i)_j \leq 0$, set $(\boldsymbol{\alpha}_j^i, \boldsymbol{\beta}_j^i) \to (\boldsymbol{\alpha}_j^i, 0)$. We denote the adjusted slope bounds as $\boldsymbol{\alpha}^{i,adj}$ and $\boldsymbol{\beta}^{i,adj}$. Note that after adjustment, $\boldsymbol{\alpha}^{i,adj} \odot \boldsymbol{\beta}^{i,adj} = 0$ is satisfied. This yields the optimal close-form solution for $\lambda_i$ on layer $L_i$ as

$$\lambda_i = \frac{2}{\sigma_{max}\left((D_{\boldsymbol{\alpha}^{i,\text{adj}}} + D_{\boldsymbol{\beta}^{i,\text{adj}}})W_i(M_{i-1})^{-1}W_i^T(D_{\boldsymbol{\alpha}^{i,\text{adj}}} + D_{\boldsymbol{\beta}^{i,\text{adj}}})\right)}. \tag{15}$$

With $\Lambda_i = \lambda_i I$, we can further derive the corresponding optimal $c_i$ from (14) as follows.

**Proposition 1.** *With $\Lambda_i = \lambda_i I$ as in (15), the optimal $c_i$ obtained from (14) is*

$$c_i = \frac{1}{\sigma_{\max}\left(W_{i+1}\left(M_i\right)^{-1}W_{i+1}^T\right)}, \tag{16}$$

*where $M_i$ is computed as*

$$M_i = \Lambda_i - \frac{1}{4}\Lambda_i\left(D_{\boldsymbol{\alpha}^{i,\text{adj}}} + D_{\boldsymbol{\beta}^{i,\text{adj}}}\right)W_i\left(X_{i-1}\right)^{-1}W_i^T\left(D_{\boldsymbol{\alpha}^{i,\text{adj}}} + D_{\boldsymbol{\beta}^{i,\text{adj}}}\right)\Lambda_i. \tag{17}$$

It is worth mentioning that the assumption $\boldsymbol{\alpha}^i \odot \boldsymbol{\beta}^i \geq 0$ holds for almost all commonly used activation functions. Notably, ECLipsE-Gen-CF **completely eliminates** the need to solve matrix inequality **SDPs** altogether, thus significantly enhancing computational efficiency.

After all $\Lambda_i$s, $i \in \mathbb{Z}_{N-1}$ are decided using any of the above algorithms, we obtain the smallest $1/F$ which yields the smallest Lipschitz estimate **L**, as

$$\mathbf{L} = \sqrt{1/F} = \sqrt{\sigma_{max}\left(W_N^T W_N(X_{N-1})^{-1}\right)} = \sqrt{\sigma_{max}\left(W_N(X_{N-1})^{-1}W_N^T\right)}. \tag{18}$$

Note that the second equality holds because of the Lemma 1 in (Xu & Sivaranjani (2024)), restated below.

**Lemma 1** (Lemma 1 in Xu & Sivaranjani (2024)). *If $M_{i-1} > 0$, then $W_i^T W_i (M_{i-1})^{-1}$ and $W_i (M_{i-1})^{-1} (W_i)^T$ share the same non-zero eigenvalues.*

**Proposition 2.** *For given $\Lambda_i$, $i \in \mathbb{Z}_{N-1}$ that satisfies $X_i > 0$, $i \in \mathbb{Z}_{N-2}$, the tightest upper bound for Lipschitz constant is $\mathbf{L} = \sqrt{\sigma_{max} \left( W_N (X_{N-1})^{-1} W_N^T \right)}$.*

ECLipsE-Gen-Acc provides accurate Lipschitz estimates by solving small semidefinite programs (SDPs). ECLipsE-Gen-Fast, with fewer decision variables, offers improved computational speed at the expense of some accuracy. Under very mild assumptions, ECLipsE-Gen-CF relaxes the sub-problems at each stage and yields a closed-form solution for each sub-problem that makes it extremely fast. These algorithms embody different trade-offs between efficiency and accuracy; one may choose ECLipsE-Gen-Acc, if pursuing accuracy, and ECLipsE-Gen-Fast or ECLipsE-Gen-CF (depending on the slope of the activation function), for applications where scalability is of the essence. Note that ECLipsE-Gen-CF involves adjustment of slope bounds as a relaxation to achieve closed-form solutions, which can introduce additional conservatism. The slack introduced is discussed in Appendix A.4.

Having introduced the three algorithms, we now describe how to generalize each to arbitrary subsets of layers and input-output pairs. To extend ECLipsE-Gen-Acc, ECLipsE-Gen-Fast, and ECLipsE-Gen-CF, we simply isolate the sub-network between the chosen layers and appropriately select the associated weights and $\Lambda$ variables. Specifically, when estimating $\mathbf{L}_{\mathcal{K},\mathcal{L}}^{(p,i)}$, $p < i$, $\mathcal{K} \subseteq \mathbb{Z}_{d_p}$, $\mathcal{L} \subseteq \mathbb{Z}_{d_i}$, we keep only $W_i$, $i = p+1, ..., i$ and $\Lambda_i$, $i = p+1, ..., i-1$ , substitute $W_{p+1}$ and $W_i$ with $(W_{p+1})_{(\bullet, \mathcal{K})}$ and $(W_i)_{(\mathcal{L}, \bullet)}$, and similarly substitute $\Lambda_p$ and $\Lambda_{i-1}$ with $(\Lambda_p)_{(\mathcal{K},\mathcal{K})}$ and $(\Lambda_{i-1})_{\mathcal{L},\mathcal{L}}$. The sequence of small sub-problems start at stage $p+1$. The remainder of the procedure and the structure of the optimization problem for each algorithm remain unchanged.

*Remark* 1. All these extensions are essential not only for exploiting local properties of the input domain, as will be illustrated in the following section, but also for facilitating the modular/compositional analysis and decomposition of NNs, paving the way for scalable certification of NNs in a wide variety of applications.

## 3.2 Utilizing Local Information

The key to utilizing the local information to obtain tighter Lipschitz estimates is to refine the slope bounds of the activation function for each individual neuron. Intuitively, more accurate (i.e., narrower) slope bounds lead to tighter Lipschitz estimates. This intuition can be formalized as follows.

**Theorem 5** (Monotonicity of Estimates with Respect to Slope Bounds). *Consider two sets of slope bounds for the activation functions, $\{\hat{\boldsymbol{\alpha}}^i, \hat{\boldsymbol{\beta}^i}\}_{i=1}^{N-1}$ and $\{\tilde{\boldsymbol{\alpha}}^i, \tilde{\boldsymbol{\beta}}^i\}_{i=1}^{N-1}$, where for some $j, i$, $[\tilde{\boldsymbol{\alpha}}_j^i, \tilde{\boldsymbol{\beta}}_j^i] \subset [\hat{\boldsymbol{\alpha}}_j^i, \hat{\boldsymbol{\beta}}_j^i]$ and for all other entries the slope bounds are identical, i.e. $\hat{\boldsymbol{\alpha}}^i = \tilde{\boldsymbol{\alpha}}^i$, $\hat{\boldsymbol{\beta}}^i = \tilde{\boldsymbol{\beta}}^i$. Let $F$ and $\tilde{F}$ denote the maximum values that satisfy (4)-(5) when using $\{\hat{\boldsymbol{\alpha}}^i, \hat{\boldsymbol{\beta}}^i\}$ and $\{\tilde{\boldsymbol{\alpha}}^i, \tilde{\boldsymbol{\beta}}^i\}$, respectively. Then we have $\tilde{F} \geq \hat{F}$, and thus the corresponding Lipschitz upper bound satisfies $\sqrt{1/\tilde{F}} \leq \sqrt{1/\hat{F}}$.*

In other words, using narrower slope bounds for the activation functions yields a tighter upper bound on the Lipschitz constant in Theorem 2. To obtain narrower slope bounds for each individual neuron, we leverage both the input region and the Lipschitz bound with respect to $v_j^{(i)}$, $j \in \mathbb{Z}_{d_i}, i \in \mathbb{N}$, and the input layer. Specifically, we use a first-order method based on the mean value theorem to estimate the range of output values at each neuron and then refine the slope bounds corresponding to various classes of activation functions.

**Theorem 6** (First-order Method via Mean Value Theorem). *We consider any $z \in \mathcal{Z} \triangleq \mathcal{B}(z_c, \delta_z)$, where $\delta_z > 0$. Let $g : \mathbb{R}^{d_0} \to \mathbb{R}$ be a locally Lipschitz continuous function over $\mathcal{Z}$ with Lipschitz constant $\boldsymbol{L} > 0$. Then for all $z \in \mathcal{Z}$,*

$$|g(z) - g(z_c)| \leq \boldsymbol{L} \|\delta z\|_2. \tag{19}$$

*In other words, we have the range for $g(z)$, $z \in \mathcal{Z}$, as*

$$g(z_c) - \boldsymbol{L} \|\delta z\|_2 \leq g(z) \leq g(z_c) + \boldsymbol{L} \|\delta z\|_2 \tag{20}$$

We can now utilize Theorem 6 to bound $v_j^{(i)}$, $j \in \mathbb{Z}_{d_i}, i \in \mathbb{N}$ at each neuron and refine the slope bounds.

**Proposition 3.** *Consider layer $L_i$ of FNN (1), $i \in \mathbb{Z}_{N-1}$. Let $\phi$ denote the activation function. If we have $v_l^{(i)} \in [a, b]$, $l \in \mathbb{Z}_{d_i} \subset \mathbb{R}$, then the refined slope bounds for neuron $l$ in layer $L_i$ are given by*

$$\alpha_l^i = \inf_{v \in [a,b]} \inf\{\partial\phi(v)\}, \qquad \beta_l^i = \sup_{v \in [a,b]} \sup\{\partial\phi(v)\} \tag{21}$$

*where $\phi'(v)$ denotes the subdifferential of $\phi$ at $v$.*

Given activation function $\phi$, $\alpha_l^i$ and $\beta_l^i$ (briefly $\alpha$ and $\beta$ here) can be computed explicitly. We present a few representative examples below.

*ReLU:* $\phi(x) = \max\{0, x\}$. The subdifferential is

$$\partial\phi(x) = \begin{cases} \{0\}, & x < 0 \\ [0, 1], & x = 0 \\ \{1\}, & x > 0 \end{cases}$$

Therefore, we have $\alpha = 0$ and $\beta = 1$.

*Tanh:* $\phi(x) = \tanh(x)$. The subdifferential is $\partial\phi(x) = \{1 - \tanh^2(x)\}$. Therefore,

$$\alpha = 1 - \tanh^2\left(\max\{|a|, |b|\}\right), \qquad \beta = 1 - \tanh^2\left(\min\{|a|, |b|\}\right)$$

*Sigmoid:* $\phi(x) = (1 + e^{-x})^{-1}$. The subdifferential is $\partial\phi(x) = \{\phi(x)[1 - \phi(x)]\}$. Therefore,

$$\alpha = \min\left\{\phi(a)[1 - \phi(a)], \ \phi(b)[1 - \phi(b)]\right\}, \qquad \beta = 0.25$$

*Leaky ReLU:* $\phi(x) = \max\{\gamma x, x\}$, with $\gamma \in (0, 1)$. The subdifferential is

$$\partial\phi(x) = \begin{cases} \{\gamma\}, & x < 0 \\ [\gamma, 1], & x = 0 \\ \{1\}, & x > 0 \end{cases}$$

Therefore, we have $\alpha = \gamma$ and $\beta = 1$.

*ELU:* $\phi(x) = \begin{cases} x, & x \geq 0 \\ \gamma(e^x - 1), & x < 0 \end{cases}$. The subdifferential is

$$\partial\phi(x) = \begin{cases} \{1\}, & x > 0 \\ \{\gamma e^x\}, & x < 0 \\ [\gamma, 1], & x = 0 \end{cases}$$

Therefore,

$$\alpha = \begin{cases} \gamma, & b \leq 0 \\ \gamma, & a < 0 < b \ \text{or} \ a \leq 0 \leq b \ , \\ 1, & a \geq 0 \end{cases} \qquad \beta = \begin{cases} \gamma, & b < 0 \\ 1, & a < 0 < b \ \text{or} \ b \geq 0 \end{cases}$$

### 3.3 ECLipsE-Gen-Local: Scalable and Accurate Algorithm for Local Lipschitz Estimates

With the generalized algorithm series ECLipsE-Gen for given slope bounds presented in Section 3.1 and the approach to utilize the local information of the input region to refine slope bounds as described in Section 3.2, we now derive the compositional algorithm series ECLipsE-Gen-Local for estimating local Lipschitz constants. The key idea is to refine the slope bounds layer-by-layer in conjunction with the determination of each $\Lambda_i$, $i \in \mathbb{Z}_{N-1}$. Intuitively, while messenger matrices propagate coupling information, local information is simultaneously propagated as we reach each new layer by computing the reachable local region and encoding this information into refined slope bounds.

Specifically, at each stage $i$, given messenger matrix $M_{i-1}$ and $W_i$, we first calculate $\mathbf{L}_{\bullet,l}^{(0,i)}$, for each $l \in \mathbb{Z}_{d_i}$ as

$$\mathbf{L}_{\bullet,l}^{(0,i)} = \sqrt{\sigma_{max}\left( \left[(W_i)_{(l,\bullet)}\right]^T (W_i)_{(l,\bullet)} (M_{i-1})^{-1}\right)}, \tag{22}$$

Let $\mathbf{L}^{(i)} = \left[\mathbf{L}_{\bullet,1}^{(0,i)}, \mathbf{L}_{\bullet,2}^{(0,i)}, \cdots, \mathbf{L}_{\bullet,d_i}^{(0,i)}\right]^T \in \mathbb{R}^{d_i}$. We claim that we can compute $\mathbf{L}_{\bullet,l}^{(0,i)}$ for all $l \in \mathbb{Z}_{d_i}$ simultaneously, accelerating the estimating process, $i \in \mathbb{Z}_{N-1}$.

**Proposition 4.** *For any $i \in \mathbb{Z}_{N-1}$, let $W_i \in \mathbb{R}^{d_i \times d_{i-1}}$ and $M_{i-1} \in \mathbb{R}^{d_{i-1} \times d_{i-1}}$ be symmetric positive definite. Then, the $l$-th diagonal entry of the matrix $W_i(M_{i-1})^{-1}W_i^T$ yields:*

$$\left(W_i(M_{i-1})^{-1}W_i^T\right)_{(l,l)} = \sigma_{\max}\left(\left[(W_i)_{(l,\bullet)}\right]^T (W_i)_{(l,\bullet)}(M_{i-1})^{-1}\right), \tag{23}$$

In other words, to obtain $\mathbf{L}^{(i)}$, it suffices to compute the matrix $W_i(M_{i-1})^{-1}W_i^T$ and take its diagonal entries. In fact, since only the diagonal entries are needed, further computational acceleration is possible.

**Lemma 2.** *Denote $\mathbf{d}_l^{(i)} = \left(W_i(M_{i-1})^{-1}W_i^T\right)_{(l,l)}$ for $l = 1, \ldots, d_i$ and $\mathbf{d}^{(i)} = \left[\mathbf{d}_1^{(i)}, \mathbf{d}_2^{(i)}, \ldots, \mathbf{d}_{d_i}^{(i)}\right]^T$. Let $A_i = (M_{i-1})^{-1}W_i^T \in \mathbb{R}^{d_{i-1} \times d_i}$.*

$$\mathbf{d}^{(i)} = \sum_{k=1}^{d_{i-1}} (W_i)_{(\bullet,k)} \odot \left((A_i)_{(k,\bullet)}\right)^T \tag{24}$$

This means that, in practice, the diagonal entries can be computed efficiently in a vectorized fashion, thereby avoiding computation of the entire matrix.

In the next step, we combine the input region $\mathcal{Z} = \mathcal{B}(z_c, \delta_z)$ and $f^{(i)}$, $i \in \mathbb{Z}_{N-1}$, to enable refinement on the slope bound of each neuron layer by layer. By the structure of neural network (1), it is natural to apply $f^{(i)}$ in a recursive manner to avoid repeated calculation. Specifically, let $v^{c,(i)} \triangleq f^{(i)}(z_c)$, $i \in \{0\} \cup \mathbb{Z}_N$. Then, starting with $v^{c,(0)} = f^{(0)}(z_c) = z_c$, we calculate for $i \in \mathbb{Z}_N$,

$$v^{c,(i)} = f^{(i)}(z_c) = \phi(W_i f^{(i-1)}(z_c) + b_i). \tag{25}$$

According to Theorem 6, the ranges for the values on neurons are given by:

$$v^{(i)} \in \mathcal{V}^i \triangleq \left[v^{c,(i)} - \|\delta z\|_2 \mathbf{L}^{(i)}, v^{c,(i)} + \|\delta z\|_2 \mathbf{L}^{(i)}\right]. \tag{26}$$

We then refine slope bounds for all the neurons on layer $L_i$ according to Proposition 3 as

$$\boldsymbol{\alpha}^i = \left[\inf_{v^{(i)} \in \mathcal{V}^i} \inf \left\{\partial\sigma\left(\left(v^{(i)}\right)_1\right)\right\}, \inf_{v^{(i)} \in \mathcal{V}^i} \inf \left\{\partial\sigma\left(\left(v^{(i)}\right)_2\right)\right\}, \cdots, \inf_{v^{(i)} \in \mathcal{V}^i} \inf \left\{\partial\sigma\left(\left(v^{(i)}\right)_{d_i}\right)\right\}\right]^T,$$

$$\boldsymbol{\beta}^i = \left[\sup_{v^{(i)} \in \mathcal{V}^i} \sup \left\{\partial\sigma\left(\left(v^{(i)}\right)_1\right)\right\}, \sup_{v^{(i)} \in \mathcal{V}^i} \sup \left\{\partial\sigma\left(\left(v^{(i)}\right)_2\right)\right\}, \cdots, \sup_{v^{(i)} \in \mathcal{V}^i} \sup \left\{\partial\sigma\left(\left(v^{(i)}\right)_{d_i}\right)\right\}\right]^T. \tag{27}$$

Consequently, we determine $\Lambda_i$ based on $M_{i-1}$, $W_i$, and the refined slope bounds $\boldsymbol{\alpha}^i$, $\boldsymbol{\beta}^i$, using any of the algorithms ECLipsE-Gen-Acc, ECLipsE-Gen-Fast, or ECLipsE-Gen-CF, and subsequently compute $M_i$. This process is repeated iteratively for each layer, starting with $M_0 = I$. When it comes to the last layer, where we already have $\Lambda_{N-1}$, and $X_{N-1} = M_{N-1}$, and the final Lipschitz bound is simply computed as (18).

Note that at each stage, we have the flexibility to choose any variant from the ECLipsE-Gen series. The algorithms are formally summarized in Algorithm 1, with the theoretical justification in Section 3.5.

***Remark* 2.** (Guidance on Variant Selection). The selection of variants within the ECLipsE-Gen series depends on the desired trade-off between tightness and computational cost: we choose ECLipsE-Gen-Local-CF

(if applicable) when speed is critical (e.g. online tasks), ECLipsE-Gen-Local-Acc when precision is paramount (e.g., safety certification), and ECLipsE-Gen-Local-Fast for a balanced tradeoff. Practically, we recommend starting with ECLipsE-Gen-Local-CF for online tasks due to its negligible cost. If the resulting bound is too conservative, one can upgrade to ECLipsE-Gen-Local-Fast or ECLipsE-Gen-Local-Acc. Additionally, the sequential nature of the algorithm allows users to estimate the total runtime after processing the first layer (as the computation time scales linearly with respect to the depth of NN), and accordingly choose the most accurate variant that fits their time budget.

---

**Algorithm 1** ECLipsE-Gen-Local: Scalable Local Lipschitz Estimation

---

1: **Input:** Weights $\{W_i\}_{i=1}^N$, biases $\{b_i\}_{i=1}^N$; activation function $\sigma$; input region $\mathcal{Z} = \mathcal{B}(z_c, \delta_z)$; variant $\text{ALGO} \in \{\texttt{Acc}, \texttt{Fast}, \texttt{CF}\}$

2: **Output:** Local Lipschitz estimate $\mathbf{L}$

3: Set $M_0 \leftarrow I$, $v^{c,(0)} \leftarrow z_c$

4: **for** $i = 1, 2, \ldots, N-1$ **do**

5:     Compute $\mathbf{d}^{(i)}$ with $\mathbf{d}_l^{(i)} = (W_i(M_{i-1})^{-1}W_i^T)_{(l,l)}$ for $l = 1, \ldots, d_i$, using acceleration techniques (24)

6:     Set $\mathbf{L}^{(i)} \leftarrow \left[\sqrt{\mathbf{d}_1^{(i)}}, \ldots, \sqrt{\mathbf{d}_{d_i}^{(i)}}\right]^T$

7:     Compute $v^{c,(i)} = f^{(i)}(z_c)$ per (25)

8:     Calculate range $\mathcal{V}^i$ for $v^{(i)}$ as in (26)

9:     Refine $\boldsymbol{\alpha}^i$, $\boldsymbol{\beta}^i$ using $\mathcal{V}^i$ as in (27)

10:     **if** Choose $\texttt{Acc/Fast}$ **then**

11:         Obtain $\Lambda_i$ via (13) or (14) using refined slope bounds $\boldsymbol{\alpha}^i$, $\boldsymbol{\beta}^i$

12:         Update $M_i$ as in (11) using $\boldsymbol{\alpha}^i$, $\boldsymbol{\beta}^i$

13:     **else if** Choose $\texttt{CF}$ **then**

14:         **Assert** $\boldsymbol{\alpha}^i \odot \boldsymbol{\beta}^i \geq 0$

15:         **for** $j = 1, 2, \ldots, d_i$ **do**

16:             **if** $0 \leq (\boldsymbol{\alpha}^i)_j \leq (\boldsymbol{\beta}^i)_j$ **then**

17:                 $(\boldsymbol{\alpha}^{i,\text{adj}})_j \leftarrow 0, \quad (\boldsymbol{\beta}^{i,\text{adj}})_j \leftarrow (\boldsymbol{\beta}^i)_j$

18:             **else if** $(\boldsymbol{\alpha}^i)_j \leq (\boldsymbol{\beta}^i)_j \leq 0$ **then**

19:                 $(\boldsymbol{\alpha}^{i,\text{adj}})_j \leftarrow (\boldsymbol{\alpha}^i)_j, \quad (\boldsymbol{\beta}^{i,\text{adj}})_j \leftarrow 0$

20:             **end if**

21:         **end for**

22:         Obtain $\lambda_i$ via (15) using $\boldsymbol{\alpha}^{i,\text{adj}}$ and $\boldsymbol{\beta}^{i,\text{adj}}$

23:         Set $\Lambda_i \leftarrow \lambda_i I$

24:         Compute $M_i$ as in (17) using $\Lambda_i$

25:     **end if**

26: **end for**

27: Using (18), compute final $\mathbf{L} = \sqrt{1/F} = \sqrt{\sigma_{max}\left(W_N(X_{N-1})^{-1}W_N^T\right)}$ with $X_{N-1} = M_{N-1}$

28: **return** $\mathbf{L}$

---

### 3.4 Acceleration and Stability Safeguards

We augment ECLipsE-Gen-Local with targeted accelerations for special cases and introduce stability safeguards for reliable performance in degenerate slope-bound scenarios, resulting in faster and more robust algorithms.

#### 3.4.1 Acceleration in Special Cases

***Affine Layers.*** In the special case of $\boldsymbol{\alpha}^i = \boldsymbol{\beta}^i$, layer $L_i$ becomes an affine layer. In this setting, we skip the layer $L_i$ and construct a new equivalent layer with weight $W_{i+1}$ and bias $b_{i+1}$ defined as

$$\tilde{W}_{i+1} = W_{i+1}D_{\boldsymbol{\alpha}^i}W_i. \quad \tilde{b}_{i+1} = W_{i+1}D_{\boldsymbol{\alpha}^i}b_i + b_{i+1}. \tag{28}$$

If there exist consecutive layers $L_j$, $j = i, i+1, ..., i+p$ such that all of them are affine, i.e. $\boldsymbol{\alpha}^j = \boldsymbol{\beta}^j$ for $j = i, i+1, ..., i+p$, we repeat this process for $p$ times. In other words, we skip layers $L_j$, $j = i, i+1, ..., i+p-1$. and directly proceed to construct a new equivalent layer $L_{i+p}$ as follows.

**Proposition 5.** *Let $\{L_j\}_{j=i}^{i+p}$ denote a sequence of consecutive affine layers, where for each $L_j$, $\boldsymbol{\alpha}^j = \boldsymbol{\beta}^j$ for $j = i, \ldots, i+p$. Then, these $(p+1)$ affine layers are equivalent to a single layer, denoted $\widetilde{L}_{i+p}$, with weight matrix and bias vector given by:*

$$\tilde{W}_{i+p} = \left( \prod_{j=i}^{i+p-1} W_{j+1} D_{\boldsymbol{\alpha}^j} \right) W_i. \quad \tilde{b}_{i+p} = \sum_{k=1}^{p+1} \left[ \prod_{j=i+k-1}^{i+p-1} (W_{j+1} D_{\boldsymbol{\alpha}^j}) \, b_j \right], \tag{29}$$

*where the product $W_{j+1} D_{\boldsymbol{\alpha}^j}$ reduces to the identity matrix if $k = p+1$.*

Note that, in Algorithm 1, only the computation of $f^{(i)}(z_c)$ in step 7 involves the biases $b_i$, and the value of $f^{(i)}(z_c)$ remains unchanged regardless of whether any layers are skipped. Therefore, whenever a sequence of consecutive layers is affine, we retain the computation of $f^{(i)}(z_c)$ as before, and for all other steps in Algorithm 1, we skip the intermediate layers and directly reach layer $L_{i+p}$, replacing the weights with the equivalent weight $\tilde{W}_{i+p}$ as in (29).

### 3.4.2 Numerical Instability in Degenerate Slope Bounds

Although the feasibility of optimization problems (13) and (14) is theoretically guaranteed (as will be discussed in Section 3.5), numerical issues can arise in cases where the entries of $\boldsymbol{\alpha}_i$ and $\boldsymbol{\beta}_i$ coincide partially. This scenario commonly arises in local Lipschitz estimation, particularly for piecewise linear activation functions such as ReLU and LeakyReLU, where the slope remains constant over certain regions. Let $\mathcal{J}_i \subseteq \mathbb{Z}_{d_i}$ be the index set where $(\boldsymbol{\alpha}^i)_{\mathcal{J}_i} = (\boldsymbol{\beta}^i)_{\mathcal{J}_i}$ and define $\mathcal{M}_i \triangleq \mathbb{Z}_{d_i} \backslash \mathcal{J}_i$. Note that if $\mathcal{J}_i = \mathbb{Z}_{d_i}$ (i.e., $\boldsymbol{\alpha}^i = \boldsymbol{\beta}^i$), then layer $L_i$ is affine; in this reduced case we directly apply the acceleration introduced in Section 3.4.1. Here we focus on the case $\mathcal{J}_i \subsetneq \mathbb{Z}_{d_i}$ and $\mathcal{M}_i \triangleq \mathbb{Z}_{d_i} \backslash \mathcal{J}_i \neq \emptyset$.

***ECLipsE-Gen-Acc.*** Intuitively, when $\boldsymbol{\alpha}_i$ and $\boldsymbol{\beta}_i$ coincide at an index set $\mathcal{J}_i \subseteq \mathbb{Z}_{d_i}$, the value of $(\Lambda_i)_{(\mathcal{J}_i, \mathcal{J}_i)}$ is not upper-bounded by the constraints and can grow arbitrarily large. As a result, directly solving (13) can lead the optimization solver to assign extremely large values to $(\Lambda_i)_{(\mathcal{J}_i, \mathcal{J}_i)}$, in stark contrast to the other diagonal entries. This scale disparity can introduce significant numerical instability, especially after multiple iterations. In the following, we formally characterize the source of this potential numerical issue and present a practical remedy.

**Proposition 6** (Unboundedness of $\Lambda_i$ on Equal Slope Bounds Subset). *Consider the optimization problem (13) at layer $i \in \mathbb{Z}_{N-1}$. Let $\mathcal{J}_i \subseteq \mathbb{Z}_{d_i}$ be an index subset for which the slope bounds satisfy $(\boldsymbol{\alpha}^i)_{\mathcal{J}_i} = (\boldsymbol{\beta}^i)_{\mathcal{J}_i}$. Then there exists a constant $l > 0$ such that when $(\Lambda_i)_{(\mathcal{J}_i, \mathcal{J}_i)} = l I_{d_i}$, the optimal value $c_i$ is attained. Moreover, for any $(\Lambda_i)_{(\mathcal{J}_i, \mathcal{J}_i)} \geq l I_{d_i}$, the value $c_i$ remains optimal and unchanged. In other words, the block $(\Lambda_i)_{(\mathcal{J}_i, \mathcal{J}_i)}$ is unbounded above at optimality without affecting the maximal $c_i$.*

Based on Proposition 6, we propose the following method to obtain $\Lambda_i$ at stage $i$, $i \in \mathbb{Z}_{N-1}$.

We first obtain $(\Lambda_i)_{(\mathcal{M}_i, \mathcal{M}_i)}$ similarly as (13).

$$\max_{c_i, (\Lambda_i)_{(\mathcal{M}_i, \mathcal{M}_i)}} c_i \quad \text{s.t.} \begin{bmatrix} (\Lambda_i)_{(\mathcal{M}_i, \mathcal{M}_i)} - c_i (W_{i+1}^T W_{i+1})_{(\mathcal{M}_i, \mathcal{M}_i)} & \frac{1}{2} (\Lambda_i)_{(\mathcal{M}_i, \mathcal{M}_i)} (D_{\boldsymbol{\alpha}^i} + D_{\boldsymbol{\beta}^i})_{(\mathcal{M}_i, \mathcal{M}_i)} (W_i)_{(\mathcal{M}_i, \bullet)} \\ \frac{1}{2} [(W_i)_{(\mathcal{M}_i, \bullet)}]^T (D_{\boldsymbol{\alpha}^i} + D_{\boldsymbol{\beta}^i})_{(\mathcal{M}_i, \mathcal{M}_i)} (\Lambda_i)_{(\mathcal{M}_i, \mathcal{M}_i)} & \hat{X}_{i-1} \end{bmatrix} > 0,$$
$$(\Lambda_i)_{(\mathcal{M}_i, \mathcal{M}_i)} \in \mathbb{D}_+, \qquad c_i > 0,$$
$$\tag{30}$$

where $\hat{X}_{i-1} = M_{i-1} + [(W_i)_{(\mathcal{M}_i, \bullet)}]^T (D_{\boldsymbol{\alpha}^i})_{(\mathcal{M}_i, \mathcal{M}_i)} (\Lambda_i)_{(\mathcal{M}_i, \mathcal{M}_i)} (D_{\boldsymbol{\beta}^i})_{(\mathcal{M}_i, \mathcal{M}_i)} (W_i)_{(\mathcal{M}_i, \bullet)}$.

Then, to avoid numerical issues, we ensure that all the elements of $\Lambda_i$ are of similar scale by setting

$$(\Lambda_i)_{j,j} = \frac{l_i}{|\mathcal{M}_i|} \sum_{m \in \mathcal{M}_i} (\Lambda_i)_{(m,m)}, \qquad j \in \mathcal{J}_i, \tag{31}$$

where $l_i$ is a moderately large scalar chosen to avoid numerical instability due to scale differences.

***ECLipsE-Gen-Fast.*** Similar to the ECLipsE-Gen-Acc case, we keep only the part corresponding to the index set $\mathcal{M}_i = \mathbb{Z}_{d_i} \setminus \mathcal{J}_i \neq \emptyset$ and solve

$$\max_{c_i, \bar{\lambda}_i} c_i \quad \text{s.t.} \quad \begin{bmatrix} \bar{\lambda}_i I - c_i (W_{i+1}^T W_{i+1})_{(\mathcal{M}_i, \mathcal{M}_i)} & \frac{1}{2}\bar{\lambda}_i (D_{\boldsymbol{\alpha}^i} + D_{\boldsymbol{\beta}^i})_{(\mathcal{M}_i, \mathcal{M}_i)}(W_i)_{(\mathcal{M}_i, \bullet)} \\ \frac{1}{2}\bar{\lambda}_i [(W_i)_{(\mathcal{M}_i, \bullet)}]^T (D_{\boldsymbol{\alpha}^i} + D_{\boldsymbol{\beta}^i})_{(\mathcal{M}_i, \mathcal{M}_i)} & \bar{X}_{i-1} \end{bmatrix} > 0, \tag{32}$$
$$\bar{\lambda}_i \geq 0 \ c_i > 0,$$

where $\bar{X}_{i-1} = M_{i-1} + \bar{\lambda}_i [(W_i)_{(\mathcal{M}_i, \bullet)}]^T (D_{\boldsymbol{\alpha}^i})_{(\mathcal{M}_i, \mathcal{M}_i)}(D_{\boldsymbol{\beta}^i})_{(\mathcal{M}_i, \mathcal{M}_i)}(W_i)_{(\mathcal{M}_i, \bullet)}$. Then we take $\Lambda_i = \bar{\lambda}_i I$.

***Remark*** 3. As $\mathcal{M}_i \neq \emptyset$, both optimization problems (30) and (32) are well-defined.

### 3.4.3 Numerical Feasibility Verification and Stability Safeguards

Despite the fact that theoretical feasibility is guaranteed (discussed shortly in Section 3.5), in practice SDP solvers may occasionally fail to converge to a truly optimal solution due to finite-precision issues. To address these issues, we employ the following practical procedure at each layer:

(i) For ECLipsE-Gen-Fast, with a candidate $\Lambda_i$ at layer $L_i$ obtained by solving (32), we explicitly verify whether the block matrix constraint is satisfied. If not, we switch to ECLipsE-Gen-CF for layer $L_i$. Note that as ECLipsE-Gen-CF provides a closed-form solution, it does not suffer numerical issues and always yields a valid solution.

(ii) Similarly for ECLipsE-Gen-Acc, with a candidate $\Lambda_i$ at layer $L_i$ obtained by solving (30), we explicitly verify whether the block matrix constraint is satisfied. If not, at layer $L_i$ we select from ECLipsE-Gen-Fast and ECLipsE-Gen-CF the algorithm that yields the larger feasible $c_i$ with the block matrix being strictly positive as a substitute. Note that if ECLipsE-Gen-Fast also fails for the block matrix constraint verification, we directly use the results from ECLipsE-Gen-CF.

(iii) For numerical stability, we impose an upper bound on the magnitude of $\Lambda_i$ for all layers.

These procedures ensure robust feasibility and numerical stability throughout the algorithm, even in the presence of solver limitations or degeneracies in the slope bounds. For clarity, we summarize these improvements in a separate algorithm with the full pseudocode deferred to Appendix A.2 for brevity of exposition.

### 3.5 Theoretical Guarantees and Mathematical Intuition

This section establishes theoretical guarantees for the feasibility of the algorithm and for the resulting estimates serving as provable upper bounds on the true Lipschitz constant, and explains the underlying intuition behind the algorithms in the ECLipsE-Gen and ECLipsE-Gen-Local series.

We first show that steps that involve solving SDPs in Algorithm 1 are always feasible under mild conditions.

**Theorem 7.** *Let $\boldsymbol{\alpha}^i, \boldsymbol{\beta}^i$ be the refined slope bounds at each stage $i \in \mathbb{Z}_{N-1}$ in Algorithm 1. If $\boldsymbol{\alpha}^i \odot \boldsymbol{\beta}^i \geq 0$ for all $i \in \mathbb{Z}_{N-1}$, then at every stage $i$, the optimization problems (13) and (14) are always feasible, and the closed-form solution (15) is always well-defined and positive. Thus, the corresponding $\Lambda_i$ can be properly determined at each stage, regardless of the algorithmic variant chosen.*

**Remark** 4. The condition $\boldsymbol{\alpha}^i \odot \boldsymbol{\beta}^i \geq 0$ is a very mild assumption. For all commonly used activation functions, such as ReLU, sigmoid, tanh, ELU, and leaky ReLU, the global slope bounds satisfy this property, since both lower and upper bounds are nonnegative for all possible intervals. Moreover, any refined local slope bounds, being subintervals of the global range, will also satisfy $\boldsymbol{\alpha}^i \odot \boldsymbol{\beta}^i \geq 0$.

We then establish the provable strictness and validity of all Lipschitz upper bounds and the refined slope bounds generated in Algorithm 1. Specifically, we show that (i) all slope bounds $\boldsymbol{\alpha}^i, \boldsymbol{\beta}^i$ and all intervals $\mathcal{V}^i$ computed at each layer are guaranteed to hold for any $z \in \mathcal{Z}$; and (ii) $\mathbf{L}^{(i)} = \left[ \mathbf{L}_{\bullet,1}^{(0,i)}, \mathbf{L}_{\bullet,2}^{(0,i)}, \cdots, \mathbf{L}_{\bullet,d_i}^{(0,i)} \right]^T$ and the final local Lipschitz estimates $\mathbf{L}$ from Algorithm 1 are strict, provable upper bounds for the corresponding Lipschitz constants over the region $\mathcal{Z}$. We have the following results.

**Theorem 8.** *Let $\mathcal{Z} = \mathcal{B}(z_c, \delta_z)$ be the input region. For any layer $L_i, i \in \mathbb{Z}_{N-1}$, if $\mathbf{L}_{\bullet,l}^{(i)}$ is valid in the sense that (6) holds for $l \in \mathbb{Z}_{d_{i-1}}$, then the range $\mathcal{V}^i$ and the corresponding refined slope bounds $\boldsymbol{\alpha}^i, \boldsymbol{\beta}^i$ produced by*

*Algorithm 1 are valid, that is, for $\forall z \in \mathcal{Z}$*

$$v^{(i)} \in \mathcal{V}^i, \qquad \boldsymbol{\alpha}^i \leq \inf\left\{\partial\sigma\left(v^{(i)}\right)\right\}, \qquad \boldsymbol{\beta}^i \geq \sup\left\{\partial\sigma\left(v^{(i)}\right)\right\}$$

**Theorem 9.** *Let $\mathcal{Z} = \mathcal{B}(z_c, \delta_z)$ be the input region. For any layer $L_i, i \in \mathbb{Z}_{N-1}$ and any neuron $l \in \mathbb{Z}_{d_i}$, the Lipschitz estimate $\boldsymbol{L}^{(0,i)}_{\bullet,l}$ produced by Algorithm 1 satisfies (7). In other words,*

$$\left|\left(f^{(i)}(z_1)\right)_l - \left(f^{(i)}(z_2)\right)_l\right| \leq \boldsymbol{L}^{(0,i)}_{\bullet,l}\|z_1 - z_2\|_2, \quad \forall z_1, z_2 \in \mathcal{Z}. \tag{33}$$

*Moreover, the final local Lipschitz constant $\boldsymbol{L}$ estimated by Algorithm 1 satisfies (4) with $\boldsymbol{L} = \sqrt{1/F}$, ensuring the strictness of the Lipschitz upper bound $\boldsymbol{L}$.*

Notice that $\mathbf{L}^{(0,1)}_{\bullet,l}$, $l \in \mathbb{Z}_{d_1}$, does not rely on any slope bounds. Consequently, its validity establishes the foundation to guarantee, via recursion, that all subsequent slope bounds and Lipschitz constants remain valid throughout the process according to Theorem 8 and Theorem 9.

Now we explain the underlying intuition behind the design of our algorithms. Specifically, we aim to decide appropriate $\Lambda_i$s at each stage that will translate to a tighter Lipschitz estimate at the output layer. At stage $i$, we have a messenger matrix $M_{i-1}$ that encapsulates information from all previous $i-1$ layers, as well as the weight matrices of the current and subsequent layers, $W_i$ and $W_{i+1}$. We analyze backwards, starting at the output layer. Recalling (18), we aim to find the largest $F$, or equivalently, minimize $\sigma_{max}\left(W_N(X_{N-1})^{-1}W_N^T\right) = \sigma_{max}\left(W_N^T W_N(X_{N-1})^{-1}\right)$. Therefore, at stage $i = N-1$ in Step 4 of Algorithm 1, when deciding $\Lambda_{N-1}$, we solve the following problem:

$$\max_{c_{N-1},\Lambda_{N-1}} c_{N-1} \quad \text{s.t. } X_{N-1} \geq c_{N-1}W_N^T W_N,$$
$$X_{N-1} = \Lambda_{N-1} - \frac{1}{4}\Lambda_{N-1}(D_{\boldsymbol{\alpha}^{N-1}} + D_{\boldsymbol{\beta}^{N-1}})W_{N-1}(X_{N-2})^{-1}W_{N-1}^T(D_{\boldsymbol{\alpha}^{N-1}} + D_{\boldsymbol{\beta}^{N-1}}). \tag{34}$$

Note that the optimization problem (34), together with the condition $X_{N-2} > 0$, is equivalent to (13) with $i = N-1$. Moving backwards, at stage $i-1$, the goal is to select $\Lambda_{i-1}$ so as to maximize the feasible region of $X_i > 0$ in the subsequent step. We observe that

$$X_i = \Lambda_i - \frac{1}{4}\Lambda_i(D_{\boldsymbol{\alpha}^i} + D_{\boldsymbol{\beta}^i})W_i(X_{i-1})^{-1}W_i^T(D_{\boldsymbol{\alpha}^i} + D_{\boldsymbol{\beta}^i})\Lambda_i + W_{i+1}^T D_{\boldsymbol{\alpha}^{i+1}}\Lambda_{i+1}D_{\boldsymbol{\beta}^{i+1}}W_{i+1}$$
$$X_{i-1} = M_{i-1} + W_i^T D_{\boldsymbol{\alpha}^i}\Lambda_i D_{\boldsymbol{\beta}^i}W_i, \qquad i \in \mathbb{Z}_{N-2}. \tag{35}$$

While $\boldsymbol{\alpha}^i$, $\boldsymbol{\beta}^i$, and $\Lambda_i$ are not yet decided in (35), we expect that minimizing the scale of $W_i(X_{i-1})^{-1}W_i^T$ in the sense of its spectrum will yield a larger feasible region for $\Lambda_i$ in the next stage. Similarly from (12), the term containing $\boldsymbol{\alpha}^i$, $\boldsymbol{\beta}^i$, and $\Lambda_i$ is not decided at the stage. However, we can still strategically minimize the scale of $W_i(M_{i-1})^{-1}W_i^T$ to enlarge the feasible region for $\Lambda_i$ in the next stage, which is directly aligned with our goal of minimizing $\sigma_{max}\left(W_N^T W_N(X_{N-1})^{-1}\right)$ at the last stage since $X_{N-1} = M_{N-1}$ as in (12).

Therefore, we solve the following optimization problem to derive $\Lambda_{i-1}$:

$$\max_{c_{i-1},\Lambda_{i-1}} c_{i-1}$$
$$\text{s.t.} \quad M_{i-1} \geq c_{i-1}W_i^T W_i, \tag{36}$$
$$M_{i-1} = \Lambda_{i-1} - \frac{1}{4}\Lambda_{i-1}(D_{\boldsymbol{\alpha}^{i-1}} + D_{\boldsymbol{\beta}^{i-1}})W_{i-1}(X_{i-2})^{-1}W_{i-1}^T(D_{\boldsymbol{\alpha}^{i-1}} + D_{\boldsymbol{\beta}^{i-1}})\Lambda_{i-1}$$

Together with the condition $X_{i-2} > 0$, applying the Schur complement shows that this is equivalent to (13).

Furthermore, in computing $\mathbf{L}^{(0,i)}_{\bullet,l}$, the procedure is identical to that for computing the Lipschitz constant of the mapping $f^{(i)} : z^{(0)} \mapsto v^{(i)}$, except that the weight matrix $W_i$ is trimmed to retain only its $l$-th row, denoted $(W_i)_{(l,\bullet)}$. Concretely, when we obtain $\mathbf{L}^{(0,i)}_{\bullet,l}$ for the $l^{th}$ neuron at stage $i$, $i \in \mathbb{Z}$, by Proposition 4, the $l^{th}$ diagonal entry of $W_i(M_{i-1})^{-1}W_i^T$ provides exactly the associated maximal eigenvalue as follows:

$$\left(W_i(M_{i-1})^{-1}W_i^T\right)_{(l,l)} = \sigma_{\max}\left((W_i)^T_{(\bullet,l)}(W_i)_{(\bullet,l)}(M_{i-1})^{-1}\right) = \sigma_{\max}\left((W_i)_{(\bullet,l)}(M_{i-1})^{-1}(W_i)^T_{(\bullet,l)}\right).$$

Therefore, our goal of minimizing the scale of $W_i(M_{i-1})^{-1}W_i^T$, is consistently applied both at the network output and at each neuron for all $i \in \mathbb{Z}_{N-1}$.

## 4 Experiments

We conduct three sets of experiments to systematically evaluate our methods. The first set considers randomly generated neural networks of both small and large sizes. We compare our methods to an extensive set of benchmarks to illustrate the scalability, efficiency and tightness of our algorithms. In the second set, we vary the size of the input region and demonstrate how our algorithm leverages local information to achieve very tight Lipschitz estimates. The final set compares the local Lipschitz estimates on two networks, one trained conventionally and the other trained with robustness objectives, highlighting the practical utility of our approach. The details of the experimental setup, and generation of the neural networks (both randomly generated and trained on the MNIST dataset), and complete experiment data are described in Appendix A.3.

**Benchmarks.** We evaluate against methods that share the same SDP framework: ECLipsE-Gen-Local (our method), ECLipsE (Xu & Sivaranjani (2024)), LipSDP (Fazlyab et al. (2019)), GLipSDP (Pauli et al. (2024)). For ECLipsE-Gen-Local-Acc, $\Lambda_i$, $i \in \mathbb{Z}_{N-1}$ can have different diagonal entries, which directly benchmarks to ECLipsE, GLipSDP, and LipSDP-Neuron. For ECLipsE-Gen-Local-Fast and ECLipsE-Gen-CF, $\Lambda_i = \lambda_i I$, $i \in \mathbb{Z}_{N-1}$, which benchmarks to LipSDP-Layer and ECLispE-Fast. Additionally, we compare our Lipschitz estimates to the naive upper bound $L_{naive} = \prod_{i=1}^{l} \|W_i\|_2$ (Szegedy et al. (2013)), SeqLip(Virmaux & Scaman (2018)), GeoLip (Wang et al. (2022)), AAO (Combettes & Pesquet (2020)), and LipDiff (Wang et al. (2024)). All Lipschitz constants are computed with respect to the $\ell_2$–induced operator norm, making the comparisons across benchmarks directly comparable.

While we consider three variants that choose among the `Acc`, `Fast`, and `CF` homogeneously for all layers, we note that our framework also offers the flexibility of combining these options on a per-layer basis. For brevity of exposition, we abbreviate our ECLipsE-Gen-Local series of algorithms as: `Acc` (ECLipsE-Gen-Local-Acc), `Fast` (ECLipsE-Gen-Local-Fast), and `CF` (ECLipsE-Gen-Local-CF).

### 4.1 Scalability, Efficiency, and Tightness on Randomly Generated Networks

We implement algorithms to estimate the local Lipschitz constant whenever applicable; otherwise, we fall back to the global estimate given by the algorithm. All the generated neural networks generated have input size $d_0 = 5$ and output size $d_N = 2$. The local region is picked as $\mathcal{Z} = \mathcal{B}(z_c, \delta_z)$ with $z_c = [0.4, 1.8, -0.5, -1.3, 0.9]^T$ and $\delta_z = 1$.

**Case 1: Small Neural Networks**.

*Setup.* We conduct a total of 20 experiments for all the 13 algorithms on randomly generated FNNs, corresponding to all combinations of the number of layers in {5,10,15,20,25} and the number of neurons in {10,20,40,60}. As the benchmark SeqLip only applies to ReLU activation function, the FNNs are all generated with ReLU. To systematically evaluate the scalability, efficiency and tightness of different algorithms, we present the normalized Lipschitz estimates with respect to the naive upper bound and the computation time in seconds. While the complete results are provided in Appendix A.3.4, for clarity of presentation, we focus on two representative cases: (i) fixing the number of layers to be 20 while varying the number of neurons; (ii) fixing the number of neurons to be 40 while varying the number of layers. We set a cutoff time of 10 minutes for all experiments.

*Effect of depth - tightness.* From Fig. 1a, we first examine the tightness. LipDiff yields the loosest bounds, while AAO provides better estimates but is still outperformed by all other methods. GeoLip achieves accuracy comparable to `Fast`. Within the SDP-based methods for the special case $\Lambda_i = \lambda_i I \geq 0$, `CF` produces slightly tighter results than ECLipsE-Fast, while `Fast` achieves a level of tightness comparable to LipSDP-layer and, notably, even approaches the tightness of ECLipsE, which allows larger flexibility in $\Lambda_i$. This improvement stems from efficiently leveraging local information. At the top end, SDP-based methods with fully flexible $\Lambda_i \geq 0$ deliver the tightest estimates: LipSDP, GLipSDP, and `Acc` consistently outperform other benchmarks,

with ECLipsE being somewhat looser. In certain cases (e.g., 20 layers), `Acc` demonstrates outstanding performance, indicating the success in capturing local information.

*Effect of depth - computation time.* Turning to the computation time in Fig. 1b. SeqLip fails to provide results even for networks with as few as 5 layers, while AAO breaks down at 20 layers. Among the methods that succeed for this case, GeoLip and LipSDP-neuron are the most time-consuming, although they demonstrate good accuracy. Within the SDP-based family, LipSDP-neuron and LipSDP-layer both incur rapidly growing computational cost with depth. In contrast, GLipSDP, `Acc`, ECLipsE, and `Fast` (in decreasing order for running time) exhibit linearly increasing computational cost with depth, demonstrating clear scalability. At the most efficient and scalable extreme, ECLipsE-Fast and `CF` have *near-instantaneous* running time thanks to closed-form solutions at each stage.

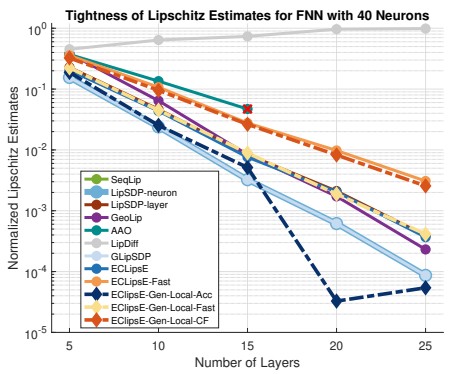

(a) Lipschitz estimates normalized to trivial bound

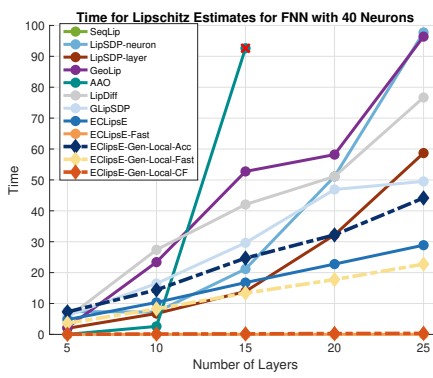

(b) Computation time (seconds)

Figure 1: Performance for increasing network depth, with 40 neurons. The red x markings indicate that the algorithm fails to provide an estimate within the computational cutoff time beyond this network size.

*Effect of width.* From Fig. 2, most trends mirror the above case where we varies network depth. Here, LipDiff remains the loosest, and GeoLip again matches the accuracy of `Fast`. Within the restricted SDP family, `CF` is tighter than ECLipsE-Fast, while `Fast` nearly matches LipSDP-layer and approaches ECLipsE. The highest tightness is still attained by LipSDP, GLipSDP, and `Acc`. In terms of computation time, AAO fails immediately and SeqLip breaks down at width 10. GeoLip, GLipSDP, and LipSDP-neuron are most affected by increasing width, with computational costs rising much faster than for our proposed methods. LipDiff is less sensitive to width than the other benchmarks; however, the computation time still grows faster than our proposed algorithms, while yielding looser estimates. LipSDP-layer also shows noticeable growth in computation time with network depth but remains acceptable. By comparison, `Acc`, ECLipsE, and `Fast` exhibit slightly faster than linear computation times, yet scale much more favorably than the benchmarks. At the most efficient extreme, ECLipsE-Fast and `CF` continue to have negligible runtime.

Taken together, these results highlight distinct groups of algorithms. LipSDP-neuron, LipSDP-layer, and GeoLip provide reasonably good accuracy but are not scalable, with costs growing rapidly as networks enlarge. LipDiff scales better but yields overly loose estimates, limiting its practical value. GLipSDP shows scalability with respect to depth but becomes increasingly costly as width increases. By contrast, the ECLipsE family demonstrates a clear trend of maintaining scalability while preserving competitive accuracy. `Acc` takes slightly more time than ECLipsE but remains equally scalable and produces bounds at the same or better level than LipSDP-neuron. `Fast` runs faster while matching the accuracy of LipSDP-layer and ECLipsE. Finally, the closed-form variants, ECLipsE-Fast and `CF`, incur *negligible* runtime, with the latter yielding tighter estimates. While the advantages are only partially revealed in this small-scale setting, the trends point toward the much clearer separation we will observe in the large-network experiments discussed next.

**Case 2: Large Neural Networks.** *Setup.* As we observed in Case 1, SeqLip and AAO fail at small network sizes and are therefore excluded from further experiments. To examine scalability with larger networks, we consider FNNs with the number of layers in $\{30, 40, 50, 60, 70\}$ and the number of neurons in $\{60, 80, 100, 120\}$. For this setting, we generate networks with the ELU activation to demonstrate the consistently superior

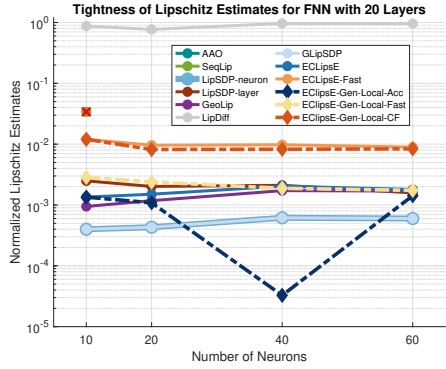

(a) Lipschitz estimates normalized to trivial bound

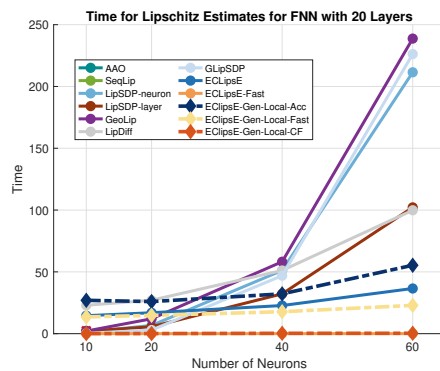

(b) Computation time (seconds)

Figure 2: Performance for increasing network width, with 20 layers. The red x markings indicate that the algorithm fails to provide an estimate within the computational cutoff time beyond this network size.

performance of our algorithms even with nonlinear activation functions. The cutoff time for this set of experiments is set to 60 minutes.

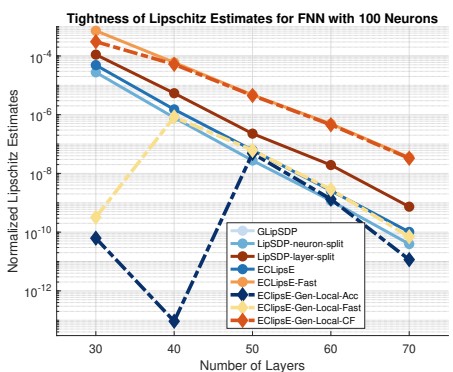

(a) Lipschitz estimates normalized to trivial bound

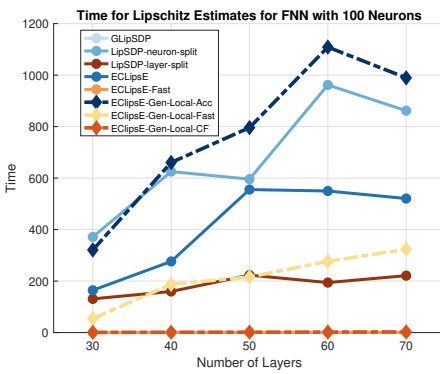

(b) Computation time (seconds)

Figure 3: Performance for increasing network depth, with 100 neurons.

Although LipSDP-neuron and LipSDP-layer have exponential running times as the number of layers increases, they adopt a splitting strategy Fazlyab et al. (2019) to mitigate the scalability issue, wherein they split the network into sub-networks, and multiply the Lipschitz constants of the sub-networks to obtain the final estimate. In our benchmarks, we consider the split versions of both algorithms, termed LipSDP-neuron-split and LipSDP-layer-split respectively. For LipSDP-neuron-split and LipSDP-layer-split, the FNNs are split into sub-networks of 10 layers each and three workers are used for parallel computation to accelerate the process.

We choose the split variants of LipSDP as a baseline primarily because the standard LipSDP variants exceed the cutoff time for Lipschitz estimation without splitting and parallelization. Note that while our method can be similarly parallelized by splitting the network into sub-networks, we apply ECLipsE-Gen-Local to the full network as it is scalable and computationally efficient by design, while prioritizing tightness. This choice allows us to retain coupling information across the full network where standard LipSDP fails.

*Results.* Among the benchmarks, we report that GeoLip fails at the smallest configuration (30 layers, 60 neurons) due to kernel crashes, while LipDiff consistently produces invalid estimates larger than the naive upper bound. Therefore, for the remainder of this discussion, we focus on the remaining algorithms, reporting results for two cases: (i) fixing the number of layers to 60 while varying the number of neurons, and (ii) fixing the number of neurons to 100 while varying the number of layers.

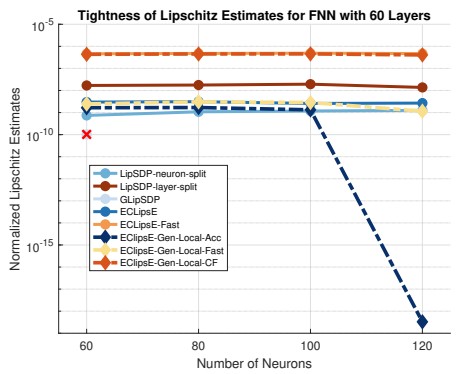

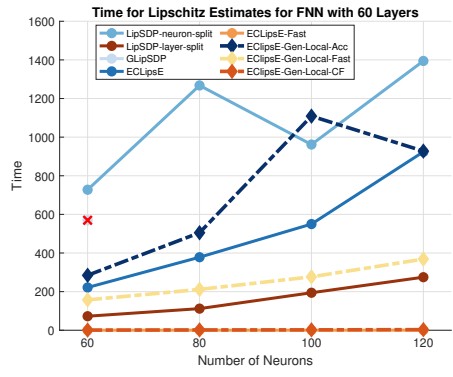

(a) Lipschitz estimates normalized to trivial bound

(b) Computation time (seconds)

Figure 4: Performance for increasing network width, with 60 layers. The red x markings indicate that the algorithm fails to provide an estimate within the computational cutoff time beyond this network size.

*Tightness.* Across both cases, namely, varying the network depth with 100 neurons and varying the network width with 60 layers, we observe that the same insights emerge according to Figs. 3 and 4. First, we observe that while GLipSDP scales with depth, it fails beyond the 60-neuron cases for the wide networks considered here. In terms of tightness, in Fig. 3a, 4a, ECLipsE-Fast and `CF` yield tighter estimates than LipSDP-layer-split, with `CF` slightly tighter than ECLipsE-Fast, but comparatively looser than `Fast`. ECLipsE, LipSDP-neuron-split, `Fast`, and `Acc` form a close cluster in accuracy, with `Acc` consistently tighter than `Fast`. Notably, `Fast` is almost as tight as ECLipsE, despite the relaxation with $\Lambda_i = \lambda_i I$. When the output landscape is locally flat over the input region, the advantage of our methods capturing local information becomes particularly pronounced. For example, in Fig. 3a, with 40 layers, `Acc` is more than $10^7$ *times tighter* than LipSDP-neuron-split; in Fig. 4a), with 120 neurons, `Acc` is over $10^9$ *tighter* than LipSDP-neuron-split.

*Computation time.* Even with splitting and parallelism that requires more computional resources (three cores versus one for our methods), LipSDP-neuron-split is only slightly faster in Fig. 3b and generally slower than `Acc` in Fig. 4b at similar tightness, while LipSDP-layer-split is essentially on par with `Fast`. However, we emphasize that our methods achieve this performance *without* relying on parallel computation. Once again, ECLipsE-Fast and `CF` remain negligible in time cost owing to closed form solutions, with `CF` uniformly tighter than ECLipsE-Fast.

In conclusion, `Acc` offers the tightest bounds with strong scalability, `Fast` matches the accuracy of LipSDP-layer-split and ECLipsE at lower cost, and `CF` is near-instantaneous while being tighter than ECLipsE-Fast. These properties underscore the practical advantage of the ECLipsE-Gen-Local family in estimating local Lipschitz constants for large networks.

## 4.2 Tightness of Local Estimates: Achieving Provable Upper Bounds at `autodiff` Level

We have demonstrated scalability, efficiency, and tightness of our algorithms in the previous section. Here, we study the tightness of the *certified local bounds* as the local region shrinks. We consider $\mathcal{Z} = \mathcal{B}(z_c, \delta_z)$ centered at $z_c = [0.4, 1.8, -0.5, -1.3, 0.9]^T$ with radius chosen from $\delta_z \in \{5, 1/5, 1/5^2, 1/5^3, 1/5^4, 1/5^5\}$. We evaluate three FNNs of 5, 30, and 60 layers (128 neurons each) with LeakyReLU ($\alpha = 0.01$). While the insights are common across all three cases, we present the 30-layer case here and defer the others to Appendix A.3. For reference, the trivial bound (valid for the entire region $\mathcal{Z}$) is $3.070 \times 10^{10}$, contrasting the scale of the certified upper bounds for the chosen regions.

From Fig. 5, we observe that as the radius of the input region decreases, the estimates from all three variants `Acc`, `Fast`, and `CF` tighten monotonically by many orders of magnitude. Among the three variants, `CF`, though generally looser, has negligible runtime and continues to improve as the radius shrinks, capturing the local behavior of FNN at small $\delta_z$. `Acc` is uniformly the tightest and `Fast` closely tracks `Acc`. The local Lipschitz estimates from `Acc`, and `Fast` drop sharply at input radius $1/25$ for `Acc` and $1/125$ for `Fast`, and approach

the `autodiff` level, that is, the gradient norm at the center $z_c$, when the radius $\delta_z$ is small enough. Notably, the gradient norm at $z_c$ generated by `autodiff` is a strict lower bound on the Lipschitz constant, making our algorithms essentially optimal in terms of tightness. Importantly, unlike `autodiff` that provides gradient norm at the center of the input region, our estimates are provable upper bounds serving as certificates for the entire local region.

***Remark*** 5. (Local vs. Global) The practical value of local estimates depends on the scale of the input region. For large regions or when model behavior is expected to be homogeneous over the full domain, the local bound is close to the global constant. In such cases, scalable global estimation algorithms like ECLipsE (if applicable) are preferable due to lower computational cost. However, when the region is small to moderate, such as in robustness certification around a specific input (as will be demonstrated in Sec. 4.3), the proposed local method exploits the local landscape to yield significantly tighter bounds, justifying the additional computation cost.

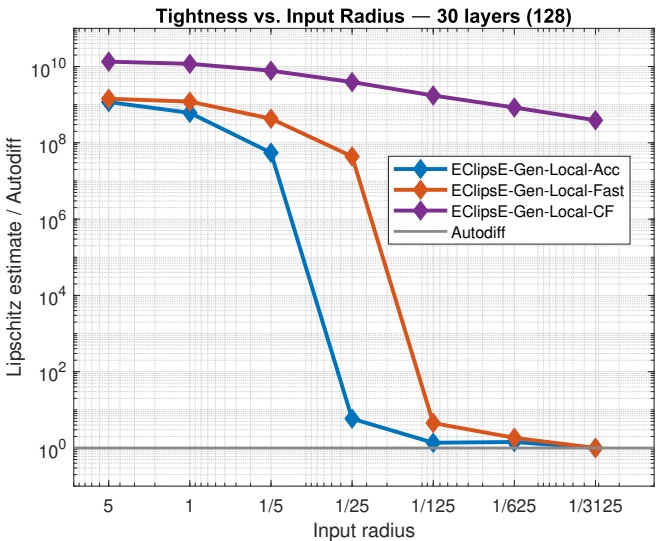

Figure 5: Lipschitz estimates normalized to `autodiff` value at $z_c$ (0.0015). Naive bound: $3.0698 \times 10^{10}$.

## 4.3 Lipschitz Estimates on Standard vs. Robustly Trained Networks

*Setup.* The final set of experiments estimate the local Lipschitz constant for various sizes of the input region on two networks, one trained conventionally and the other trained with robustness objectives, highlighting the practical utility of tight estimates from our methods. We use the MNIST dataset and train two FNNs with identical architectures: three hidden layers of 128 units with ELU activations. The baseline network is trained with standard cross-entropy loss, while robustly trained network employs Jacobian regularization (*JacobianReg*) (Hoffman et al. (2019)), which penalizes the norm of the derivatives of the network's outputs with respect to its inputs in order to encourage smoother mappings and improve robustness. Both FNNs achieve an accuracy of at least 98% on the test set. We assess robustness using a standard $\ell_2$ projected gradient descent (PGD) attack on the test set (Madry et al. (2017)): for each test point $x$, we search for misclassifications under attack within the $\ell_2$ ball $\{x' : \|x' - x\|_2 \leq \epsilon\}$. Details on the training and testing of both networks are included in the Appendix A.3.

To establish the relationship between robustness and Lipschitz estimates, we first empirically quantify the robustness of the two networks by recording the failure rate of both networks under an $\ell_2$ PGD attack with radius $\epsilon$ chosen from $\{1/2, 1/4, 1/8, 1/16, 1/32, 1/64, 1/128, 1/256\}$. This means that adversarial perturbations are chosen from an $\ell_2$ ball of size $\epsilon$ around each test point. Independently, we randomly sample 20 data points from the valid input region of the MNIST dataset and compute certified local Lipschitz constants at 20 points on the same $\epsilon$-balls using `Fast`. This experiment design provides a statistically meaningful comparison between robustness to adversarial perturbations and Lipschitz estimates at matched scales. `Fast` is chosen

for its balance of accuracy and efficiency, as it is computationally cheaper than `Acc` and captures local region information more effectively than `CF`.

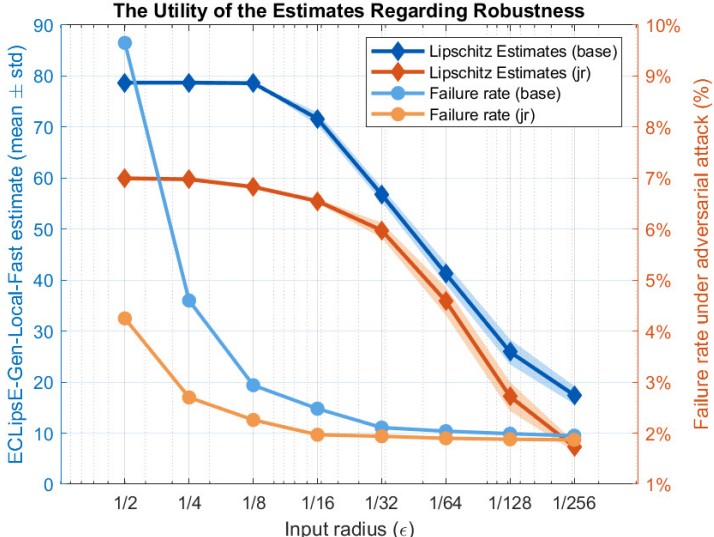

Figure 6: Lipschitz estimates for different size of input region on baseline model and robustly trained model. The blue and orange solid lines plot the mean of the Lipschitz estimates for the baseline and robustly trained models, with the shaded interval representing the standard deviation of the Lipschitz estimates across the sampled points. On the same axes we report the failure rate of each network under adversarial perturbations of radius $\epsilon$, shown as lighter curves on the secondary vertical axis.

*Results.* Figure 6 shows how our certified Lipschitz estimates relate to robustness, which is quantified as the empirical failure rate under adversarial attacks. For each $\epsilon$, we compare the local Lipschitz estimates on the input region of radius $\epsilon$, and the failure rate of each network under adversarial perturbations of radius $\epsilon$. It is clear that the robustly trained model consistently exhibits smaller Lipschitz estimates together with lower failure rates for every $\epsilon$. Meanwhile, the standard deviation of the estimates shows an increasing trend as $\epsilon$ decreases, indicating that our method manages to capture the diversity of local landscapes around different points. These alignments between the certified Lipschitz estimates and observed robustness illustrates the practical utility of our method in capturing robustness through provable and tight Lipschitz upper bounds.

*Robustness certificates.* To further demonstrate the utility of tight and strict Lipschitz estimates, we report robustness certificates on the regularized MNIST model using the standard Lipschitz-margin certificate for multi-class logits (Tsuzuku et al. (2018)). This Lipschitz-based robustness certificate provides a lower bound on the $\ell_2$ perturbation magnitude required to eliminate the logit margin between the predicted class and its closest competitor, thereby guaranteeing label invariance within the certified radius. We sweep the same radii $\epsilon = 1/2, 1/4, \ldots, 1/256$ and use the same test points as in our local Lipschitz evaluation. For each test point $x$, we compute a certified local Lipschitz bound $L(x, \epsilon)$ that is valid within the analyzed radius $\epsilon$, compute the logit margin $m(x) = f_{\hat{y}}(x) - \max_{j \neq \hat{y}} f_j(x)$, and form the candidate radius $r_{est}(x, \epsilon) = m(x)/(\sqrt{2}L(x, \epsilon))$. Since a Lipschitz bound certified on radius $\epsilon$ is also valid for any smaller radius, we use the valid certificate $r_{cert}(x, \epsilon) = \min(r_{est}(x, \epsilon), \epsilon)$, which guarantees that any perturbation with $\|\delta\|_2 < r_{cert}(x, \epsilon)$ cannot change the predicted label. We then report the least conservative certified radius across the swept scales via $r(x) = \max_{\epsilon \in \{1/2, 1/4, \ldots, 1/256\}} r_{cert}(x, \epsilon)$. Detailed results, along with comparisons against the certified radius obtained from the global trivial bound $L_{triv} = 236.4329$, are reported in Appendix A.3. The mean certified radius produced by our local Lipschitz estimates is 6.14 times larger than the mean certified radius obtained from $L_{triv}$, indicating a substantial reduction in conservatism.

## 5 Conclusion

In this work, we introduced ECLipsE-Gen-Local, a compositional framework that provides certified upper bounds for the Lipschitz constants of deep feedforward networks. By adapting SDP-based Lipschitz certificates to accommodate heterogeneous slope bounds for the activation functions, systematically incorporating local information on the input-region, and decomposing the large-scale SDP for Lipschitz estimation into sequential sub-problems, our algorithms provide provably valid and tight estimates with linear complexity in depth. Notably, we propose a variant that provides closed-form solutions at each sequential sub-problem, achieving near-instantaneous computation while retaining certification guarantees. Through extensive experiments, we showed that our methods deliver outstanding scalability and produce substantially tighter bounds than global approaches, with local estimates approaching the the exact Lipschitz constant in small regions. Finally, since our certificate relies only on affine transformations and sector-bounded activations, it is natural to extend this framework to other architectures such as Convolutional Neural Networks and Residual Networks. Future work will focus on extending the framework to other architectures, and on integrating local Lipschitz certificates into robust training for safety-critical tasks.

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

# A  Appendix

## A.1  Proofs

**Proof of Theorem 2**. Let $z_1^{(i)}$, $z_2^{(i)}$ be the outputs of layer $L_i$ for two arbitrary inputs, and define $\Delta z^{(i)} = z_1^{(i)} - z_2^{(i)}$, $\Delta v^{(i)} = v_1^{(i)} - v_2^{(i)}$ for all $i \in \{0\} \cup \mathbb{Z}_N$. Consider the stacked vector $\Delta \mathbf{z} = [\Delta z^{(0)^T}, \ldots, \Delta z^{(N-1)^T}]^T$. Left and right multiplying the matrix in (4) by $\Delta \mathbf{z}^T$ and $\Delta \mathbf{z}$ and utilizing the fact that $\Delta z^{(N)} = W_N \Delta z^{(N-1)}$, we obtain

$$(\Delta z^{(0)})^T \Delta z^{(0)} + \sum_{i=1}^{N-1} \begin{bmatrix} \Delta z^{(i-1)} \\ \Delta z^{(i)} \end{bmatrix}^T Y_i \begin{bmatrix} \Delta z^{(i-1)} \\ \Delta z^{(i)} \end{bmatrix} - F(\Delta z^{(N)})^T (\Delta z^{(N)}) > 0, \qquad (37)$$

where $Y_i \triangleq \begin{bmatrix} W_i^T D_{\boldsymbol{\alpha}^i} \Lambda_i D_{\boldsymbol{\beta}^i} W_i & -\frac{1}{2} W_i^T (D_{\boldsymbol{\alpha}^i} + D_{\boldsymbol{\beta}^i}) \Lambda_i \\ -\frac{1}{2} \Lambda_i (D_{\boldsymbol{\alpha}^i} + D_{\boldsymbol{\beta}^i}) W_i & \Lambda_i \end{bmatrix}$.

By the slope-restrictedness condition in Assumption 1, for $i \in \mathbb{Z}_{N-1}$, we have $\boldsymbol{\alpha}^i \Delta v^{(i)} \leq \Delta z^{(i)} \leq \boldsymbol{\beta}^i \Delta v^{(i)}$.

Equivalently, for any $\Lambda_i \in \mathbb{D}_+$,

$$\begin{bmatrix} \Delta v^{(i)} \\ \Delta z^{(i)} \end{bmatrix}^T \begin{bmatrix} D_{\boldsymbol{\alpha}^i} \Lambda_i D_{\boldsymbol{\beta}^i} & -\frac{1}{2}(D_{\boldsymbol{\alpha}^i} + D_{\boldsymbol{\beta}^i})\Lambda_i \\ -\frac{1}{2}\Lambda_i(D_{\boldsymbol{\alpha}^i} + D_{\boldsymbol{\beta}^i}) & \Lambda_i \end{bmatrix} \begin{bmatrix} \Delta v^{(i)} \\ \Delta z^{(i)} \end{bmatrix} \leq 0.$$

Since $\Delta v^{(i)} = W_i \Delta z^{(i-1)}$, this yields

$$\begin{bmatrix} \Delta z^{(i-1)} \\ \Delta z^{(i)} \end{bmatrix}^T Y_i \begin{bmatrix} \Delta z^{(i-1)} \\ \Delta z^{(i)} \end{bmatrix} \leq 0.$$

Thus, every summand in (37) involving $Y_i$ is non-positive. Therefore,

$$(\Delta z^{(0)})^T \Delta z^{(0)} - F(\Delta z^{(N)})^T (\Delta z^{(N)}) \geq 0,$$

which further gives

$$\|\Delta z^{(N)}\|_2 \leq \sqrt{1/F}\|\Delta z^{(0)}\|_2,$$

implying that the Lipschitz constant is at most $\sqrt{1/F}$. $\qquad\square$

**Proof of Theorem 3**. The proof is similar to that of Theorem 2. Let $z_1^{(j)}$, $z_2^{(j)}$ be the outputs at layer $L_j$ for two arbitrary inputs, and define $\Delta z^{(j)} = z_1^{(j)} - z_2^{(j)}$ for $j = p, \ldots, i$. Let $\Delta z_{\mathcal{K}}^{(p)}$ and $\Delta z_{\mathcal{L}}^{(i)}$ denote the subvectors of $\Delta z^{(p)}$ and $\Delta z^{(i)}$ indexed by $\mathcal{K} \subseteq \mathbb{Z}_{d_p}$ and $\mathcal{L} \subseteq \mathbb{Z}_{d_i}$, respectively.

Consider the stacked vector

$$\Delta \mathbf{z} = \left[ (\Delta z_{\mathcal{K}}^{(p)})^T, (\Delta z^{(p+1)})^T, \ldots, (\Delta z^{(i-1)})^T \right]^T.$$

Left and right multiplying the matrix in (7) by $\Delta \mathbf{z}$ yields and utilizing the fact that $\Delta z_{\mathcal{L}}^{(i)} = (W_i)_{(\mathcal{L},\bullet)} \Delta z^{(i-1)}$:

$$(\Delta z_{\mathcal{K}}^{(p)})^T (\Delta z_{\mathcal{K}}^{(p)}) + \sum_{m=p+1}^{i-1} \begin{bmatrix} \Delta z^{(m-1)} \\ \Delta z^{(m)} \end{bmatrix}^T Y_m \begin{bmatrix} \Delta z^{(m-1)} \\ \Delta z^{(m)} \end{bmatrix} - F(\Delta z_{\mathcal{L}}^{(i-1)})^T((\Delta z_{\mathcal{L}}^{(i-1)}) > 0, \qquad (38)$$

where

$$Y_m \triangleq \begin{bmatrix} W_m^T D_{\boldsymbol{\alpha}^m} \Lambda_m D_{\boldsymbol{\beta}^m} W_m & -\frac{1}{2}W_m^T(D_{\boldsymbol{\alpha}^m} + D_{\boldsymbol{\beta}^m})\Lambda_m \\ -\frac{1}{2}\Lambda_m(D_{\boldsymbol{\alpha}^m} + D_{\boldsymbol{\beta}^m})W_m & \Lambda_m \end{bmatrix}.$$

Similarly, by the slope-restrictedness condition in Assumption 1, for each $m$, and for any $\Lambda_m \in \mathbb{D}_+$,

$$\begin{bmatrix} \Delta v^{(m)} \\ \Delta z^{(m)} \end{bmatrix}^T \begin{bmatrix} D_{\boldsymbol{\alpha}^m} \Lambda_m D_{\boldsymbol{\beta}^m} & -\frac{1}{2}(D_{\boldsymbol{\alpha}^m} + D_{\boldsymbol{\beta}^m})\Lambda_m \\ -\frac{1}{2}\Lambda_m(D_{\boldsymbol{\alpha}^m} + D_{\boldsymbol{\beta}^m}) & \Lambda_m \end{bmatrix} \begin{bmatrix} \Delta v^{(m)} \\ \Delta z^{(m)} \end{bmatrix} \leq 0.$$

Since $\Delta v^{(m)} = W_m \Delta z^{(m-1)}$, this yields

$$\begin{bmatrix} \Delta z^{(m-1)} \\ \Delta z^{(m)} \end{bmatrix}^T Y_m \begin{bmatrix} \Delta z^{(m-1)} \\ \Delta z^{(m)} \end{bmatrix} \leq 0.$$

Therefore, all terms involving $Y_m$ in (38) are non-positive.

Hence,

$$(\Delta z_{\mathcal{K}}^{(p)})^T (\Lambda_p)_{(\mathcal{K},\mathcal{K})}(\Delta z_{\mathcal{K}}^{(p)}) - F(\Delta z_{\mathcal{L}}^{(i)})^T (W_i)_{(\mathcal{L},\bullet)}^T (W_i)_{(\mathcal{L},\bullet)} (\Delta z_{\mathcal{L}}^{(i)}) \geq 0.$$

Since $(\Delta z_{\mathcal{L}}^{(i)}) = (W_i)_{(\mathcal{L},\bullet)} \Delta z^{(i-1)}$, this further gives

$$\|\Delta z_{\mathcal{L}}^{(i)}\|_2 \leq \sqrt{1/F}\|\Delta z_{\mathcal{K}}^{(p)}\|_2,$$

establishing that inequality (6) holds. $\qquad\square$

**Proof of Theorem 4.** The result follows by applying Lemma 2 in Agarwal et al. (2019) to the symmetric block tridiagonal matrix in (4). By Lemma 2 in Agarwal et al. (2019), a symmetric block tri-diagonal matrix defined as

$$
\begin{bmatrix}
\widetilde{\mathcal{P}}_1 & \widetilde{\mathcal{R}}_2 & 0 & \cdots & & 0 \\
\widetilde{\mathcal{R}}_2^T & \widetilde{\mathcal{P}}_2 & \widetilde{\mathcal{R}}_3 & \cdots & & 0 \\
0 & \widetilde{\mathcal{R}}_3^T & \widetilde{\mathcal{P}}_3 & \cdots & & 0 \\
& & & \vdots & & \\
0 & \cdots & 0 & \widetilde{\mathcal{R}}_{N-1}^T & \widetilde{\mathcal{P}}_{N-1} & \widetilde{\mathcal{R}}_N \\
0 & \cdots & 0 & 0 & \widetilde{\mathcal{R}}_N^T & \widetilde{\mathcal{P}}_N
\end{bmatrix}
$$

is positive definite if and only if

$$
\widetilde{X}_i > 0, \quad \forall i \in \{0\} \cup \mathbb{Z}_{l-1},
$$

where

$$
\widetilde{X}_i = \begin{cases}
\widetilde{\mathcal{P}}_i & \text{if } i = 0, \\
\widetilde{\mathcal{P}}_i - \widetilde{\mathcal{R}}_i^T \widetilde{X}_{i-1}^{-1} \widetilde{\mathcal{R}}_i & \text{if } i \in \mathbb{Z}_{l-1}.
\end{cases}
$$

For our matrix, we directly substitute $\widetilde{\mathcal{P}}_i, \widetilde{\mathcal{R}}_i$ with $\mathcal{P}_i, \mathcal{R}_i$ as defined in (5) and we have the result that the block tridiagonal matrix in (4) is positive definite if and only if the sequence of inequalities in (9) holds, with $X_i$ defined in (10). $\qquad\square$

**Proof of Proposition 1.** Applying the Schur complement to the LMI in (14) with slope bounds $\boldsymbol{\alpha}^{i,\mathrm{adj}}$ and $\boldsymbol{\beta}^{i,\mathrm{adj}}$ directly gives an equivalent condition to be $M_i - c_i W_{i+1}^T W_{i+1} \geq 0$, where

$$
M_i = \Lambda_i - \frac{1}{4} \Lambda_i \left( D_{\boldsymbol{\alpha}^{i,\mathrm{adj}}} + D_{\boldsymbol{\beta}^{i,\mathrm{adj}}} \right) W_i \left( X_{i-1} \right)^{-1} W_i^T \left( D_{\boldsymbol{\alpha}^{i,\mathrm{adj}}} + D_{\boldsymbol{\beta}^{i,\mathrm{adj}}} \right) \Lambda_i.
$$

This holds if and only if $c_i \leq \frac{1}{\sigma_{\max}\left(W_{i+1}^T W_{i+1} M_i^{-1}\right)}$. According to Lemma 1, $\sigma_{\max}\left(W_{i+1}^T W_{i+1} M_i^{-1}\right) = \sigma_{\max}\left(W_{i+1} M_i^{-1} W_{i+1}^T\right)$, completing the proof. $\qquad\square$

**Proof of Proposition 2.** By Theorem 4, with $X_i > 0$, $i \in \mathbb{Z}_{N-2}$, it remains to prove that $X_{N-1} - F W_N^T W_N > 0$. This is equivalent to $X_{l-1}/F > W_N^T W_N$. Then, the smallest possible $1/F$ is $\sigma_{max}(W_N^T W_N (M_{N-1})^{-1})$. By Theorem 1, the tightest upper bound for the Lipschitz constant is then $\sqrt{1/F} = \sqrt{\sigma_{max}(W_N^T W_N (M_{N-1})^{-1})}$. Further, from Lemma 1 and the fact that $M_{N-1} > 0$, we have the certified Lipschitz constant to be $\sqrt{\sigma_{max}(W_N (M_{N-1})^{-1}) W_N^T}$. $\qquad\square$

**Proof of Theorem 5.** Let $\{\hat{\boldsymbol{\alpha}}^i, \hat{\boldsymbol{\beta}}^i\}_{i=1}^{N-1}$ and $\{\tilde{\boldsymbol{\alpha}}^i, \tilde{\boldsymbol{\beta}}^i\}_{i=1}^{N-1}$ be two sets of slope bounds for the activation functions, where for some $j, i$, $[\tilde{\alpha}_j^i, \tilde{\beta}_j^i] \subset [\hat{\alpha}_j^i, \hat{\beta}_j^i]$, and all other entries are identical. Let $F$ and $\tilde{F}$ denote the maximal values such that the matrix inequalities in (4) and (5) are satisfied for the respective choices of slope bounds. Adopting the notations in the proof of Theorem 2, we left and right multiply the matrix in (4) by $\Delta\mathbf{z}^T$ and $\Delta\mathbf{z}$, where $\Delta\mathbf{z} = [\Delta z^{(0)^T}, \dots, \Delta z^{(N-1)^T}]^T$, and obtain

$$
(\Delta z^{(0)})^T \Delta z^{(0)} + \sum_{i=1}^{N-1} \begin{bmatrix} \Delta v^{(i)} \\ \Delta z^{(i)} \end{bmatrix}^T \begin{bmatrix} D_{\boldsymbol{\alpha}^i} \Lambda_i D_{\boldsymbol{\beta}^i} & -\frac{1}{2}(D_{\boldsymbol{\alpha}^i} + D_{\boldsymbol{\beta}^i})\Lambda_i \\ -\frac{1}{2}\Lambda_i(D_{\boldsymbol{\alpha}^i} + D_{\boldsymbol{\beta}^i}) & \Lambda_i \end{bmatrix} \begin{bmatrix} \Delta v^{(i)} \\ \Delta z^{(i)} \end{bmatrix} - F(\Delta z^{(N)})^T (\Delta z^{(N)}) > 0. \quad (39)
$$

For the second term in (39), for each layer $L_i$, we have

$$
\begin{bmatrix} \Delta v^{(i)} \\ \Delta z^{(i)} \end{bmatrix}^T \begin{bmatrix} D_{\boldsymbol{\alpha}^i} \Lambda_i D_{\boldsymbol{\beta}^i} & -\frac{1}{2}(D_{\boldsymbol{\alpha}^i} + D_{\boldsymbol{\beta}^i})\Lambda_i \\ -\frac{1}{2}\Lambda_i(D_{\boldsymbol{\alpha}^i} + D_{\boldsymbol{\beta}^i}) & \Lambda_i \end{bmatrix} \begin{bmatrix} \Delta v^{(i)} \\ \Delta z^{(i)} \end{bmatrix} \leq 0.
$$

Expanding this block-diagonal form, with $\lambda_j^i$ being the $j^{th}$ diagonal entry of $\Lambda_i$, we obtain equivalently

$$
Q_j^{(i)}(\alpha_j^i, \beta_j^i, \lambda_j^i) \triangleq \sum_{j=1}^{d_i} \begin{bmatrix} \Delta v_j^{(i)} \\ \Delta z_j^{(i)} \end{bmatrix}^T \begin{bmatrix} \alpha_j^i \lambda_j^i \beta_j^i & -\frac{1}{2}(\alpha_j^i + \beta_j^i)\lambda_j^i \\ -\frac{1}{2}(\alpha_j^i + \beta_j^i)\lambda_j^i & \lambda_j^i \end{bmatrix} \begin{bmatrix} \Delta v_j^{(i)} \\ \Delta z_j^{(i)} \end{bmatrix} \leq 0. \quad (40)
$$

Let $\mathcal{J}_{i,1} = \{j : [\tilde{\alpha}_j^i, \tilde{\beta}_j^i] \subset [\hat{\alpha}_j^i, \hat{\beta}_j^i]\}$ and $\mathcal{J}_{i,2} = \{j : [\tilde{\alpha}_j^i, \tilde{\beta}_j^i] = [\hat{\alpha}_j^i, \hat{\beta}_j^i]\}$. Then, we split (40) as

$$\sum_{j=1}^{d_i} Q_j^{(i)}(\alpha_j^i, \beta_j^i, \lambda_j^i) = \sum_{j \in \mathcal{J}_{i,1}} Q_j^{(i)}(\alpha_j^i, \beta_j^i, \lambda_j^i) + \sum_{j \in \mathcal{J}_{i,2}} Q_j^{(i)}(\alpha_j^i, \beta_j^i, \lambda_j^i) \leq 0.$$

For $j \in \mathcal{J}_{i,1}$, $[\tilde{\alpha}_j^i, \tilde{\beta}_j^i] \subset [\hat{\alpha}_j^i, \hat{\beta}_j^i]$. With the slope bounds assumption as in Assumption (1) and $\lambda_j^i \geq 0$, we have

$$0 \leq Q_j^{(i)}(\tilde{\alpha}_j^i, \tilde{\beta}_j^i, \lambda_j^i) \leq Q_j^{(i)}(\hat{\alpha}_j^i, \hat{\beta}_j^i, \lambda_j^i), \quad \forall \Delta v_j^{(i)}, \Delta z_j^{(i)}, \lambda_j^i \geq 0,$$

For $j \in \mathcal{J}_{i,2}$,

$$Q_j^{(i)}(\tilde{\alpha}_j^i, \tilde{\beta}_j^i, \lambda_j^i) = Q_j^{(i)}(\hat{\alpha}_j^i, \hat{\beta}_j^i, \lambda_j^i).$$

Then, for all $(\Delta v^{(i)}, \Delta z^{(i)})$, we have

$$\sum_{j=1}^{d_i} Q_j^{(i)}(\tilde{\alpha}_j^i, \tilde{\beta}_j^i, \lambda_j^i) \leq \sum_{j=1}^{d_i} Q_j^{(i)}(\hat{\alpha}_j^i, \hat{\beta}_j^i, \lambda_j^i)$$

Substituting the above inequality into (39), we have, for any fixed choice of $\{\Lambda_i\}_{i=1}^{N-1}$ with $\Lambda_i \in \mathbb{D}_+$,

$$(\Delta z^{(0)})^T \Delta z^{(0)} + \sum_{i=1}^{N-1} \sum_{j=1}^{d_i} Q_j^{(i)}(\tilde{\alpha}_j^i, \tilde{\beta}_j^i, \lambda_j^i) - F(\Delta z^{(N)})^T(\Delta z^{(N)})$$

$$\leq (\Delta z^{(0)})^T \Delta z^{(0)} + \sum_{i=1}^{N-1} \sum_{j=1}^{d_i} Q_j^{(i)}(\hat{\alpha}_j^i, \hat{\beta}_j^i, \lambda_j^i) - F(\Delta z^{(N)})^T(\Delta z^{(N)}).$$

This means that for any fixed choice of $\{\Lambda_i\}$ with $\Lambda_i \in \mathbb{D}_+$, the largest value $\tilde{F}$ and the largest value $\hat{F}$ such that (39) holds with slope bounds $\{\tilde{\boldsymbol{\alpha}}^i, \tilde{\boldsymbol{\beta}}^i\}$ and $\{\hat{\boldsymbol{\alpha}}^i, \hat{\boldsymbol{\beta}}^i\}$ respectively, satisfy

$$\tilde{F}(\{\Lambda_i\}) \geq \hat{F}(\{\Lambda_i\}).$$

Consequently, taking the supremum over all choices $\{\Lambda_i\}$ with $\Lambda_i \in \mathbb{D}_+$, we obtain

$$\tilde{F}^* \geq \hat{F}^*,$$

where $\tilde{F}^* = \sup_{\{\Lambda_i\}} \tilde{F}(\{\Lambda_i\})$, $\hat{F}^* = \sup_{\{\Lambda_i\}} F(\{\Lambda_i\})$.

Therefore, the corresponding Lipschitz upper bounds satisfy $\sqrt{1/\tilde{F}^*} \leq \sqrt{1/\hat{F}^*}$. $\qquad\square$

**Proof of Theorem 6**. Let $z \in \mathcal{Z} = \mathcal{B}(z_c, \delta_z)$, and define $\delta z = z - z_c$. By Clarke's Mean Value Theorem, for any $z, z_c \in \mathcal{Z}$, there exists $X = z_c + t(z - z_c) \in \mathcal{Z}$ for some $t \in [0, 1]$, and $v \in \partial g(X)$, such that

$$g(z) - g(z_c) = v^T(z - z_c),$$

where $\partial g(X)$ denotes the Clarke subdifferential of $g$ at $X$, defined as

$$\partial g(X) := \mathrm{co}\left\{\lim_{k\to\infty} \nabla g(z_k) : z_k \to X, \ g \text{ differentiable at } z_k\right\},$$

with $\mathrm{co}$ denoting the convex hull.

Since $g$ is locally Lipschitz with constant $\mathbf{L}$ over $\mathcal{Z}$, for any $v \in \partial g(X)$ and any $X \in \mathcal{Z}$,

$$|g(z) - g(z_c)| \leq \mathbf{L}\|\delta z\|_2.$$

Therefore,

$$g(z_c) - \mathbf{L}\|\delta z\|_2 \le g(z) \le g(z_c) + \mathbf{L}\|\delta z\|_2,$$

which is exactly (20).

**Proof of Proposition 3**. We require $[\alpha_l^i, \beta_l^i]$ to contain all subgradients of $\phi$ over $v \in [a, b]$, that is,

$$\bigcup_{v \in [a,b]} \partial\phi(v) \subseteq [\alpha_l^i, \beta_l^i].$$

Thus, the minimal (tightest) interval is naturally given by

$$\alpha_l^i = \inf_{v \in [a,b]} \inf\{\partial\phi(v)\}, \quad \beta_l^i = \sup_{v \in [a,b]} \sup\{\partial\phi(v)\}.$$

□

**Proof of Proposition 4**. Observe that for any row index $l$, the $l$-th diagonal entry of $W_i(M_{i-1})^{-1}W_i^T$ is

$$\left(W_i(M_{i-1})^{-1}W_i^T\right)_{(l,l)} = (W_i)_{(l,\bullet)}(M_{i-1})^{-1}(W_i)_{(l,\bullet)}^T. \tag{41}$$

By Lemma 1, if $M_{i-1} > 0$, then $AA^T(M_{i-1})^{-1}$ and $A(M_{i-1})^{-1}A^T$ share the same nonzero eigenvalues for any matrix $A$. Applying this to the row vector $(W_i)_{(l,\bullet)}$, we see that $(W_i)_{(l,\bullet)}^T(W_i)_{(l,\bullet)}(M_{i-1})^{-1}$ and $(W_i)_{(l,\bullet)}(M_{i-1})^{-1}(W_i)_{(l,\bullet)}^T$ share the same nonzero eigenvalues. Therefore,

$$\sigma_{\max}\left((W_i)_{(l,\bullet)}^T(W_i)_{(l,\bullet)}(M_{i-1})^{-1}\right) = \sigma_{\max}\left((W_i)_{(l,\bullet)}(M_{i-1})^{-1}(W_i)_{(l,\bullet)}^T\right) = (W_i)_{(l,\bullet)}(M_{i-1})^{-1}(W_i)_{(l,\bullet)}^T.$$

The last equality holds as $(W_i)_{(l,\bullet)}(M_{i-1})^{-1}(W_i)_{(l,\bullet)}^T$ is a scalar. Combining with (41) completes the proof. □

**Proof of Lemma 2**. Notice that for each $l = 1, \ldots, d_i$,

$$\mathbf{d}_l^{(i)} = \left(W_i(M_{i-1})^{-1}W_i^T\right)_{(l,l)} = (W_i)_{(l,\bullet)}(M_{i-1})^{-1}(W_i)_{(l,\bullet)}^T.$$

Let $A_i = (M_{i-1})^{-1}W_i^T \in \mathbb{R}^{d_{i-1} \times d_i}$. Then, for each $l$,

$$\mathbf{d}_l^{(i)} = \sum_{k=1}^{d_{i-1}} (W_i)_{(l,k)}(A_i)_{(k,l)}.$$

Now, observe that $(W_i)_{(\bullet,k)}$ is the $k$-th column of $W_i$, and $(A_i)_{(k,\bullet)}$ is the $k$-th row of $A_i$. The elementwise product $(W_i)_{(\bullet,k)} \odot \left((A_i)_{(k,\bullet)}\right)^T$ is a vector in $\mathbb{R}^{d_i}$ whose $l$-th entry is $(W_i)_{(l,k)}(A_i)_{(k,l)}$.

Summing over $k$, we have $\mathbf{d}^{(i)} = \sum_{k=1}^{d_{i-1}} (W_i)_{(\bullet,k)} \odot \left((A_i)_{(k,\bullet)}\right)^T$, which establishes (24). □

**Proof of Proposition 5.** For each $j \in \{i, \ldots, i+p\}$ with $\boldsymbol{\alpha}^j = \boldsymbol{\beta}^j$, layer $L_j$ acts as

$$z^{(j)} = D_{\boldsymbol{\alpha}^j}v^{(j)} = D_{\boldsymbol{\alpha}^j}\left(W_j z^{(j-1)} + b_j\right). \tag{42}$$

We prove (29) by induction on $s \in \{1, \ldots, p\}$ that

$$v^{(i+s)} = \tilde{W}_{i+s}\, z^{(i-1)} + \tilde{b}_{i+s}, \tag{43}$$

where

$$\tilde{W}_{i+s} = \left(\prod_{j=i}^{i+s-1} W_{j+1}D_{\boldsymbol{\alpha}^j}\right) W_i, \tag{44}$$

$$\tilde{b}_{i+s} = \sum_{k=1}^{s+1} \left[ \prod_{j=i+k-1}^{i+s-1} (W_{j+1} D_{\boldsymbol{\alpha}^j}) \, b_{i+k-1} \right], \tag{45}$$

where the product $W_{j+1} D_{\boldsymbol{\alpha}^j}$ reduces to the identity matrix if $k = s + 1$.

When $s = 1$, at layer $L_{j+1}$, we have

$$v^{(j+1)} = W_{j+1} z^{(j)} + b_{j+1}. \tag{46}$$

Substituting (42) into $v^{(j+1)}$ directly gives

$$v^{(j+1)} = W_{j+1} D_{\boldsymbol{\alpha}^j} \left( W_j z^{(j-1)} + b_j \right) + b_{j+1} = \underbrace{W_{j+1} D_{\boldsymbol{\alpha}^j} W_j}_{\triangleq \tilde{W}_{j+1}} z^{(j-1)} + \underbrace{W_{j+1} D_{\boldsymbol{\alpha}^j} b_j + b_{j+1}}_{\triangleq \tilde{b}_{j+1}}. \tag{47}$$

Now assume (43)-(45) holds for $s$, $s \in \mathbb{Z}_{p-1}$. Using $\boldsymbol{\alpha}^{i+s} = \boldsymbol{\beta}^{i+s}$,

$$
\begin{aligned}
v^{(i+s+1)} &= W_{i+s+1} z^{(i+s)} + b_{i+s+1} \\
&= W_{i+s+1} D_{\boldsymbol{\alpha}^{i+s}} v^{(i+s)} + b_{i+s+1} \\
&= W_{i+s+1} D_{\boldsymbol{\alpha}^{i+s}} \tilde{W}_{i+s} z^{(i-1)} + W_{i+s+1} D_{\boldsymbol{\alpha}^{i+s}} \tilde{b}_{i+s} + b_{i+s+1}.
\end{aligned}
\tag{48}$$

By induction, we have

$$\tilde{W}_{i+s+1} = W_{i+s+1} D_{\boldsymbol{\alpha}^{i+s}} \tilde{W}_{i+s} = \left( \prod_{j=i}^{i+s} W_{j+1} D_{\boldsymbol{\alpha}^j} \right) W_i, \tag{49}$$

and

$$\tilde{b}_{i+s+1} = W_{i+s+1} D_{\boldsymbol{\alpha}^{i+s}} \tilde{b}_{i+s} + b_{i+s+1} = \sum_{k=1}^{s+1} \left[ \prod_{j=i+k-1}^{i+s-1} (W_{j+1} D_{\boldsymbol{\alpha}^j}) \, b_{i+k-1} \right] \tag{50}$$

$\square$

**Proof of Proposition 6.** We first show that the feasible set is non-empty and the maximum of $c_i$ can be attained. As the feasible set with strict inequalities is open, we consider the closed relaxation of (13):

$$\max_{c_i, \Lambda_i} \; c_i \quad \text{s.t.} \quad \begin{bmatrix} \Lambda_i - c_i W_{i+1}^T W_{i+1} & \frac{1}{2} \Lambda_i (D_{\boldsymbol{\alpha}^i} + D_{\boldsymbol{\beta}^i}) W_i \\ \frac{1}{2} W_i^T (D_{\boldsymbol{\alpha}^i} + D_{\boldsymbol{\beta}^i}) \Lambda_i & X_{i-1} \end{bmatrix} \geq 0, \; \Lambda_i \in \mathbb{D}_+, \; c_i \geq 0. \tag{51}$$

Select an $\varepsilon$ such that $0 < \varepsilon < \min\{1, 2/\sigma_{max}((D_{\boldsymbol{\alpha}^i} + D_{\boldsymbol{\beta}^i}) W_i X_{i-1}^{-1} W_i^T (D_{\boldsymbol{\alpha}^i} + D_{\boldsymbol{\beta}^i}))\}$ and let

$$\Lambda_i = \varepsilon I, \qquad c_i = \frac{\varepsilon}{4 \, \sigma_{max}(W_{i+1}^T W_{i+1})}.$$

Applying the Schur complement to the LMI in (51) with the fact that $X_{i-1} > 0$, we obtain that (51) is strictly feasible. Thus Slater's condition holds, implying that the feasible set of (51) is closed and convex. As the objective is linear, and the optimal value is finite, we conclude that the maximum $c_i$ is attainable.

Denote $(c_i^*, \Lambda_i^*)$ to be an optimal solution. Now we prove that for any $\Lambda_i > \Lambda_i^*$ and $c_i^*$, the constraints in (13) are all satisfied. Note that constraints on $\Lambda_i$ and $c_i$ are automatically satisfied and we focus on the LMI. The LMI in (13) is equivalent to the following statement. For any $\forall \begin{bmatrix} x \\ y \end{bmatrix} \in \mathbb{R}^{d_i + d_{i-1}} \setminus \{0\}$,

$$
\begin{aligned}
x^T \Lambda_i x - c_i \, (W_{i+1} x)^T (W_{i+1} x) &+ \frac{1}{2} x^T \Lambda_i (D_{\boldsymbol{\alpha}^i} + D_{\boldsymbol{\beta}^i}) W_i y + \frac{1}{2} y^T W_i^T (D_{\boldsymbol{\alpha}^i} + D_{\boldsymbol{\beta}^i}) \Lambda_i x \\
&+ y^T \Big( M_{i-1} + W_i^T D_{\boldsymbol{\alpha}^i} \Lambda_i D_{\boldsymbol{\beta}^i} W_i \Big) y \; > \; 0.
\end{aligned}
\tag{52}$$

Let $\mathcal{J}_i \subsetneq \mathbb{Z}_{d_i}$ be the index set where $(\boldsymbol{\alpha}^i)_{\mathcal{J}_i} = (\boldsymbol{\beta}^i)_{\mathcal{J}_i}$ and define $\mathcal{M}_i \triangleq \mathbb{Z}_{d_i} \backslash \mathcal{J}_i \neq \emptyset$. We further split the left hand side of (52) by index sets $\mathcal{J}_i$ and $\mathcal{M}_i$ as

$$
\begin{aligned}
H(\Lambda_i, c_i) \triangleq\ & [(x)_{\mathcal{J}_i}]^T (\Lambda_i)_{(\mathcal{J}_i, \mathcal{J}_i)}(x)_{\mathcal{J}_i} + [(x)_{\mathcal{M}_i}]^T (\Lambda_i)_{(\mathcal{M}_i, \mathcal{M}_i)}(x)_{\mathcal{M}_i} + y^T M_{i-1} y \\
& + [(W_i y)_{\mathcal{J}_i}]^T \Big( D_{\boldsymbol{\alpha}^i} \Lambda_i D_{\boldsymbol{\beta}^i} \Big)_{(\mathcal{J}_i, \mathcal{J}_i)} (W_i y)_{\mathcal{J}_i} + [(W_i y)_{\mathcal{M}_i}]^T \Big( D_{\boldsymbol{\alpha}^i} \Lambda_i D_{\boldsymbol{\beta}^i} \Big)_{(\mathcal{M}_i, \mathcal{M}_i)} (W_i y)_{\mathcal{M}_i} \\
& + \frac{1}{2} \Big( x^T \Lambda_i (D_{\boldsymbol{\alpha}^i} + D_{\boldsymbol{\beta}^i}) \Big)_{(\bullet, \mathcal{J}_i)} (W_i y)_{\mathcal{J}_i} + \frac{1}{2} \Big( x^T \Lambda_i (D_{\boldsymbol{\alpha}^i} + D_{\boldsymbol{\beta}^i}) \Big)_{(\bullet, \mathcal{M}_i)} (W_i y)_{\mathcal{M}_i} \\
& + \frac{1}{2} [(W_i y)_{\mathcal{J}_i}]^T \Big( (D_{\boldsymbol{\alpha}^i} + D_{\boldsymbol{\beta}^i}) \Lambda_i x \Big)_{\mathcal{J}_i} + \frac{1}{2} [(W_i y)_{\mathcal{M}_i}]^T \Big( (D_{\boldsymbol{\alpha}^i} + D_{\boldsymbol{\beta}^i}) \Lambda_i x \Big)_{\mathcal{M}_i} \\
& - c_i \Big( [(x)_{\mathcal{J}_i}]^T \Big( [(W_{i+1})_{(\bullet, \mathcal{J}_i)}]^T (W_{i+1})_{(\bullet, \mathcal{J}_i)} \Big) (x)_{\mathcal{J}_i} + [(x)_{\mathcal{M}_i}]^T \Big( [(W_{i+1})_{(\bullet, \mathcal{M}_i)}]^T (W_{i+1})_{(\bullet, \mathcal{M}_i)} \Big) (x)_{\mathcal{M}_i} \\
& + [(x)_{\mathcal{J}_i}]^T \Big( [(W_{i+1})_{(\bullet, \mathcal{J}_i)}]^T (W_{i+1})_{(\bullet, \mathcal{M}_i)} \Big) (x)_{\mathcal{M}_i} + [(x)_{\mathcal{M}_i}]^T \Big( [(W_{i+1})_{(\bullet, \mathcal{M}_i)}]^T (W_{i+1})_{(\bullet, \mathcal{J}_i)} (x)_{\mathcal{J}_i} \Big) > 0.
\end{aligned}
$$

We denote the part dependent on $(\Lambda_i)_{(\mathcal{J}_i, \mathcal{J}_i)}$ to be

$$
\begin{aligned}
G((\Lambda_i)_{(\mathcal{J}_i, \mathcal{J}_i)}) \triangleq\ & [(x)_{\mathcal{J}_i}]^T (\Lambda_i)_{(\mathcal{J}_i, \mathcal{J}_i)}(x)_{\mathcal{J}_i} + [(W_i y)_{\mathcal{J}_i}]^T \Big( D_{\boldsymbol{\alpha}^i} \Lambda_i D_{\boldsymbol{\beta}^i} \Big)_{(\mathcal{J}_i, \mathcal{J}_i)} (W_i y)_{\mathcal{J}_i} \\
& + \frac{1}{2} \Big( x^T \Lambda_i (D_{\boldsymbol{\alpha}^i} + D_{\boldsymbol{\beta}^i}) \Big)_{(\mathcal{J}_i, \mathcal{J}_i)} (W_i y)_{\mathcal{J}_i} + \frac{1}{2} [(W_i y)_{\mathcal{J}_i}]^T \Big( (D_{\boldsymbol{\alpha}^i} + D_{\boldsymbol{\beta}^i}) \Lambda_i x \Big)_{\mathcal{J}_i} \\
=\ & ((x)_{\mathcal{J}_i} + D_{\boldsymbol{\alpha}^i} (W_i y)_{\mathcal{J}_i})^T \Lambda_i ((x)_{\mathcal{J}_i} + D_{\boldsymbol{\alpha}^i} (W_i y)_{\mathcal{J}_i})
\end{aligned}
$$

The last equality holds by the definition of $\mathcal{J}_i$, $(D_{\boldsymbol{\alpha}^i})_{\mathcal{J}_i} = (D_{\boldsymbol{\beta}^i})_{\mathcal{J}_i}$. Therefore for any $\Lambda_i > \Lambda_i^*$,

$$
\begin{aligned}
& H(\Lambda_i, c_i^*) - H(\Lambda_i^*, c_i^*) \\
& = G((\Lambda_i)_{(\mathcal{J}_i, \mathcal{J}_i)}) - G((\Lambda_i)_{(\mathcal{J}_i, \mathcal{J}_i)}^*) \\
& = ((x)_{\mathcal{J}_i} + D_{\boldsymbol{\alpha}^i} (W_i y)_{\mathcal{J}_i})^T \Lambda_i ((x)_{\mathcal{J}_i} + D_{\boldsymbol{\alpha}^i} (W_i y)_{\mathcal{J}_i}) - ((x)_{\mathcal{J}_i} + D_{\boldsymbol{\alpha}^i} (W_i y)_{\mathcal{J}_i})^T \Lambda_i^* ((x)_{\mathcal{J}_i} + D_{\boldsymbol{\alpha}^i} (W_i y)_{\mathcal{J}_i}) \\
& \geq 0.
\end{aligned}
$$

By the arbitrariness of $x$ and $y$, we conclude that constraints in (13) are satisfied with any $\Lambda_i > \Lambda_i^*$ while the optimum $c_i^*$ is attained. Taking $l = \max_{j \in \mathbb{Z}_{d_i}} \left\{ (\Lambda_i^*)_{(j,j)} \right\}$ completes the proof. $\qquad \square$

**Proof of Theorem 7.** Under the mild assumption $\boldsymbol{\alpha}^i \odot \boldsymbol{\beta}^i \geq 0$ for all $i \in \mathbb{Z}_{N-1}$ and using the fact that $M_0 = I > 0$, it suffices to show by induction that at layer $L_i$, $i \in \mathbb{Z}_{N-1}$, the following claims hold:

(i) Given $M_{i-1} > 0$, the feasible set of the SDP constraints in (13) and (14) is always nonempty and the closed-form solution (15) is always well-defined and positive.

(ii) Optimization problems (13), (14), and equation (15) can each yield a solution $\Lambda_i$ such that $M_i > 0$.

Since (14) is a special case of (13) with $\Lambda_i = \lambda_i I$, it suffices to prove feasibility and $M_i > 0$ for (14); the same conclusions for (13) then follow directly, and no separate proof is required.

Regarding (14), we prove that at stage $i \in \mathbb{Z}_{N-1}$, there exists a $\lambda_i \geq 0$ and $c_i > 0$, such that

$$
S_i \triangleq \begin{bmatrix} \lambda_i I - c_i W_{i+1}^T W_{i+1} & \frac{1}{2} \lambda_i (D_{\boldsymbol{\alpha}^i} + D_{\boldsymbol{\beta}^i}) W_i \\ \frac{1}{2} \lambda_i W_i^T (D_{\boldsymbol{\alpha}^i} + D_{\boldsymbol{\beta}^i}) & \tilde{M}_{i-1} \end{bmatrix} > 0, \quad \tilde{M}_i > 0, \tag{53}
$$

where $\tilde{M}_i \triangleq \lambda_i I - \frac{1}{4} \lambda_i^2 (D_{\boldsymbol{\alpha}^i} + D_{\boldsymbol{\beta}^i}) W_i (X_{i-1})^{-1} W_i^T (D_{\boldsymbol{\alpha}^i} + D_{\boldsymbol{\beta}^i})$, $i \in \mathbb{Z}_N$ and $\tilde{M}_0 = I$.

Given $\tilde{M}_{i-1} > 0$, by the Schur complement, $S_i > 0$ is equivalent to

$$
T_i \triangleq \lambda_i I - c_i W_{i+1}^T W_{i+1} - \frac{1}{4} \lambda_i^2 (D_{\boldsymbol{\alpha}^i} + D_{\boldsymbol{\beta}^i}) W_i \tilde{M}_{i-1}^{-1} W_i^T (D_{\boldsymbol{\alpha}^i} + D_{\boldsymbol{\beta}^i}) > 0.
$$

Let

$$
\sigma_i = \sigma_{\max} \Big( (D_{\boldsymbol{\alpha}^i} + D_{\boldsymbol{\beta}^i}) W_i \tilde{M}_{i-1}^{-1} W_i^T (D_{\boldsymbol{\alpha}^i} + D_{\boldsymbol{\beta}^i}) \Big) > 0, \qquad \eta_i = \sigma_{\max}(W_{i+1}^T W_{i+1}) > 0. \tag{54}
$$

Choose $\lambda_i = \frac{2}{\sigma_i} > 0$ and $c_i = \frac{0.9}{\sigma_i \eta_i} > 0$. Then,

$$T_i \geq \frac{2}{\sigma_i}I - \frac{0.9}{\sigma_i \eta_i}W_{i+1}^T W_{i+1} - \frac{1}{4}\frac{4}{\sigma_i^2}\sigma_i I$$

$$= \frac{2}{\sigma_i}I - \frac{0.9}{\sigma_i \eta_i}W_{i+1}^T W_{i+1} - \frac{1}{\sigma_i}I$$

$$\geq \frac{1}{\sigma_i}I - \frac{0.9}{\sigma_i}I > 0,$$

where the last inequality uses $W_{i+1}^T W_{i+1} \leq \eta_i I$. Finally, $\tilde{M}_i = T_i + c_i W_{i+1}^T W_{i+1} \geq T_i > 0$.

With (53) being feasible for $i \in \mathbb{Z}_{N-1}$, the LMI in (14) is naturally satisfied with $\lambda_i \geq 0$ and $c_i > 0$ because for $i \in \mathbb{Z}_{N-2}$,

$$\begin{bmatrix} \lambda_i I - c_i W_{i+1}^T W_{i+1} & \frac{1}{2}\lambda_i(D_{\boldsymbol{\alpha}^i} + D_{\boldsymbol{\beta}^i})W_i \\ \frac{1}{2}\lambda_i W_i^T(D_{\boldsymbol{\alpha}^i} + D_{\boldsymbol{\beta}^i}) & \tilde{X}_{i-1} \end{bmatrix} = S_i + \begin{bmatrix} 0 & 0 \\ 0 & \lambda_{i+1}W_{i+1}^T D_{\boldsymbol{\alpha}^{i+1}} D_{\boldsymbol{\beta}^{i+1}} W_{i+1} \end{bmatrix} \geq S_i > 0,$$

and for $i = N - 1$,

$$\begin{bmatrix} \lambda_i I - c_i W_{i+1}^T W_{i+1} & \frac{1}{2}\lambda_i(D_{\boldsymbol{\alpha}^i} + D_{\boldsymbol{\beta}^i})W_i \\ \frac{1}{2}\lambda_i W_i^T(D_{\boldsymbol{\alpha}^i} + D_{\boldsymbol{\beta}^i}) & \tilde{X}_{i-1} \end{bmatrix} = S_i > 0.$$

Then we proceed to show that claims (i) and (ii) hold for the closed-form solution (15). Given that $M_{i-1} > 0$ at stage $i$, (15) is well-defined and positive. So it suffices to show that using $\lambda_i$ as in (15) always guarantees $M_i > 0$, $i \in \{0\} \cup \mathbb{Z}_{N-1}$.

At stage $i$, recall that

$$M_i = \lambda_i I - \frac{1}{4}\lambda_i^2(D_{\boldsymbol{\alpha}^{i,\mathrm{adj}}} + D_{\boldsymbol{\beta}^{i,\mathrm{adj}}})W_i X_{i-1}^{-1} W_i^T(D_{\boldsymbol{\alpha}^{i,\mathrm{adj}}} + D_{\boldsymbol{\beta}^{i,\mathrm{adj}}})$$

$$= \lambda_i I - \frac{1}{4}\lambda_i^2(D_{\boldsymbol{\alpha}^{i,\mathrm{adj}}} + D_{\boldsymbol{\beta}^{i,\mathrm{adj}}})W_i M_{i-1}^{-1} W_i^T(D_{\boldsymbol{\alpha}^{i,\mathrm{adj}}} + D_{\boldsymbol{\beta}^{i,\mathrm{adj}}})$$

The second equality holds because $D_{\boldsymbol{\alpha}^{i,\mathrm{adj}}} \odot D_{\boldsymbol{\beta}^{i,\mathrm{adj}}} = 0$.

Let $\bar{\sigma}_i = \sigma_{\max}\left((D_{\boldsymbol{\alpha}^{i,\mathrm{adj}}} + D_{\boldsymbol{\beta}^{i,\mathrm{adj}}})W_i M_{i-1}^{-1} W_i^T(D_{\boldsymbol{\alpha}^{i,\mathrm{adj}}} + D_{\boldsymbol{\beta}^{i,\mathrm{adj}}})\right) > 0$. For the closed-form solution (15), $\lambda_i = \frac{2}{\bar{\sigma}_i} > 0$. By definition of $\bar{\sigma}_i$, we have $(D_{\boldsymbol{\alpha}^{i,\mathrm{adj}}} + D_{\boldsymbol{\beta}^{i,\mathrm{adj}}})W_i M_{i-1}^{-1} W_i^T(D_{\boldsymbol{\alpha}^{i,\mathrm{adj}}} + D_{\boldsymbol{\beta}^{i,\mathrm{adj}}}) \leq \bar{\sigma}_i I$. Therefore, $M_i \geq \frac{2}{\bar{\sigma}_i}I - \frac{1}{\bar{\sigma}_i}I = \frac{1}{\bar{\sigma}_i}I > 0$. $\qquad\square$

**Proof of Theorem 8.** Let $\mathcal{Z} = \mathcal{B}(z_c, \|\delta_z\|_2)$ be the input region. Consider any layer $L_i, i \in \mathbb{Z}_{N-1}$. Given the validity of the Lipschitz constant $\mathbf{L}_{\bullet,l}^{(i)}$ for each $l \in \mathbb{Z}_{d_{i-1}}$, i.e., for any $z_1, z_2 \in \mathcal{Z}$, the map from $z$ to the $l$-th component of $z^{(i)}$ satisfies

$$|z_l^{(i)}(z_1) - z_l^{(i)}(z_2)| \leq \mathbf{L}_{\bullet,l}^{(i)}\|z_1 - z_2\|_2.$$

In particular, taking $z_2 = z_c$ and arbitrary $z_1 \in \mathcal{Z}$, and using the fact that $\|z_1 - z_c\|_2 \leq \|\delta_z\|_2$, we have

$$|z_l^{(i)}(z) - z_l^{(i)}(z_c)| \leq \mathbf{L}_{\bullet,l}^{(i)}\|\delta_z\|_2, \qquad \forall z \in \mathcal{Z}.$$

Therefore, for each $l$,

$$z_l^{(i)}(z) \in \left[z_l^{(i)}(z_c) - \mathbf{L}_{\bullet,l}^{(i)}\|\delta_z\|_2, \ z_l^{(i)}(z_c) + \mathbf{L}_{\bullet,l}^{(i)}\|\delta_z\|_2\right], \qquad \forall z \in \mathcal{Z}.$$

Thus, the region $\mathcal{V}^i$ defined in Algorithm 1 is a valid enclosure for all $v^{(i)}$ over $\mathcal{Z}$. Next, by construction in Algorithm 1, the refined slope bounds are given by

$$\alpha_l^i = \inf_{v \in \mathcal{V}_l^i} \inf \partial\sigma(v), \qquad \beta_l^i = \sup_{v \in \mathcal{V}_l^i} \sup \partial\sigma(v).$$

Since $v_l^{(i)}(z) \in \mathcal{V}_l^i$ for all $z \in \mathcal{Z}$, it follows that

$$\alpha_l^i \leq \inf \partial\sigma(v_l^{(i)}(z)), \qquad \beta_l^i \geq \sup \partial\sigma(v_l^{(i)}(z)), \qquad \forall z \in \mathcal{Z}.$$

Thus, the refined slope bounds $\boldsymbol{\alpha}^i, \boldsymbol{\beta}^i$ are valid for all $z \in \mathcal{Z}$, as claimed. $\qquad\square$

**Proof of Theorem 9**. Let $z_1^{(i)}$, $z_2^{(i)}$ be the layer outputs for two arbitrary inputs, and define $\Delta z^{(i)} = z_1^{(i)} - z_2^{(i)}$, $\Delta v^{(i)} = v_1^{(i)} - v_2^{(i)}$ for all $i \in \{0\} \cup \mathbb{Z}_N$. We claim and prove by induction on the layer index $i$:

(i) $\sqrt{\sigma_{max}\left(\left[(W_i)_{(l,\bullet)}\right]^T (W_i)_{(l,\bullet)}(M_{i-1})^{-1}\right)}$ are valid Lipschitz constants for $\forall l \in \mathbb{Z}_{d_i}$ as defined in (6),

(ii) $X_{i-1} = M_{i-1} + W_i^T D_{\boldsymbol{\alpha}^i} \Lambda_i D_{\boldsymbol{\beta}^i} W_i > 0$, and

(iii) $M_i = \Lambda_i - \frac{1}{4}\Lambda_i(D_{\boldsymbol{\alpha}^i} + D_{\boldsymbol{\beta}^i})W_i(X_{i-1})^{-1}W_i^T(D_{\boldsymbol{\alpha}^i} + D_{\boldsymbol{\beta}^i})\Lambda_i > 0.$

For the first layer ($i = 1$), we have

$$\Delta v^{(1)} = W_1 \Delta z^{(0)}.$$

For any neuron $l \in \mathbb{Z}_{d_1}$, the $l$-th entry is

$$\left(\Delta v^{(1)}\right)_l = (W_1)_{(l,\bullet)}\Delta z^{(0)}.$$

By the Cauchy-Schwarz inequality,

$$\left|\left(\Delta v^{(1)}\right)_l\right| \leq \|(W_1)_{(l,\bullet)}\|_2 \|\Delta z^{(0)}\|_2 = \sqrt{\sigma_{max}\left(\left[(W_1)_{(l,\bullet)}\right]^T (W_1)_{(l,\bullet)}\right)}\|\Delta z^{(0)}\|_2.$$

Since $M_0 = I$, the bound can be written equivalently as

$$\mathbf{L}_{\bullet,l}^{(0,1)} = \sqrt{\sigma_{max}\left(\left[(W_1)_{(l,\bullet)}\right]^T (W_1)_{(l,\bullet)}(M_0)^{-1}\right)}$$

From ECLipsE-Gen-Acc in (13) at $i = 1$, $\Lambda_1$ satisfies

$$\begin{bmatrix} \Lambda_1 - c_1 W_2^T W_2 & \frac{1}{2}\Lambda_1(D_{\boldsymbol{\alpha}^1} + D_{\boldsymbol{\beta}^1})W_1 \\ \frac{1}{2}W_1^T(D_{\boldsymbol{\alpha}^1} + D_{\boldsymbol{\beta}^1})\Lambda_1 & X_0 \end{bmatrix} > 0, \Lambda_1 \in \mathbb{D}_+, \; c_1 > 0.$$

By Schur complement, this is equivalent to

$$X_0 > 0, \quad \Lambda_1 - c_1 W_2^T W_2 - \frac{1}{4}\Lambda_1(D_{\boldsymbol{\alpha}^1} + D_{\boldsymbol{\beta}^1})W_1(X_0)^{-1}W_1^T(D_{\boldsymbol{\alpha}^1} + D_{\boldsymbol{\beta}^1})\Lambda_1 > 0.$$

Therefore, $(X_0)^{-1}$ is well-defined and with $c_1 > 0$,

$$\begin{aligned} M_1 &= (\Lambda_1 - c_1 W_2^T W_2 - \frac{1}{4}\Lambda_1(D_{\boldsymbol{\alpha}^1} + D_{\boldsymbol{\beta}^1})W_1(X_0)^{-1}W_1^T(D_{\boldsymbol{\alpha}^1} + D_{\boldsymbol{\beta}^1})\Lambda_1) + c_1 W_2^T W_2 \\ &> c_1 W_2^T W_2 \geq 0. \end{aligned}$$

Now by induction, assume that at layer $L_i$, $i \in \mathbb{Z}_{N-1}$, we have $X_k > 0$, $\forall k \in \mathbb{Z}_{i-2}$, and $M_j > 0$, $\forall j \in \mathbb{Z}_{i-1}$.

By Theorem 8, $\boldsymbol{\alpha}^j, \boldsymbol{\beta}^j, \forall j \in \mathbb{Z}_{i-1}$, are valid slope bounds. To show the validity of $\mathbf{L}_{\bullet,l}^{(0,i)}$, by Theorem 3, it suffices to prove that there exists $\tilde{\Lambda}_j$, $j \in \mathbb{Z}_{i-1}$ such that

$$\begin{bmatrix} \mathcal{P}_1 & \mathcal{R}_2 & 0 & \cdots & 0 \\ \mathcal{R}_2^T & \mathcal{P}_2 & \mathcal{R}_3 & \cdots & 0 \\ 0 & \mathcal{R}_3^T & \mathcal{P}_3 & \cdots & 0 \\ \vdots & \vdots & \vdots & \ddots & \mathcal{R}_i \\ 0 & 0 & 0 & \mathcal{R}_i^T & \mathcal{P}_i \end{bmatrix} > 0$$

where

$$\mathcal{P}_m = \begin{cases} \tilde{\Lambda}_0 + W_1^T D_{\boldsymbol{\alpha}^1} \tilde{\Lambda}_1 D_{\boldsymbol{\beta}^1} W_1, & m = 1 \\ \tilde{\Lambda}_{m-1} + W_m^T D_{\boldsymbol{\alpha}^m} \tilde{\Lambda}_m D_{\boldsymbol{\beta}^m} W_m, & 2 \le m < i \\ \tilde{\Lambda}_{i-1} - \tilde{F} \left[ (W_i)_{(l,\bullet)} \right]^T (W_i)_{(l,\bullet)}, & m = i \end{cases}$$

$$\mathcal{R}_m = \begin{cases} -\dfrac{1}{2} W_{m-1}^T (D_{\boldsymbol{\alpha}^{m-1}} + D_{\boldsymbol{\beta}^{m-1}}) \tilde{\Lambda}_{m-1}, & 2 \le m \le i-1 \\ -\dfrac{1}{2} W_{i-1}^T \left( D_{\boldsymbol{\alpha}^{i-1}} + D_{\boldsymbol{\beta}^{i-1}} \right)_{(l,l)} (\tilde{\Lambda}_{i-1})_{(l,l)}, & m = i \end{cases}$$

with

$$\tilde{F} = \frac{1}{\sigma_{\max} \left( \left[ (W_i)_{(l,\bullet)} \right]^T (W_i)_{(l,\bullet)} (M_{i-1})^{-1} \right)}. \tag{55}$$

According to Theorem 4, it is equivalent to show that

$$\tilde{X}_k > 0, \quad \forall k \in \mathbb{Z}_{i-2}, \qquad \tilde{X}_{i-1} - \tilde{F}(W_i)_{(l,\bullet)}^T (W_i)_{(l,\bullet)} > 0,$$

where

$$\tilde{X}_k = \begin{cases} I + W_1^T D_{\boldsymbol{\alpha}^1} \tilde{\Lambda}_i D_{\boldsymbol{\beta}^1} W_1 & k = 0 \\ \tilde{\Lambda}_k - \frac{1}{4} \tilde{\Lambda}_k (D_{\boldsymbol{\alpha}^k} + D_{\boldsymbol{\beta}^k}) W_k (X_{k-1})^{-1} W_k^T (D_{\boldsymbol{\alpha}^k} + D_{\boldsymbol{\beta}^k}) \tilde{\Lambda}_k + W_{i+1}^T D_{\boldsymbol{\alpha}^{i+1}} \tilde{\Lambda}_{i+1} D_{\boldsymbol{\beta}^{i+1}} W_{i+1} & k \in \mathbb{Z}_{i-2} \\ \tilde{\Lambda}_{i-1} - \frac{1}{4} \tilde{\Lambda}_{i-1} (D_{\boldsymbol{\alpha}^{i-1}} + D_{\boldsymbol{\beta}^{i-1}}) W_{i-1} (X_{i-2})^{-1} W_{i-1}^T (D_{\boldsymbol{\alpha}^{i-1}} + D_{\boldsymbol{\beta}^{i-1}}) \tilde{\Lambda}_{i-1} & k = i-1 \end{cases}.$$

Let $\tilde{\Lambda}_j$, $j \in \mathbb{Z}_{i-1}$, be decided according to Algorithm 1 from the previous layer, that is, $\tilde{\Lambda}_j = \Lambda_j$, $j \in \mathbb{Z}_{i-1}$. Then $\tilde{X}_k = X_k$, $k \in \mathbb{Z}_{i-2}$ and $\tilde{X}_{i-1} = M_{i-1}$. Then the following statements hold:

(a) (55) is well-defined because by induction, $M_{i-1} > 0$,
(b) by induction, $\tilde{X}_k = X_k > 0$, $k \in \mathbb{Z}_{i-2}$, and
(c) with $\tilde{F}$ from (55), $\tilde{X}_{i-1} - \tilde{F}(W_i)_{(l,\bullet)}^T (W_i)_{(l,\bullet)} = M_{i-1} - \tilde{F}(W_i)_{(l,\bullet)}^T (W_i)_{(l,\bullet)} > 0$.

Therefore, $\sqrt{\sigma_{max} \left( \left[ (W_i)_{(l,\bullet)} \right]^T (W_i)_{(l,\bullet)} (M_{i-1})^{-1} \right)}$ are valid Lipschitz constants for $\forall l \in \mathbb{Z}_{d_i}$ as in (6).

Similarly, the ECLipsE-Gen series of algorithms gives $\Lambda_i$ that satisfy

$$\begin{bmatrix} \Lambda_i - c_i W_{i+1}^T W_{i+1} & \frac{1}{2} \Lambda_i (D_{\boldsymbol{\alpha}^i} + D_{\boldsymbol{\beta}^i}) W_i \\ \frac{1}{2} W_i^T (D_{\boldsymbol{\alpha}^i} + D_{\boldsymbol{\beta}^i}) \Lambda_i & X_{i-1} \end{bmatrix} > 0, \Lambda_i \in \mathbb{D}_+ \ c_i > 0.$$

By Schur complement, this is equivalent to

$$X_{i-1} > 0, \quad \Lambda_i - c_i W_{i+1}^T W_{i+1} - \frac{1}{4} \Lambda_i (D_{\boldsymbol{\alpha}^i} + D_{\boldsymbol{\beta}^i}) W_i (X_{i-1})^{-1} W_i^T (D_{\boldsymbol{\alpha}^i} + D_{\boldsymbol{\beta}^i}) \Lambda_i > 0.$$

Therefore, $(X_{i-1})^{-1}$ is well-defined and with $c_i > 0$,

$$M_i = (\Lambda_i - c_i W_{i+1}^T W_{i+1} - \frac{1}{4} \Lambda_i (D_{\boldsymbol{\alpha}^i} + D_{\boldsymbol{\beta}^i}) W_i (X_{i-1})^{-1} W_i^T (D_{\boldsymbol{\alpha}^i} + D_{\boldsymbol{\beta}^i}) \Lambda_i) + c_i W_{i+1}^T W_{i+1}$$
$$> c_i W_{i+1}^T W_{i+1} \ge 0.$$

This completes the proof of claims (i)-(iii). Proceeding, by Proposition 4 and Lemma 2, $\mathbf{L}_{\bullet,l}^{(0,i)}$ produced by Algorithm 1 is a strict upper bound for the local Lipschitz constant of the $l$-th neuron on layer $L_i$, $l \in \mathbb{Z}_{d_i}, i \in \mathbb{Z}_{N-1}$.

For the final local Lipschitz constant $\mathbf{L}$ estimated by Algorithm 1, the proof follows identically to the neuron-wise case above, except that the final matrix inequality involves $W_N$ instead of $(W_N)_{(l,\bullet)}$. With $F = 1/\sigma_{\max} \left( W_N (M_{N-1})^{-1} W_N^T \right)$, we obtain $\mathbf{L} = \sqrt{1/F}$ to be a valid Lipschitz constant (strict upper bound) for the entire network. □

## A.2 Algorithm

We present the practical algorithm that enhances ECLipsE-Gen-Local with acceleration and stability here.

---

**Algorithm 2** Enhanced ECLipsE-Gen-Local with Acceleration and Numerical Stability

---

1: **Input:** Weights $\{W_i\}_{i=1}^N$, biases $\{b_i\}_{i=1}^N$; activation function $\sigma$; input region $\mathcal{Z} = \mathcal{B}(z_c, \delta_z)$; large scalar $Cap > 0$ for numerical upper bound; variant $\text{ALGO} \in \{\texttt{Acc}, \texttt{Fast}, \texttt{CF}\}$
2: **Output:** Local Lipschitz estimate **L**
3: Set $M_0 \leftarrow I$, $v^{c,(0)} \leftarrow z_c$, $\texttt{skip} \leftarrow 0$
4: **for** $i = 1, 2, \ldots, N - 1$ **do**
5:     Set $W_i^{\text{orig}} \leftarrow W_i$
6:     **if** $\texttt{skip} = 1$ **then**
7:         $W_i \leftarrow W_i \, D_{\boldsymbol{\alpha}^{i-1}} \, W_{i-1}$
8:     **end if**
9:     Compute $\mathbf{d}^{(i)}$ with $\mathbf{d}_l^{(i)} = (W_i(M_{i-1})^{-1}W_i^T)_{(l,l)}$ for $l = 1, \ldots, d_i$, using (24)
10:    Set $\mathbf{L}^{(i)} \leftarrow \left[\sqrt{\mathbf{d}_1^{(i)}}, \ldots, \sqrt{\mathbf{d}_{d_i}^{(i)}}\right]^T$
11:    Compute $v^{c,(i)} = f^{(i)}(z_c)$ per (25) with $(W_i^{\text{orig}}, b_i)$
12:    Calculate range $\mathcal{V}^i$ for $v^{(i)}$ as in (26)
13:    Refine $\boldsymbol{\alpha}^i, \boldsymbol{\beta}^i$ using $\mathcal{V}^i$ as in (27)
14:    **if** $\boldsymbol{\alpha}^i = \boldsymbol{\beta}^i$ **then**
15:       $\texttt{skip} \leftarrow 1$
16:       **continue**
17:    **else**
18:       $\texttt{skip} \leftarrow 0$
19:    **end if**
20:    Let $\mathcal{J}_i = \{j : \alpha_j^i = \beta_j^i\}$, $\mathcal{M}_i = \mathbb{Z}_{d_i} \setminus \mathcal{J}_i \neq \emptyset$
21:    **Obtain $\Lambda_i$ (and $c_i$) according to Algorithm** 3 (`ACC`), 4 (`Fast`), 5 (`CF`)
22:    Update $M_i$ as in (11) using $D_{\boldsymbol{\alpha}^i}, D_{\boldsymbol{\beta}^i}$
23: **end for**
24: Using (18), compute final $\mathbf{L} = \sqrt{1/F} = \sqrt{\sigma_{max}\left(W_N(X_{N-1})^{-1}W_N^T\right)}$ with $X_{N-1} = M_{N-1}$
25: **return L**

---

**Algorithm 3** Procedure `Acc`: Obtain $\Lambda_i$

---

1: **Input:** $W_i, M_{i-1}, D_{\boldsymbol{\alpha}^i}, D_{\boldsymbol{\beta}^i}$, index sets $\mathcal{J}_i, \mathcal{M}_i$, large scalar $Cap > 0$
2: **Output:** $(\Lambda_i, c_i)$
3: Solve (30) on $\mathcal{M}_i$ to get $(\Lambda_i)_{(\mathcal{M}_i, \mathcal{M}_i)}$ and $c_i$
4: Set $(\Lambda_i)_{j,j} \leftarrow \frac{l_i}{|\mathcal{M}_i|} \sum_{m \in \mathcal{M}_i} (\Lambda_i)_{(m,m)}$ for all $j \in \mathcal{J}_i$   (cf. 31)
5: $(\Lambda_i)_{j,j} \leftarrow \min\{Cap, (\Lambda_i)_{j,j}\}$ for all $j$
6: Compute $X_i$ as in (10) using $\Lambda_i$
7: **if** $X_i > 0$ **then**
8:    **return** $(\Lambda_i, c_i)$
9: **else**
10:   $(\Lambda_i^{\text{fast}}, c_i^{\text{fast}}) \leftarrow$ **Algorithm** 4
11:   $(\Lambda_i^{\text{cf}}, c_i^{\text{cf}}) \leftarrow$ **Algorithm** 5
12:   **return** the pair with larger $c_i$ among $\{(\Lambda_i^{\text{fast}}, c_i^{\text{fast}}), (\Lambda_i^{\text{cf}}, c_i^{\text{cf}})\}$
13: **end if**

---

---

**Algorithm 4** Procedure `Fast`: Obtain $\Lambda_i$

---

1: **Input:** $W_i$, $M_{i-1}$, $D_{\boldsymbol{\alpha}^i}$, $D_{\boldsymbol{\beta}^i}$, index set $\mathcal{M}_i$, large scalar $Cap > 0$
2: **Output:** $(\Lambda_i, c_i)$
3: Solve (32) on $\mathcal{M}_i$ to get $\bar{\lambda}_i$ and $\bar{c}_i$;
4: $\bar{\lambda}_i \leftarrow \min\{Cap, \bar{\lambda}_i\}$
5: Set $\bar{\Lambda}_i \leftarrow \bar{\lambda}_i I$
6: Compute $X_i$ as in (10) using $\bar{\Lambda}_i$
7: **if** $X_i > 0$ **then**
8:     **return** $(\bar{\Lambda}_i, \bar{c}_i)$
9: **else**
10:     **return** $(\Lambda_i, c_i) \leftarrow$ **Algorithm** 5
11: **end if**

---

**Algorithm 5** Procedure `CF`: Obtain $\Lambda_i$

---

1: **Input:** $W_i$, $M_{i-1}$, $D_{\boldsymbol{\alpha}^i}$, $D_{\boldsymbol{\beta}^i}$, activation function $\sigma$
2: **Output:** $(\Lambda_i, c_i)$
3: **Assert** $\boldsymbol{\alpha}^i \odot \boldsymbol{\beta}^i \geq 0$
4: **for** $j = 1, 2, \ldots, d_i$ **do**
5:     **if** $0 \leq (\boldsymbol{\alpha}^i)_j \leq (\boldsymbol{\beta}^i)_j$ **then**
6:         $(\boldsymbol{\alpha}^{i,\mathrm{adj}})_j \leftarrow 0, \quad (\boldsymbol{\beta}^{i,\mathrm{adj}})_j \leftarrow (\boldsymbol{\beta}^i)_j$
7:     **else if** $(\boldsymbol{\alpha}^i)_j \leq (\boldsymbol{\beta}^i)_j \leq 0$ **then**
8:         $(\boldsymbol{\alpha}^{i,\mathrm{adj}})_j \leftarrow (\boldsymbol{\alpha}^i)_j, \quad (\boldsymbol{\beta}^{i,\mathrm{adj}})_j \leftarrow 0$
9:     **end if**
10: **end for**
11: Obtain $\lambda_i$ via (15) using $\boldsymbol{\alpha}^{i,\mathrm{adj}}$ and $\boldsymbol{\beta}^{i,\mathrm{adj}}$
12: Set $\Lambda_i \leftarrow \lambda_i I$
13: Compute $M_i$ as in (17) using $\Lambda_i$
14: Compute $c_i \leftarrow 1/\sigma_{\max}\big(W_{i+1}\,(M_i)^{-1}\,W_{i+1}^T\big)$
15: **return** $(\Lambda_i, c_i)$

---

### A.3   Experimental Details

#### A.3.1   Computational Resources

All algorithms except LipDiff are implemented on a Windows laptop with a 12-core CPU and 16 GB of RAM. LipDiff is accelerated using a compute node equipped with a single NVIDIA A100 GPU (80 GB onboard memory) and 512 GB of system RAM.

#### A.3.2   Randomly Generated Neural Networks

For the experiments in Section 4.1, network weights are generated such that the $\ell_2$-norm of each layer weight lies in $[0.8, 2.5]$. This is accomplished by first sampling a target value uniformly from $[0.8, 2.5]$ for each layer, and then normalizing the randomly generated weight matrix. Similarly, for the networks in Section 4.2, the $\ell_2$-norm of each layer weight is constrained to $[2, 2.5]$. When applying Algorithm EClipsE-Gen-Local-Acc, we set $l_i = 100$ in (31).

#### A.3.3   MNIST Training and Robustness Evaluation

We evaluate adversarial robustness on the MNIST dataset, which consists of $28 \times 28$ grayscale images of handwritten digits from 0 to 9. Each image is vectorized into a 784-dimensional input, and the networks output a 10-dimensional vector corresponding to the ten digit classes. All feedforward networks used in this experiment therefore have input size 784 and output size 10. The models are trained with Adam (learning rate $10^{-3}$, weight decay $10^{-4}$) for up to 50 epochs with early stopping at 98% test accuracy. The baseline uses cross-entropy loss, while the Jacobian regularized model adds a Frobenius norm penalty estimated with one Hutchinson probe (Hoffman et al. (2019)) and penalizing weight $\lambda = 1$.

Adversarial robustness is measured using projected gradient descent (PGD) attacks. Given an image $x$ with label $y$, we optimize the cross-entropy loss with respect to a perturbation $\delta$ subject to the $L_2$ constraint $\|\delta\|_2 \leq \varepsilon$. Starting from a small randomized initialization, we perform 40 steps of gradient ascent with normalized gradients and step size $\alpha = \varepsilon/10$. After each step, the perturbed input $x + \delta$ is projected back onto the $L_2$ ball of radius $\varepsilon$ and clipped to the valid pixel range $[0, 1]^{784}$. We sweep $\epsilon \in \{1/2, 1/4, 1/8, 1/16, 1/32, 1/64, 1/128, 1/256\}$ and report the *failure rate*, defined as the fraction of test examples for which the classifier prediction changes under attack.

### A.3.4 Complete Experimental Results

**Case 1 of Section 4.1:** The Lipschitz constant estimates and computation times for the randomly generated neural networks with the number of layers chosen from $\{5, 10, 15, 20, 25\}$, and number of neurons chosen from $\{10, 20, 40, 60\}$ (small neural networks), are provided below.

| Table 1a: Lipschitz constant estimates | | | | | | |
|---|---|---|---|---|---|---|
| | Neurons\Layers | 5 | 10 | 15 | 20 | 25 |
| Trivial | 10 | 21.028 | 105.687 | 1530.490 | 12360.291 | 564727.209 |
| | 20 | 3.314 | 14.138 | 98.836 | 32738.399 | 34901.424 |
| | 40 | 24.280 | 81.681 | 1208.555 | 5187.447 | 24404.492 |
| | 60 | 2.567 | 109.017 | 4524.267 | 2693.936 | 106596.360 |
| SeqLip | Neurons\Layers | 5 | 10 | 15 | 20 | 25 |
| | 10 | 8.724 | 10.281 | 91.219 | 419.907 | 2206.167 |
| | 20 | >10min | >10min | >10min | >10min | >10min |
| | 40 | | | | | |
| | 60 | | | | | |
| LipSDP-neuron | Neurons\Layers | 5 | 10 | 15 | 20 | 25 |
| | 10 | 4.943 | 2.049 | 8.263 | 4.937 | 26.230 |
| | 20 | 0.635 | 0.305 | 0.415 | 14.109 | 4.502 |
| | 40 | 3.766 | 1.950 | 3.911 | 3.193 | 2.113 |
| | 60 | 0.447 | 2.446 | 16.205 | 1.615 | 7.947 |
| LipSDP-layer | Neurons\Layers | 5 | 10 | 15 | 20 | 25 |
| | 10 | 6.784 | 4.843 | 27.348 | 30.823 | 328.444 |
| | 20 | 0.988 | 0.709 | 1.243 | 66.064 | 26.522 |
| | 40 | 5.537 | 3.800 | 10.111 | 10.932 | 9.567 |
| | 60 | 0.616 | 4.824 | 34.621 | 4.264 | 30.627 |
| GeoLIP | Neurons\Layers | 5 | 10 | 15 | 20 | 25 |
| | 10 | 12.028 | 5.291 | 23.537 | 11.698 | 61.922 |
| | 20 | 1.632 | 0.765 | 0.992 | 38.569 | 11.774 |
| | 40 | 9.319 | 5.257 | 10.111 | 8.932 | 5.644 |
| | 60 | 1.270 | 6.872 | 45.046 | 4.562 | 22.026 |
| AAO | Neurons\Layers | 5 | 10 | 15 | 20 | 25 |
| | 10 | 10.300 | 13.010 | 102.766 | >10min | >10min |
| | 20 | 1.469 | 1.904 | 5.245 | | |
| | 40 | 9.006 | 11.023 | 56.829 | | |
| | 60 | 0.994 | 13.741 | 208.126 | | |

| | Neurons\Layers | 5 | 10 | 15 | 20 | 25 |
|---|---|---|---|---|---|---|
| **LipDiff** | 10 | 8.939 | 56.672 | 1448.136 | 10801.918 | 417370.969 |
| | 20 | 0.936 | 6.281 | 49.406 | 25191.828 | 29123.996 |
| | 40 | 10.958 | 52.335 | 888.581 | 5007.855 | 24095.277 |
| | 60 | 1.254 | 56.494 | 4461.550 | 2589.449 | 94250.664 |
| **GLipSDP** | Neurons\Layers | 5 | 10 | 15 | 20 | 25 |
| | 10 | 4.943 | 2.049 | 8.263 | 4.937 | 26.230 |
| | 20 | 0.635 | 0.305 | 0.415 | 14.109 | 4.502 |
| | 40 | 3.766 | 1.950 | 3.911 | 3.193 | 2.113 |
| | 60 | 0.447 | 2.446 | 16.205 | 1.615 | 7.947 |
| **EClipsE** | Neurons\Layers | 5 | 10 | 15 | 20 | 25 |
| | 10 | 6.935 | 4.599 | 28.314 | 16.643 | 239.018 |
| | 20 | 0.814 | 0.554 | 1.039 | 49.224 | 22.429 |
| | 40 | 4.941 | 3.559 | 9.166 | 10.572 | 8.977 |
| | 60 | 0.538 | 3.970 | 31.956 | 4.849 | 34.150 |
| **EClipsE-Fast** | Neurons\Layers | 5 | 10 | 15 | 20 | 25 |
| | 10 | 9.373 | 11.770 | 72.343 | 148.690 | 2234.919 |
| | 20 | 1.301 | 1.577 | 3.586 | 314.856 | 149.244 |
| | 40 | 8.685 | 8.755 | 33.455 | 50.758 | 75.140 |
| | 60 | 0.924 | 10.869 | 130.256 | 24.092 | 270.349 |
| **EClipsE-Gen-Local-Acc** | Neurons\Layers | 5 | 10 | 15 | 20 | 25 |
| | 10 | 5.926 | 4.617 | 35.025 | 16.637 | 0.004 |
| | 20 | 0.273 | 0.010 | 0.007 | 35.641 | 24.639 |
| | 40 | 4.572 | 2.069 | 6.211 | 0.170 | 1.322 |
| | 60 | 0.235 | 0.581 | 28.264 | 3.830 | 0.002 |
| **EClipsE-Gen-Local-Fast** | Neurons\Layers | 5 | 10 | 15 | 20 | 25 |
| | 10 | 7.184 | 5.058 | 27.996 | 35.167 | 307.104 |
| | 20 | 0.680 | 0.276 | 0.766 | 78.350 | 32.456 |
| | 40 | 5.428 | 3.672 | 10.755 | 9.752 | 10.066 |
| | 60 | 0.417 | 4.334 | 35.002 | 4.689 | 26.923 |

Table 1a: Lipschitz constant estimates (Continued)

| Table 1a: Lipschitz constant estimates (Continued) | | | | | | |
|---|---|---|---|---|---|---|
| | Neurons\Layers | 5 | 10 | 15 | 20 | 25 |
| EClipsE-Gen-Local-CF | 10 | 8.944 | 11.707 | 66.348 | 148.690 | 1736.535 |
| | 20 | 0.959 | 0.526 | 2.256 | 267.457 | 147.714 |
| | 40 | 7.887 | 7.796 | 31.828 | 42.772 | 62.424 |
| | 60 | 0.629 | 9.091 | 120.746 | 22.579 | 205.990 |

| Table 1b: Computation time (seconds) | | | | | | |
|---|---|---|---|---|---|---|
| | Neurons\Layers | 5 | 10 | 15 | 20 | 25 |
| SeqLip | 10 | 0.460 | 1.042 | 1.576 | 2.330 | 2.656 |
| | 20 | >10min | >10min | >10min | >10min | >10min |
| | 40 | | | | | |
| | 60 | | | | | |
| | Neurons\Layers | 5 | 10 | 15 | 20 | 25 |
| LipSDP-neuron | 10 | 1.176 | 1.097 | 1.040 | 1.307 | 2.247 |
| | 20 | 1.463 | 1.303 | 3.194 | 6.677 | 9.003 |
| | 40 | 7.396 | 7.229 | 21.173 | 51.123 | 97.725 |
| | 60 | 3.761 | 27.857 | 100.512 | 211.487 | 417.170 |
| | Neurons\Layers | 5 | 10 | 15 | 20 | 25 |
| LipSDP-layer | 10 | 12.368 | 1.468 | 1.229 | 1.992 | 2.540 |
| | 20 | 1.851 | 1.800 | 2.650 | 5.592 | 9.672 |
| | 40 | 1.973 | 6.753 | 13.897 | 32.194 | 58.683 |
| | 60 | 2.175 | 12.753 | 43.746 | 102.167 | 186.108 |
| | Neurons\Layers | 5 | 10 | 15 | 20 | 25 |
| GeoLIP | 10 | 0.487 | 0.867 | 1.968 | 2.107 | 3.290 |
| | 20 | 0.456 | 3.746 | 10.257 | 12.034 | 17.586 |
| | 40 | 2.230 | 23.392 | 52.746 | 58.190 | 96.382 |
| | 60 | 13.128 | 50.634 | 101.734 | 238.743 | 329.327 |
| | Neurons\Layers | 5 | 10 | 15 | 20 | 25 |
| AAO | 10 | 0.050 | 0.764 | 14.693 | 733.7 | >10min |
| | 20 | 0.033 | 2.071 | 23.885 | 1032.956 | |
| | 40 | 0.020 | 2.587 | 92.613 | 3348.118 | |
| | 60 | 0.051 | 5.319 | 200.619 | 8336.229 | |

| Table 1b: Computation time (seconds) (Continued) | | | | | |
|---|---|---|---|---|---|
| **LipDiff** | Neurons\Layers | 5 | 10 | 15 | 20 | 25 |
| | 10 | 3.728 | 3.163 | 4.373 | 22.935 | 18.885 |
| | 20 | 3.057 | 14.661 | 25.913 | 27.072 | 30.921 |
| | 40 | 5.419 | 27.389 | 42.046 | 51.114 | 76.681 |
| | 60 | 30.183 | 36.831 | 74.760 | 99.843 | 144.564 |
| **GLipSDP** | Neurons\Layers | 5 | 10 | 15 | 20 | 25 |
| | 10 | 0.158 | 0.127 | 0.284 | 0.304 | 0.508 |
| | 20 | 0.524 | 0.949 | 1.775 | 2.912 | 3.157 |
| | 40 | 6.016 | 16.463 | 29.642 | 46.906 | 49.518 |
| | 60 | 24.422 | 86.695 | 185.803 | 226.214 | 339.035 |
| **EClipsE** | Neurons\Layers | 5 | 10 | 15 | 20 | 25 |
| | 10 | 3.635 | 6.893 | 10.706 | 14.542 | 18.357 |
| | 20 | 3.603 | 7.674 | 11.762 | 16.931 | 21.438 |
| | 40 | 4.794 | 10.346 | 16.767 | 22.757 | 28.872 |
| | 60 | 6.405 | 15.514 | 26.935 | 36.545 | 43.624 |
| **EClipsE-Fast** | Neurons\Layers | 5 | 10 | 15 | 20 | 25 |
| | 10 | 0.002 | 0.003 | 0.002 | 0.004 | 0.003 |
| | 20 | 0.006 | 0.003 | 0.006 | 0.006 | 0.009 |
| | 40 | 0.008 | 0.015 | 0.017 | 0.030 | 0.035 |
| | 60 | 0.009 | 0.018 | 0.028 | 0.043 | 0.050 |
| **EClipsE-Gen-Local-Acc** | Neurons\Layers | 5 | 10 | 15 | 20 | 25 |
| | 10 | 3.605 | 10.674 | 15.059 | 26.974 | 15.522 |
| | 20 | 4.383 | 12.651 | 9.502 | 25.950 | 33.685 |
| | 40 | 7.289 | 14.403 | 24.690 | 32.200 | 44.148 |
| | 60 | 5.966 | 16.271 | 40.690 | 55.340 | 35.773 |
| **EClipsE-Gen-Local-Fast** | Neurons\Layers | 5 | 10 | 15 | 20 | 25 |
| | 10 | 3.143 | 6.484 | 10.065 | 13.545 | 17.497 |
| | 20 | 3.158 | 6.812 | 10.517 | 14.793 | 19.061 |
| | 40 | 3.614 | 8.268 | 13.356 | 17.745 | 22.745 |
| | 60 | 3.709 | 10.290 | 16.386 | 22.882 | 33.784 |

| Table 1b: Computation time (seconds) (Continued) | | | | | | |
|---|---|---|---|---|---|---|
| | Neurons\Layers | 5 | 10 | 15 | 20 | 25 |
| | 10 | 0.013 | 0.017 | 0.021 | 0.033 | 0.032 |
| EClipsE-Gen-Local-CF | 20 | 0.014 | 0.043 | 0.057 | 0.076 | 0.102 |
| | 40 | 0.039 | 0.131 | 0.175 | 0.280 | 0.332 |
| | 60 | 0.039 | 0.120 | 0.203 | 0.310 | 0.313 |

**Case 2 of Section 4.1:** We now present the complete results for large networks, where the number of layers is chosen from $\{30, 40, 50, 60, 70\}$, and number of neurons is chosen from $\{60, 80, 100, 120\}$.

| Table 2a: Lipschitz constant estimates | | | | | | |
|---|---|---|---|---|---|---|
| | Neurons\Layers | 30 | 40 | 50 | 60 | 70 |
| Trivial | 60 | 50682.053 | 306543.948 | $1.037 \times 10^9$ | $3.337 \times 10^{10}$ | $7.383 \times 10^{13}$ |
| | 80 | 138841.582 | 13156333.51 | $4.359 \times 10^{10}$ | $7.420 \times 10^{11}$ | $2.339 \times 10^{13}$ |
| | 100 | 28052.064 | 557532783.5 | $3.942 \times 10^{10}$ | $4.882 \times 10^{12}$ | $9.530 \times 10^{14}$ |
| | 120 | 152201.865 | $8.456 \times 10^9$ | $1.724 \times 10^{12}$ | $2.860 \times 10^{11}$ | $8.036 \times 10^{14}$ |
| LipSDP-neuron | Neurons\Layers | 30 | 40 | 50 | 60 | 70 |
| | 60 | 1.289 | 0.239 | 24.916 | 24.717 | 1760.327 |
| | 80 | 3.607 | 13.234 | 1259.817 | 802.203 | 808.099 |
| | 100 | 0.777 | 443.184 | 1085.570 | 5743.067 | 37482.662 |
| | 120 | 4.932 | 7655.233 | 48749.689 | 360.014 | 32328.464 |
| LipSDP-layer | Neurons\Layers | 30 | 40 | 50 | 60 | 70 |
| | 60 | 6.557 | 1.614 | 240.577 | 563.315 | 49382.204 |
| | 80 | 18.590 | 84.237 | 10936.840 | 13099.736 | 20612.931 |
| | 100 | 3.120 | 3003.417 | 8924.823 | 94284.796 | 701407.284 |
| | 120 | 16.737 | 46840.490 | 350361.282 | 3935.794 | 678055.572 |
| LipDiff | Neurons\Layers | 30 | 40 | 50 | 60 | 70 |
| | 60 | 46488.457 | 278692.438 | $1.530 \times 10^{12}$ | $8.720 \times 10^{16}$ | $1.980 \times 10^{27}$ |
| | 80 | 131847.219 | 11986194 | $1.060 \times 10^{17}$ | $4.000 \times 10^{20}$ | $1.860 \times 10^{24}$ |
| | 100 | 27995.227 | $1.205 \times 10^{11}$ | $2.470 \times 10^{17}$ | $4.500 \times 10^{23}$ | $4.340 \times 10^{29}$ |
| | 120 | 152062.031 | $1.812 \times 10^{15}$ | $1.210 \times 10^{23}$ | $2.260 \times 10^{20}$ | $6.340 \times 10^{29}$ |

| Table 2a: Lipschitz constant estimates (Continued) | | | | | | |
|---|---|---|---|---|---|---|
| **GLipSDP** | Neurons\Layers | 30 | 40 | 50 | 60 | 70 |
| | 60 | 0.577 | 0.078 | 5.321 | 3.410 | 200.713 |
| | 80 | 1.657 | >1h | >1h | >1h | >1h |
| | 100 | | | | | |
| | 120 | | | | | |
| **EClipsE** | Neurons\Layers | 30 | 40 | 50 | 60 | 70 |
| | 60 | 3.214 | 0.734 | 71.574 | 99.558 | 9117.857 |
| | 80 | 6.626 | 30.452 | 3184.903 | 2311.671 | 2686.612 |
| | 100 | 1.368 | 843.970 | 2436.388 | 12623.369 | 96935.788 |
| | 120 | 7.943 | 13707.418 | 91196.080 | 766.803 | 67675.878 |
| **EClipsE-Fast** | Neurons\Layers | 30 | 40 | 50 | 60 | 70 |
| | 60 | 40.357 | 22.005 | 4872.017 | 15334.703 | 2762962.864 |
| | 80 | 106.481 | 963.729 | 227516.592 | 354590.087 | 958744.578 |
| | 100 | 20.276 | 33545.214 | 185024.991 | 2392851.817 | $3.331 \times 10^{7}$ |
| | 120 | 101.125 | 473490.349 | 7784374.437 | 134759.991 | $3.058 \times 10^{7}$ |
| **EClipsE-Gen-Local-Acc** | Neurons\Layers | 30 | 40 | 50 | 60 | 70 |
| | 60 | $2.561 \times 10^{-6}$ | $3.626 \times 10^{-7}$ | $2.637 \times 10^{-8}$ | 55.177 | $1.317 \times 10^{-10}$ |
| | 80 | $6.019 \times 10^{-5}$ | $1.216 \times 10^{-6}$ | $2.970 \times 10^{-6}$ | 1253.190 | $3.797 \times 10^{-10}$ |
| | 100 | $1.746 \times 10^{-6}$ | $5.142 \times 10^{-5}$ | 1902.287 | 6496.172 | 11026.134 |
| | 120 | $6.309 \times 10^{-3}$ | 8306.746 | 37259.193 | $9.428 \times 10^{-8}$ | 4764.049 |
| **EClipsE-Gen-Local-Fast** | Neurons\Layers | 30 | 40 | 50 | 60 | 70 |
| | 60 | $1.083 \times 10^{-4}$ | $5.799 \times 10^{-5}$ | $5.501 \times 10^{-5}$ | 82.116 | $1.655 \times 10^{-5}$ |
| | 80 | 2.238 | $2.568 \times 10^{-5}$ | $2.343 \times 10^{-3}$ | 2272.633 | $1.348 \times 10^{-5}$ |
| | 100 | $9.010 \times 10^{-6}$ | 466.795 | 2421.066 | 13916.972 | 66516.583 |
| | 120 | 5.423 | 14380.517 | 76385.146 | 329.169 | 52766.628 |
| **EClipsE-Gen-Local-CF** | Neurons\Layers | 30 | 40 | 50 | 60 | 70 |
| | 60 | 28.629 | 20.747 | 4378.720 | 14540.358 | 2180308.626 |
| | 80 | 97.800 | 657.517 | 204800.834 | 334655.765 | 862880.611 |
| | 100 | 8.597 | 28903.208 | 177659.392 | 2219501.561 | $3.127 \times 10^{7}$ |
| | 120 | 96.418 | 453716.467 | 6822130.953 | 116706.452 | $2.842 \times 10^{7}$ |

| Table 2b: Computation times (seconds) | | | | | | |
|---|---|---|---|---|---|---|
| **LipSDP-neuron** | Neurons\Layers | 30 | 40 | 50 | 60 | 70 |
| | 60 | 109.070 | 152.040 | 149.638 | 727.651 | 172.854 |
| | 80 | 261.885 | 313.565 | 295.342 | 1267.712 | 422.347 |
| | 100 | 371.482 | 625.229 | 595.804 | 961.680 | 861.610 |
| | 120 | 692.723 | 1065.468 | 1179.913 | 1394.512 | 1603.990 |
| **LipSDP-layer** | Neurons\Layers | 30 | 40 | 50 | 60 | 70 |
| | 60 | 94.361 | 99.127 | 77.136 | 72.664 | 79.716 |
| | 80 | 89.473 | 187.850 | 151.267 | 111.975 | 136.632 |
| | 100 | 130.927 | 159.697 | 223.201 | 194.210 | 221.094 |
| | 120 | 201.007 | 222.053 | 313.597 | 274.748 | 411.751 |
| **LipDiff** | Neurons\Layers | 30 | 40 | 50 | 60 | 70 |
| | 60 | 179.812 | 469.768 | 716.592 | 1257.711 | 1897.470 |
| | 80 | 390.254 | 794.557 | 1625.415 | 2519.319 | 3523.333 |
| | 100 | 696.663 | 1617.582 | 3127.469 | 4006.535 | 5803.644 |
| | 120 | 1177.954 | 2459.011 | 3977.822 | 6283.629 | 9192.057 |
| **GLipSDP** | Neurons\Layers | 30 | 40 | 50 | 60 | 70 |
| | 60 | 126.677 | 454.171 | 633.618 | 569.647 | 562.881 |
| | 80 | >1h | 3902.751 | >1h | >1h | >1h |
| | 100 | >1h | | | | |
| | 120 | | | | | |
| **EClipsE** | Neurons\Layers | 30 | 40 | 50 | 60 | 70 |
| | 60 | 57.061 | 82.713 | 121.571 | 221.483 | 163.846 |
| | 80 | 103.047 | 164.912 | 207.042 | 378.171 | 309.558 |
| | 100 | 164.315 | 276.264 | 555.429 | 549.838 | 520.602 |
| | 120 | 275.917 | 472.231 | 893.199 | 923.932 | 770.500 |
| **EClipsE-Fast** | Neurons\Layers | 30 | 40 | 50 | 60 | 70 |
| | 60 | 0.099 | 0.076 | 0.110 | 0.127 | 0.130 |
| | 80 | 0.117 | 0.158 | 0.192 | 0.251 | 0.294 |
| | 100 | 0.204 | 0.222 | 0.292 | 0.314 | 0.382 |
| | 120 | 0.251 | 0.335 | 0.414 | 0.496 | 0.557 |

| Table 2b: Computation times (seconds) (Continued) | | | | | | |
|---|---|---|---|---|---|---|
| **EClipsE-Gen-Local-Acc** | Neurons\Layers | 30 | 40 | 50 | 60 | 70 |
| | 60 | 127.009 | 100.012 | 159.432 | 284.597 | 130.782 |
| | 80 | 442.548 | 117.682 | 263.666 | 504.913 | 229.356 |
| | 100 | 320.838 | 660.533 | 795.924 | 1108.792 | 989.315 |
| | 120 | 611.638 | 956.523 | 1109.196 | 927.641 | 1431.236 |
| **EClipsE-Gen-Local-Fast** | Neurons\Layers | 30 | 40 | 50 | 60 | 70 |
| | 60 | 34.299 | 48.792 | 125.322 | 157.239 | 95.297 |
| | 80 | 54.192 | 79.973 | 157.899 | 211.596 | 237.253 |
| | 100 | 54.362 | 187.273 | 215.903 | 276.621 | 323.633 |
| | 120 | 92.174 | 228.984 | 181.819 | 368.509 | 352.321 |
| **EClipsE-Gen-Local-CF** | Neurons\Layers | 30 | 40 | 50 | 60 | 70 |
| | 60 | 0.398 | 0.535 | 0.809 | 0.839 | 1.048 |
| | 80 | 0.676 | 0.970 | 1.178 | 1.555 | 2.047 |
| | 100 | 0.894 | 1.225 | 1.700 | 2.019 | 2.632 |
| | 120 | 1.187 | 1.834 | 2.257 | 3.117 | 3.406 |

**Results for Section 4.2:**

| Lipschitz Estimates on FNN with 5 Layers and 128 Neurons for Different Input Radii | | | | | | | |
|---|---|---|---|---|---|---|---|
| Trivial bound: 66.975, `autodiff`: 0.235 | | | | | | | |
| Algorithm | $r = 5$ | $r = 1$ | $r = 1/5$ | $r = 1/5^2$ | $r = 1/5^3$ | $r = 1/5^4$ | $r = 1/5^5$ |
| EClipsE-Gen-Local-Acc | 12.206 | 9.175 | 1.612 | 0.946 | 0.304 | 0.235 | 0.235 |
| EClipsE-Gen-Local-Fast | 13.798 | 12.465 | 6.855 | 3.290 | 0.824 | 0.235 | 0.235 |
| EClipsE-Gen-Local-CF | 23.028 | 20.702 | 15.210 | 11.510 | 10.875 | 10.294 | 10.271 |

| Lipschitz Estimates on FNN with 5 Layers and 128 Neurons for Different Input Radii | | | | | | | |
|---|---|---|---|---|---|---|---|
| Trivial bound: $3.070 \times 10^{10}$, `autodiff`: $1.539 \times 10^{-3}$ | | | | | | | |
| Algorithm | $r = 5$ | $r = 1$ | $r = 1/5$ | $r = 1/5^2$ | $r = 1/5^3$ | $r = 1/5^4$ | $r = 1/5^5$ |
| EClipsE-Gen-Local-Acc | $1.804 \times 10^6$ | $9.382 \times 10^5$ | $8.471 \times 10^4$ | $9.031 \times 10^{-3}$ | $2.113 \times 10^{-3}$ | $2.209 \times 10^{-3}$ | $1.539 \times 10^{-3}$ |
| EClipsE-Gen-Local-Fast | $2.201 \times 10^6$ | $1.849 \times 10^6$ | $6.603 \times 10^5$ | $6.754 \times 10^4$ | $6.934 \times 10^{-3}$ | $2.843 \times 10^{-3}$ | $1.539 \times 10^{-3}$ |
| EClipsE-Gen-Local-CF | $2.062 \times 10^7$ | $1.811 \times 10^7$ | $1.197 \times 10^7$ | $6.010 \times 10^6$ | $2.672 \times 10^6$ | $1.296 \times 10^6$ | $6.008 \times 10^5$ |

| Lipschitz Estimates on FNN with 60 Layers and 128 Neurons for Different Input Radii | | | | | | | |
|---|---|---|---|---|---|---|---|
| Trivial bound: $4.324 \times 10^{21}$, `autodiff`: $6.324 \times 10^{-6}$ | | | | | | | |
| Algorithm | $r = 5$ | $r = 1$ | $r = 1/5$ | $r = 1/5^2$ | $r = 1/5^3$ | $r = 1/5^4$ | $r = 1/5^5$ |
| ECLipsE-Gen-Local-Acc | $1.68 \times 10^{14}$ | $1.37 \times 10^{14}$ | $9.29 \times 10^{11}$ | $2.91 \times 10^{-5}$ | $1.14 \times 10^{-5}$ | $6.32 \times 10^{-6}$ | $6.32 \times 10^{-6}$ |
| ECLipsE-Gen-Local-Fast | $1.81 \times 10^{14}$ | $1.70 \times 10^{14}$ | $7.87 \times 10^{13}$ | $6.92 \times 10^{12}$ | $2.00 \times 10^{-5}$ | $6.32 \times 10^{-6}$ | $6.32 \times 10^{-6}$ |
| ECLipsE-Gen-Local-CF | $1.33 \times 10^{15}$ | $1.29 \times 10^{15}$ | $9.86 \times 10^{14}$ | $5.78 \times 10^{14}$ | $2.62 \times 10^{14}$ | $1.31 \times 10^{14}$ | $5.74 \times 10^{13}$ |

**Robust training in Section 4.3:** The complete results are as follows.

| Local Lipschitz Estimates on Baseline MNIST Model for Different Input Radii | | | | | | | | |
|---|---|---|---|---|---|---|---|---|
| Sample | $r = 1/2$ | $r = 1/2^2$ | $r = 1/2^3$ | $r = 1/2^4$ | $r = 1/2^5$ | $r = 1/2^6$ | $r = 1/2^7$ | $r = 1/2^8$ |
| 1 | 78.71098 | 78.70734 | 78.61906 | 74.36818 | 58.13149 | 44.6177 | 26.05712 | 16.33641 |
| 2 | 78.71098 | 78.70682 | 78.60065 | 71.53018 | 57.27216 | 42.98288 | 28.55629 | 18.88016 |
| 3 | 78.71098 | 78.70626 | 78.59112 | 69.64929 | 55.56146 | 43.45022 | 29.32234 | 21.11356 |
| 4 | 78.71098 | 78.70635 | 78.59286 | 71.67161 | 54.69806 | 41.45828 | 26.19779 | 17.52604 |
| 5 | 78.71099 | 78.70774 | 78.6268 | 73.33843 | 58.15701 | 42.77248 | 22.93037 | 14.57802 |

(Continued)

| | | | | | | | |
|---|---|---|---|---|---|---|---|
| 6 | 78.71098 | 78.70589 | 78.58708 | 69.20899 | 55.67301 | 41.0769 | 26.20351 | 16.5844 |
| 7 | 78.71098 | 78.70628 | 78.59743 | 71.60393 | 56.94831 | 40.92923 | 25.57324 | 17.9801 |
| 8 | 78.71099 | 78.70736 | 78.61574 | 71.19789 | 58.25083 | 40.83705 | 25.64 | 16.33402 |
| 9 | 78.71098 | 78.70682 | 78.60921 | 71.91748 | 58.53037 | 41.81912 | 23.86733 | 16.24717 |
| 10 | 78.71099 | 78.70749 | 78.62238 | 71.71447 | 58.31806 | 37.37913 | 21.85074 | 15.73135 |
| 11 | 78.71098 | 78.70659 | 78.53775 | 72.1146 | 56.06148 | 42.75571 | 28.95411 | 18.57977 |
| 12 | 78.71099 | 78.7079 | 78.6297 | 72.59098 | 56.81893 | 40.59961 | 25.43528 | 18.3823 |
| 13 | 78.71098 | 78.70692 | 78.60969 | 72.93557 | 55.63917 | 40.56271 | 25.40254 | 17.13129 |
| 14 | 78.71099 | 78.70701 | 78.60111 | 71.52446 | 56.58831 | 39.78379 | 24.79018 | 18.5912 |
| 15 | 78.71098 | 78.70657 | 78.59976 | 70.00147 | 56.8946 | 38.23369 | 25.27076 | 16.24316 |
| 16 | 78.71098 | 78.70637 | 78.60351 | 71.28439 | 58.42006 | 38.85839 | 23.427 | 15.95628 |
| 17 | 78.71098 | 78.70606 | 78.59739 | 71.30552 | 55.61067 | 43.27397 | 28.03724 | 18.91537 |
| 18 | 78.71098 | 78.70686 | 78.60872 | 73.12516 | 58.85767 | 44.68016 | 31.1804 | 20.87262 |
| 19 | 78.71099 | 78.708 | 78.63192 | 71.6684 | 54.38538 | 38.4346 | 22.14221 | 14.83256 |
| 20 | 78.71098 | 78.70636 | 78.59858 | 70.45079 | 54.85386 | 40.67774 | 27.49389 | 18.38262 |

| Local Lipschitz Estimates on Robustly Trained MNIST Model for Different Input Radii | | | | | | | | |
|---|---|---|---|---|---|---|---|---|
| Sample | $r = 1/2$ | $r = 1/2^2$ | $r = 1/2^3$ | $r = 1/2^4$ | $r = 1/2^5$ | $r = 1/2^6$ | $r = 1/2^7$ | $r = 1/2^8$ |
| 1 | 59.95561 | 59.76889 | 58.32222 | 55.58214 | 47.47068 | 34.9161 | 14.29093 | 6.055722 |
| 2 | 59.95561 | 59.94913 | 58.33661 | 55.45571 | 49.27195 | 39.23655 | 23.30358 | 9.705545 |
| 3 | 59.95561 | 59.77215 | 58.33239 | 54.83087 | 51.01784 | 38.43811 | 20.2736 | 7.281093 |
| 4 | 59.95561 | 59.73319 | 58.24574 | 55.33866 | 49.84949 | 38.30965 | 17.83903 | 8.109753 |
| 5 | 59.95561 | 59.94582 | 58.32322 | 55.47847 | 48.87655 | 34.17816 | 13.47338 | 7.282262 |
| 6 | 59.95561 | 59.68513 | 57.9313 | 55.21375 | 50.24844 | 40.44054 | 18.9907 | 7.287764 |

(Continued)

| | | | | | | | |
|---|---|---|---|---|---|---|---|
| 7 | 59.95561 | 59.71394 | 58.32489 | 55.74624 | 49.53914 | 34.69735 | 16.43406 | 7.998252 |
| 8 | 59.95561 | 59.94728 | 58.341 | 55.34873 | 50.55051 | 38.52287 | 18.44644 | 7.479875 |
| 9 | 59.95561 | 59.67834 | 58.32869 | 55.74314 | 48.33398 | 34.24256 | 15.96027 | 7.561291 |
| 10 | 59.95561 | 59.94594 | 58.32607 | 55.66811 | 49.44189 | 33.5797 | 17.50552 | 6.006573 |
| 11 | 59.95561 | 59.94523 | 58.33124 | 55.08831 | 49.56107 | 37.93311 | 22.16714 | 8.042922 |
| 12 | 59.95561 | 59.59303 | 58.32403 | 55.49748 | 52.38679 | 32.91855 | 15.00071 | 8.27143 |
| 13 | 59.95561 | 59.88001 | 58.33205 | 56.58481 | 52.77126 | 35.00035 | 18.14338 | 7.062218 |
| 14 | 59.95561 | 59.80707 | 58.33666 | 55.47049 | 49.36892 | 35.28665 | 12.42868 | 5.485892 |
| 15 | 59.95561 | 59.45832 | 58.16557 | 55.67141 | 47.28045 | 34.83791 | 16.80016 | 7.039371 |
| 16 | 59.95561 | 59.7419 | 58.32438 | 55.95442 | 50.13239 | 36.0773 | 17.53038 | 8.491852 |
| 17 | 59.95561 | 59.68881 | 58.32994 | 54.71305 | 47.55917 | 34.46926 | 15.94367 | 7.291772 |
| 18 | 59.95561 | 59.9465 | 58.32893 | 55.67932 | 51.16427 | 32.14587 | 13.33776 | 7.268891 |
| 19 | 59.95561 | 59.49008 | 58.02768 | 54.94707 | 49.34491 | 31.38572 | 16.15183 | 6.080519 |
| 20 | 59.95561 | 59.71931 | 58.11977 | 55.95374 | 51.04859 | 41.01665 | 19.68835 | 7.429283 |

| Failure Rate of Models on MNIST Under PGD Attacks within Given Range (%) | | | | | | | | |
|---|---|---|---|---|---|---|---|---|
| Model | $r = 1/2$ | $r = 1/2^2$ | $r = 1/2^3$ | $r = 1/2^4$ | $r = 1/2^5$ | $r = 1/2^6$ | $r = 1/2^7$ | $r = 1/2^8$ |
| Baseline | 9.65% | 4.6% | 2.94% | 2.48% | 2.11% | 2.04% | 1.99% | 1.95% |
| Robustly Trained | 4.25% | 2.7% | 2.26% | 1.97% | 1.94% | 1.9% | 1.88% | 1.87% |

### A.3.5 Detailed Results for Robustness Certificates in Section 4.3

| Sample | $r_{0.5}$ | $r_{0.25}$ | $r_{0.125}$ | $r_{0.0625}$ | $r_{0.03125}$ | $r_{0.015625}$ | $r_{0.0078125}$ | $r_{0.0039062}$ | $r_{\text{cert}}$ | $r_{\text{triv}}$ |
|---|---|---|---|---|---|---|---|---|---|---|
| 1 | 0.011458 | 0.011494 | 0.011779 | 0.01236 | 0.014472 | 0.015625 | 0.0078125 | 0.0039062 | 0.015625 | 0.0029057 |
| 2 | 0.0045758 | 0.0045763 | 0.0047028 | 0.0049471 | 0.005568 | 0.0069921 | 0.0078125 | 0.0039062 | 0.0078125 | 0.0011604 |
| 3 | 0.010478 | 0.01051 | 0.010769 | 0.011457 | 0.012313 | 0.015625 | 0.0078125 | 0.0039062 | 0.015625 | 0.002657 |
| 4 | 0.0059098 | 0.0059318 | 0.0060833 | 0.0064029 | 0.0071079 | 0.009249 | 0.0078125 | 0.0039062 | 0.009249 | 0.0014986 |
| 5 | 0.013115 | 0.013117 | 0.013482 | 0.014173 | 0.016087 | 0.015625 | 0.0078125 | 0.0039062 | 0.016087 | 0.0033257 |
| 6 | 0.0067095 | 0.0067399 | 0.0069439 | 0.0072857 | 0.0080057 | 0.0099472 | 0.0078125 | 0.0039062 | 0.0099472 | 0.0017014 |
| 7 | 0.0045022 | 0.0045204 | 0.0046281 | 0.0048422 | 0.0054489 | 0.0077796 | 0.0078125 | 0.0039062 | 0.0078125 | 0.0011417 |
| 8 | 0.00016575 | 0.00016578 | 0.00017034 | 0.00017955 | 0.00019659 | 0.00025797 | 0.00053874 | 0.0013286 | 0.0013286 | 4.20e-05 |
| 9 | 0.0096509 | 0.0096957 | 0.0099201 | 0.01038 | 0.011971 | 0.015625 | 0.0078125 | 0.0039062 | 0.015625 | 0.0024473 |
| 10 | 0.017381 | 0.017384 | 0.017867 | 0.01872 | 0.021077 | 0.015625 | 0.0078125 | 0.0039062 | 0.021077 | 0.0044075 |
| 11 | 0.0023155 | 0.0023159 | 0.00238 | 0.0025201 | 0.0028012 | 0.0036598 | 0.0062628 | 0.0039062 | 0.0062628 | 0.00058718 |
| 12 | 0.0032582 | 0.003278 | 0.0033494 | 0.0035199 | 0.003729 | 0.0059343 | 0.0078125 | 0.0039062 | 0.0078125 | 0.00082623 |
| 13 | 0.0071663 | 0.0071753 | 0.0073657 | 0.0075932 | 0.0081419 | 0.012276 | 0.0078125 | 0.0039062 | 0.012276 | 0.0018172 |
| 14 | 0.010999 | 0.011026 | 0.011304 | 0.011888 | 0.013358 | 0.015625 | 0.0078125 | 0.0039062 | 0.015625 | 0.0027892 |
| 15 | 0.008975 | 0.0090501 | 0.0092512 | 0.0096657 | 0.011381 | 0.015446 | 0.0078125 | 0.0039062 | 0.015446 | 0.0022759 |
| 16 | 0.010617 | 0.010655 | 0.010914 | 0.011377 | 0.012698 | 0.015625 | 0.0078125 | 0.0039062 | 0.015625 | 0.0026924 |
| 17 | 0.0056556 | 0.0056809 | 0.0058132 | 0.0061975 | 0.0071297 | 0.0098373 | 0.0078125 | 0.0039062 | 0.0098373 | 0.0014342 |
| 18 | 0.0083519 | 0.0083532 | 0.0085849 | 0.0089934 | 0.009787 | 0.015577 | 0.0078125 | 0.0039062 | 0.015577 | 0.0021179 |
| 19 | 0.0065651 | 0.0066164 | 0.0067832 | 0.0071635 | 0.0079768 | 0.012541 | 0.0078125 | 0.0039062 | 0.012541 | 0.0016648 |
| 20 | 0.0091699 | 0.0092062 | 0.0094596 | 0.0098258 | 0.01077 | 0.013404 | 0.0078125 | 0.0039062 | 0.013404 | 0.0023254 |

|  |  |
|---|---|
| Mean | 0.01222977  0.001990887 |
| Ratio mean($r_{\text{cert}}$)/mean($r_{\text{triv}}$) | 6.142874701 |

Table 1: Certified radii across $\epsilon \in \{1/2, 1/4, \ldots, 1/256\}$ for 20 test points. Columns $r_\epsilon$ report $r_{\text{cert}}(x; \epsilon)$. Column $r_{\text{cert}}$ reports the least conservative (largest) certified radius obtained by sweeping over the same set of $\epsilon$ values. Column $r_{\text{triv}}$ reports the certified radius using the global trivial Lipschitz bound.

### A.3.6 Additional Plots for Section 4.2

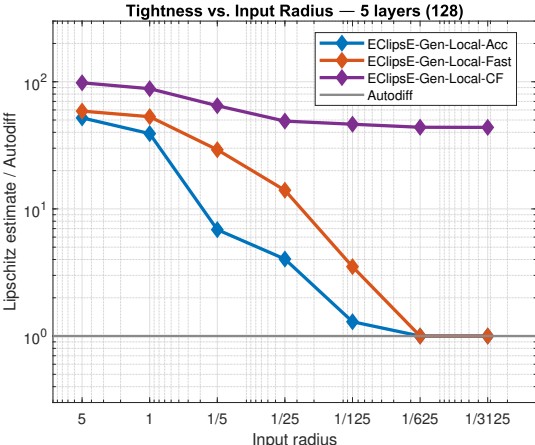

Figure 7: Lipschitz estimates are normalized to `autodiff` value at $z_c$ (0.2347). Naive bound: 66.9754.

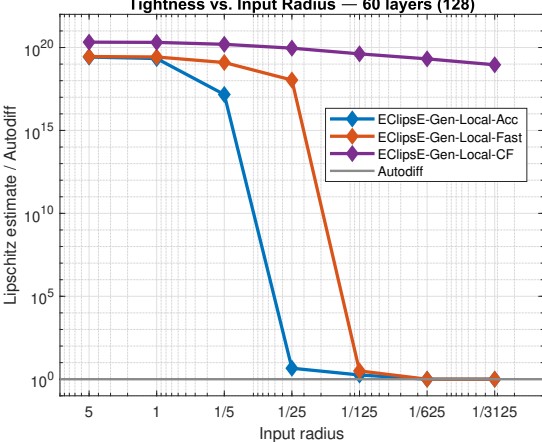

Figure 8: Lipschitz estimates normalized to `autodiff` value at $z_c$ ($6.3224 \times 10^{-6}$). Naive bound: $4.3244 \times 10^{21}$.

### A.4 Slack introduced by the CF slope adjustment

In ECLipsE-Gen-Local-CF, at each stage $i$, after obtaining refined per-neuron slope bounds $[\boldsymbol{\alpha}^i, \boldsymbol{\beta}^i]$ from local propagation, we replace them with a valid superset $[\boldsymbol{\alpha}^{i,adj}, \boldsymbol{\beta}^{i,adj}]$ that satisfies $\boldsymbol{\alpha}^{i,adj} \odot \boldsymbol{\beta}^{i,adj} = 0$ to enable a closed-form update, $i \in \mathbb{Z}_{N-1}$. Under $\Lambda_i = \lambda_i I$, the term removed from $\tilde{X}_{i-1}$ in (14) is

$$\Delta_{slack}^i = \lambda_i W_i^\top D_{\boldsymbol{\alpha}^i} D_{\boldsymbol{\beta}^i} W_i. \tag{56}$$

Therefore, the induced slack is small when the refined lower slope bounds are already close to zero ($D_{\boldsymbol{\alpha}^i} D_{\boldsymbol{\beta}^i}$ is small). The slack can be substantial when both $\boldsymbol{\alpha}^i$ and $\boldsymbol{\beta}^i$ are non-negligible and the scale or alignment of $W_i$ amplifies $W_i^\top D_{\boldsymbol{\alpha}^i} D_{\boldsymbol{\beta}^i} W_i$.

