# OpenReview forum: "ECLipsE-Gen-Local: Efficient Compositional Local Lipschitz Estimates for Deep Neural Networks"
_TMLR — Accepted by TMLR_

### Review · Reviewer_tQzA · 2025-12-19

**Summary Of Contributions:**

This paper presents ECLipsE-Gen-Local, a method for efficiently estimating certified local Lipschitz constants for deep neural networks. It extends the prior ECLipsE framework to handle activation functions with heterogeneous slope bounds (like LeakyReLU, ELU) and, most importantly, incorporates local information about the input region to produce much tighter bounds than global methods, especially for small regions. The method decomposes a large, intractable Semidefinite Program (SDP) into a sequence of small sub-problems, achieving computational complexity linear in network depth. The authors propose three algorithmic variants (Accurate, Fast, Closed-Form) offering a trade-off between bound tightness and computation speed. Extensive experiments show significant speedups over SDP benchmarks and demonstrate that the local bounds can approach the exact Jacobian norm for very small input regions.

-----------
Pros:
--------------
-Strong Technical Contribution: The generalization to heterogeneous slope bounds and the systematic integration of local information are clear and valuable advances over the state-of-the-art (ECLipsE, LipSDP).
-Solid Theoretical Foundation: The paper provides rigorous proofs for the validity, feasibility, and tightness of the proposed bounds, building on a well-established SDP certification framework.
-Comprehensive and Convincing Experiments: The evaluation is thorough, covering scalability (depth/width), tightness vs. input region size, and a practical link to adversarial robustness. The comparison against 13 different benchmarks is impressive.
-Practical Engineering: The discussion of numerical stability safeguards, handling of affine layers, and the provision of three algorithmic variants show attention to real-world implementation challenges.
-Clear Presentation: The paper is well-structured and the methodology is explained logically.

--------------------------------------
Cons / Points for Clarification
-------------------------------------
-Practical meaning of “local” bounds needs clearer discussion.
The advantage of the local variants appears strongest when the input region is very small, as shown clearly in Experiment Set 2, where the bounds approach the Jacobian norm. However, in the scalability experiments (Set 1), which use much larger input regions, the local methods achieve accuracy that is similar to the global variants, while incurring additional computational cost. This suggests a nontrivial trade-off between region size and practical benefit. The paper does not explicitly discuss this trade-off, which makes it harder to understand in which regimes the local approach provides a clear advantage.
A more explicit discussion would be helpful: for what region sizes does locality meaningfully improve the bound, and when does it effectively reduce to a global estimate?

-Limited guidance on choosing among Acc, Fast, and CF variants.
The paper introduces three algorithmic variants (Acc, Fast, and CF), but offers little guidance on how a practitioner should choose between them. While performance differences are visible in the experiments, the paper does not summarize typical tightness gaps or computational trade-offs in a direct way.
For example, it would be useful to explicitly state scenarios such as: CF is preferred when speed is critical and moderate looseness is acceptable; Acc should be used when tightness on small regions is the main goal; Fast offers a compromise. Without such guidance, the practical decision-making burden is left entirely to the reader.

-Effect of the closed-form (CF) slope adjustment is not analyzed.
The CF variant relies on adjusting the refined slope bounds to satisfy the condition a*b=0. While this adjustment is stated to preserve validity, the paper does not quantify or analyze the loss in tightness introduced by this constraint. This omission is important, because it likely explains why CF bounds in Fig. 5 remain noticeably looser than those produced by Acc and Fast, even on small regions.
A clearer discussion would help here: how much tightness is typically lost due to this adjustment, and is this loss expected to grow with depth or region size? Understanding this cost would clarify the real price paid for the closed-form speedup.

**Additional Comments:**

Overall, the paper is technically solid and addresses an important problem in robustness certification. Clarifying the practical regimes where locality matters most and providing more guidance on algorithm selection would further improve its usefulness to practitioners and reviewers.

**Audience:**

Yes

**Audience Explanation:**

Yes. Researchers working on certified robustness, neural network verification, and safety-critical learning would be interested, as the paper proposes tighter and more scalable Lipschitz bounds with formal guarantees. The local perspective is especially relevant for those studying robustness near specific inputs rather than worst-case global behavior.

**Broader Impact Concerns:**

No major ethical concerns. The work focuses on theoretical methods for certifying robustness and is primarily relevant to safety-critical and verification settings; potential misuse is limited and not unique to this approach.

**Claims And Evidence:**

Yes

**Claims Explanation:**

The claim of linear scalability with depth is convincingly demonstrated by the clear, linear growth curves in runtime vs. depth plots (Figs. 1b, 3b), significantly outperforming non-compositional SDP methods.

The claim of tighter local bounds is compellingly validated by Experiment Set 2 (Fig. 5), which shows local estimates monotonically tightening by many orders of magnitude as the input region shrinks, eventually approaching the exact Jacobian norm—a direct and clear measure of optimal tightness.

The claim of practical utility is supported by the robustness correlation experiment (Fig. 6), which shows a clear alignment between lower certified Lipschitz estimates and lower empirical adversarial failure rates.

The only nuance is that the evidence for the advantage of "local" is split across experiments. The dramatic tightness gain is proven for small regions, while the scalability experiments use a larger region where the benefit is less pronounced, a point that could be stated more clearly. Overall, the evidence is accurate, comprehensive, and convincing.

**Requested Changes:**

* Clarify when and why local Lipschitz bounds provide a clear advantage over global ones, especially as the input region grows.
* Provide practical guidance on choosing between Acc, Fast, and CF, including typical tightness–cost trade-offs.
* Quantify and discuss the tightness loss introduced by the closed-form (CF) slope adjustment.

---

> ### Author Response · Authors · 2026-01-02
> **Author's response to the review**
>
> We thank the reviewer for the insightful comments emphasizing practical usage and trade-offs. We have added guidance on when locality is most beneficial, how to choose among  the Acc, Fast, and CF variants, and   included a discussion of the conservatism introduced by the CF slope adjustment as summarized in the General Response. We respond to each point below.
> >1. (Cons) Practical meaning of “local” bounds needs clearer discussion... (Claim accuracy) The only
> nuance is that the evidence for the advantage of ”local” is split across experiments... (Requested Change 1)  Clarify when and why local Lipschitz bounds provide a clear advantage over global ones, especially as the input region grows.
>
> We thank the reviewer for this insightful observation. We agree that there is a practical trade-off: as the input region grows large, the local Lipschitz constant naturally approaches the global constant, reducing the benefit of the additional computational cost even though the local algorithm is by design of the same scalability. Conversely, when the input region is small enough to capture a landscape distinct from the global worst-case, the gain in tightness is substantial. In practice, we suggest using the global variant when a domain-wide certificate is required or when the region of interest is large. The local variant is best suited for specific neighborhoods where tightness is critical, such as robustness certification around test inputs. \
> To make this clear for readers, we have added a short remark explicitly discussing this trade-off and providing guidance on when ***ECLipsE-Gen-Local*** is most beneficial. \
> ***Remark to add at the end of Section 4.2.*** \
> ***Remark (Local vs. Global).*** The practical value of local estimates depends on the scale of the input region. For large regions or when model behavior is expected to be homogeneous over the full domain, the local bound is close to the global constant. In such cases, scalable global estimation algorithms like ECLipsE (if applicable) are preferable due to its lower computational cost. However, when the region is small to moderate, such as in robustness certification around a specific input (as will be demonstrated in Sec. 4.3), the proposed local method exploits the local landscape to yield significantly tighter bounds, justifying the additional computation cost.
>
> >2. (Cons) Limited guidance on choosing among Acc, Fast, and CF variants... (Requested Changes 2) Provide practical guidance on choosing between Acc, Fast, and CF, including typical tightness–cost trade-offs.
>
> We agree that summarizing the trade-offs into a practical decision-making framework is valuable for users. We have added a remark that outlines the tightness-cost trade-off and also recommend a progressive workflow as follows.
> ***Remark to add at the end of Section 3.3.*** \
> ***Remark (Guidance on Variant Selection).*** The selection of variants within the ECLipsE-Gen series depends on the desired trade-off between tightness and computational cost: we choose CF (if applicable) when speed is critical (e.g. online tasks), Acc when precision is paramount (e.g., safety certification), and Fast for a balanced tradeoff. Practically, we recommend starting with CF for online tasks due to its negligible cost. If the resulting bound is too conservative, one can upgrade to Fast or Acc. Additionally, the sequential nature of the algorithm allows users to estimate the total runtime after processing the first layer (as the computation time scales linearly with respect to the depth of NN), and accordingly choose the most accurate variant that fits their time budget.
>
> >3. (Cons) Effect of the closed-form (CF) slope adjustment is not analyzed... (Requested Changes 3) Quantify and discuss the tightness loss introduced by the closed-form (CF) slope adjustment.
>
> We thank the reviewer for pointing this out. It is indeed correct that to yield a closed-form solution, some relaxation is unavoidable.
> While quantifying the exact loss in tightness for the final Lipschitz constant is analytically intractable due to the complexity of the underlying SDPs and the iteratively passing on of the messenger matrices, we can characterize the slack introduced by the specific slope bound adjustment required for the CF variant. The detailed analysis that will be added to appendix and the clarifications addressing the conservatism in the main text are provided in **General Response (C)**. Note that we replace $\otimes$ with $\odot$ to denote elementwise multiplication, aligning with standard notation.

---

### Review · Reviewer_bFAb · 2025-12-21

**Summary Of Contributions:**

This paper studies the problem of efficiently computing tight and certified local Lipschitz constant upper bounds for deep feedforward neural networks. Building upon SDP-based Lipschitz certificates (notably LipSDP) and recent compositional decompositions (ECLipsE), the authors propose a generalized framework—ECLipsE-Gen—that supports heterogeneous neuron-wise slope bounds, arbitrary input–output index selection, and subnetwork-level certification. More importantly, the paper introduces ECLipsE-Gen-Local, which explicitly incorporates local input region information to iteratively refine neuron-wise slope bounds, yielding substantially tighter local Lipschitz estimates with provable guarantees.

Strengths:
The paper is technically solid and presents a well-motivated combination of generalization, locality, and scalability. The proposed framework substantially expands the applicability of certified Lipschitz estimation and is supported by both theoretical analysis and empirical validation.

Weaknesses:
The presentation is mathematically dense, and the intuition behind the compositional and local refinement procedures could be explained more clearly. In addition, the distinction between the proposed method and prior ECLipsE-based approaches could be made more explicit.

**Audience:**

Yes

**Audience Explanation:**

Compared to prior work, this paper goes beyond incremental improvements: it significantly expands the expressive power of Lipschitz certification frameworks while maintaining computational feasibility. The ability to compute local, neuron-level, and subnetwork-specific Lipschitz bounds is particularly appealing and aligns well with TMLR’s emphasis on foundational yet practically meaningful advances.

**Claims And Evidence:**

Yes

**Claims Explanation:**

The above assessment focuses on the paper’s main technical contributions while also reflecting areas where the work could be improved. Although the proposed framework extends prior SDP-based and compositional approaches in meaningful ways, several of the contributions rely heavily on existing ideas (e.g., ECLipsE-style decomposition and slope-bounding techniques), which makes it important to clearly delineate what is fundamentally new versus what is a generalization or refinement. Additionally, while the theoretical development is rigorous, the presentation places a substantial burden on the reader, and some algorithmic choices and intuitions are not immediately transparent. These considerations motivate a balanced summary that acknowledges the technical depth and practical relevance of the work while also noting limitations in clarity and positioning that may affect accessibility and perceived novelty.

**Requested Changes:**

1. Add illustrative examples: The paper would benefit from including concrete examples demonstrating how the proposed method operates in practice. This would help clarify the theoretical developments and make the contributions more tangible to the reader.

2. Explain the role of the estimated Lipschitz constant: The authors should explicitly discuss how estimating the Lipschitz constant contributes to robustness certificates or enhances resilience against adversarial attacks. A clear connection between the theoretical results and practical robustness implications would strengthen the paper.

3. Empirical validation: If possible, include experiments demonstrating the effectiveness of the proposed method in real-world applications, particularly in the context of adversarial robustness. Even small-scale empirical results illustrating the method’s utility would significantly improve the paper’s impact and persuasiveness.

---

> ### Author Response · Authors · 2026-01-02
> **Author's response to the review**
>
> We appreciate the reviewer for the thoughtful feedback on presentation and practical relevance. In response, we have added clarifications to present the intuition behind our mathematical results and distinguish our method from prior ECLipsE-based approaches. We also include a  new robustness-certification experiment as summarized in the General Response. We address the specific comments below.
> >1. Weakness on Dense Mathematical Exposition
>
> We acknowledge that the presentation involves dense notation. This mathematical complexity is unfortunately necessary to establish a general framework for global and local Lipschitz estimation. Indeed, the theoretical machinery developed in this paper forms the core of our contribution. Specifically, to capture the essence of local information propagation, our framework must handle heterogeneous slope bounds at the neuron level. This necessitates a more involved mathematical structure than standard global methods. However, we believe this rigorous general framework establishes a necessary foundation that will facilitate future research. For readers primarily interested in implementation, Algorithm 1 provides a self-contained, step-by-step procedure that can be implemented directly without requiring the full theoretical derivation. We will also release the code publicly upon acceptance. \
> With that being said, we offer the following comment on the intuition behind our algorithms. "Intuitively, while messenger matrices propagate coupling information, local information is simultaneously propagated as we reach each new layer by calculating the reachable local region and encoding this information into refined slope bounds". (**Text addition in Section 3.3 (ECLipsE-Gen-Local)**).
>
> >2. Distinction from ECLipsE-based approaches:
>
> Regarding the distinction, we would like to refer readers to Section 2.2: "Unlike prior ECLipsE-based approaches which are restricted to specific activation functions and estimates over the entire domain, our method utilizes local information to handle heterogeneous slope bounds for general activation".  This distinction is also summarized in Contribution 2 in the Introduction. Broadly, this allows us to adopt heterogeneous slope bounds across layers, iteratively refine them and propagate local information across layers to obtain provably valid local Lipschitz upper bounds, none of which are achievable through standard ECLipsE-based approaches without the theoretical and algorithmic developments in this work.
>
> >3. Requested changes 1, 2, \& 3
>
> Thank you for these considerate suggestions aimed at improving clarity, practical relevance, and empirical persuasiveness. To provide a tangible illustration of the method’s utility, we have added a new explicitly robustness-oriented experiment in Sec. 4.3. The details of the added experiment and the results included in the new Appendix A.3 are provided in **General Response (B)**. \
> While we respectfully refer readers to the first paragraph of the Introduction for the applications already discussed, to further clarify, we add a remark after the contributions list as detailed in **General Response (A)** listing specific downstream tasks that will benefit from our work. To keep the paper focused on the core theoretical and algorithmic contributions of developing a scalable and efficient estimation framework, we defer further application-specific implementations of this highly general framework and its extensions to downstream pipelines to future work.

---

### Review · Reviewer_24gS · 2025-12-22

**Summary Of Contributions:**

The paper addresses the problem of estimating Lipschitz constants to certify the robustness and safety of neural networks. This is an NP-hard problem where existing approximation approaches either fail to scale or yield looser bounds. The paper argues that (1) the global Lipschitz constants are often overly conservative, especially when robustness is needed only on a local input region, and (2) existing scalable SDP relaxations either sacrifice tightness or cannot exploit local slope information of activations. The paper proposes that the Lipschitz constants can be estimated by decomposing global SDPs compositionally across layers (ECLipsE-Gen), and iteratively refining neuron-wise slope bounds using local input information (ECLipsE-Gen-Local).

The experiments are conducted by systematically comparing different algorithms on randomly synthesized neural networks with varied configurations and trained networks. The results indicate that the proposed local bounds are consistently much smaller than global bounds and asymptotically exact. In addition, the robustness analysis shows that local bounds correlate better with observed robustness, whereas global bounds often overestimate worst-case effects.

**Audience:**

Yes

**Audience Explanation:**

Estimating the Lipschitz constant of neural networks is a relevant problem. TMLR’s audience working on theoretical neural network robustness would find this paper's findings interesting.

**Claims And Evidence:**

No

**Claims Explanation:**

The paper’s claims are well supported by theoretical guarantees and carefully designed experiments, with empirical results demonstrating tightness, scalability, and the correlation with robustness in local Lipschitz estimation.

1. ECLipsE-Gen-Local series estimates tight Lipschitz constants and scalable for both depth and width. The paper presents the empirical evidence in Section 4.1 (Figure 1 - 4).
2. Both theory (Section 3.5) and experiments (Section 4.2) demonstrate that the proposed local Lipschitz estimates tighten monotonically with shrinking input regions and converge to the exact Jacobian norm at the region center, as verified by automatic differentiation (Figure 5, Figures 7 - 8). In addition, the local estimation is related to the robustness of the trained network (Section 4.3; Figure 6).

However, the generality claim and downstream analysis (Contribution 4) might not have a direct empirical evidence support and need further clarification.

**Requested Changes:**

1. Clarify which contributions are both theoretically and empirically supported, which are only theoretically supported.
2. Demonstrate or clarify some downstream applications, such as sensitivity analysis, output range estimation, or partial verification.
3. Discuss extension beyond fully connected networks.
4. Justify the choice of neuron networks. In the LipSDP paper, they experimented with up to 500 layers. In addition, it would be nice to clarify the need for LipSDP parallelization. Would it be possible to do the same with ECLipsE-Gen-Local?

---

> ### Author Response · Authors · 2026-01-02
> **Author's response to the review**
>
> We thank the reviewer for the careful reading and constructive suggestions. We revise the manuscript and respond to each point below.
>
> >1. (Claim accuracy) However, the generality claim and downstream analysis (Contribution 4) might not have a direct empirical evidence support and need further clarification.
>
> We clarify that Contribution 4 is not intended as a separate downstream application claim to be benchmarked in this paper. Instead, it highlights a structural flexibility enabled by our generalized certificate (Theorem 3), which is used internally by ECLipsE-Gen-Local for layer-wise slope-bound refinement, and can also be used as a building block in modular certification pipelines. \
> Concretely, arbitrary index selection $(K,L)$ yields certified coordinate-level bounds, which are essential when outputs represent distinct, independently used physical quantities. Representative examples include reach-avoid safety constraints that act component-wise on the state in safe control with learned dynamics models, and physical surrogates such as neural networks for power grid physics where each output dimension corresponds to component-level setpoints with individual operating limits. \
> Similarly, arbitrary consecutive layer selection $(p,i)$ supports certified analysis of intermediate outputs in modular architectures, such as early-exit networks, autoencoder latent spaces, and multi-task shared trunks. \
> We also note that our framework directly supports reachability analysis which finds application in set-propagation based safety verification. In particular, Theorem 6 yields a certified reachable output region as an over-approximation, and tighter Lipschitz estimates reduce conservatism in such settings. \
> See **General Response (A)** for revisions to the Contribution section.
>
> >2. Requested Changes 1 \& 2
>
> We address these two related points together. We appreciate the request for a sharper separation between what is empirically validated in this paper and what is presented as structural flexibility. We now categorize Contributions 1 to 3 as both theoretically and empirically supported, since they constitute the main algorithmic and theoretical deliverables evaluated in the experiments. We treat Contribution 4 as theoretically supported with internal algorithmic validation without benchmarking its application to separate downstream tasks. \
> Accordingly, we will edit Contribution 4 and added the clarifying remark as in **General Response (A)**.
>
> >3. Requested Changes 3
>
> We appreciate the suggestion to clarify applicability to broader architectures. Our framework relies only on affine transformations and sector-bounded activation properties. Therefore, direct generalization is possible for architectures such as Convolutional Neural Networks (CNNs) and Residual Networks by unrolling the network or computing slope bounds for substructures using similar procedures. We focused on fully connected networks in this paper to clearly establish the core mathematical machinery, including the decompositions and local information propagation, and leave extensions to other architectures as future work. \
> ***Text to add in Conclusion.*** Finally, since our certificate relies only on affine transformations and sector-bounded activations, it is natural to extend this framework to other architectures such as Convolutional Neural Networks and Residual Networks.
>
> >4. Requested Changes 4
>
> In Sec. 4.1 Case 2, we deliberately apply ECLipsE-Gen-Local to the full network to preserve coupling while remaining scalable. We compared against the split version of LipSDP because standard LipSDP could not solve these networks within the cutoff time without splitting. \
> We note that the same parallelization strategy used in LipSDP, namely splitting the network into smaller sub-networks and multiplying their Lipschitz bounds, can also be applied to ECLipsE-Gen-Local to achieve speedups. However, this splitting introduces the same trade-off, it improves runtime by solving smaller independent problems, while it sacrifices tightness by discarding cross-block coupling information. In our method, this coupling is precisely the layer-to-layer information carried by the messenger-matrix recursion, so splitting breaks that coupling and can increase conservatism. We also reiterate that our algorithms remain scalable without the need for such splitting. \
> ***Text to add in Sec. 4.1 Case 2 (Setup).*** We choose the split variants of LipSDP as a baseline primarily because the standard LipSDP variants exceed the cutoff time for Lipschitz estimation without splitting and parallelization. Note that while our method can be similarly parallelized by splitting the network into sub-networks, we apply ECLipsE-Gen-Local to the full network as it is scalable and computationally efficient by design, while prioritizing tightness. This choice allows us to retain coupling information across the full network where standard LipSDP fails.

---

### Author Response · Authors · 2026-01-02
**Authors' revision on the paper based on reviewer's comments**

We sincerely thank the reviewers for their constructive feedback. We consolidate three main responses and corresponding manuscript updates below to address recurring comments across reviewers.

**(A) Clarify the claim in Contribution 4.**
We revise Contribution 4 and add a clarifying remark after the contributions list. \
***Contribution 4.*** We establish a generalized certificate that supports ***arbitrary input-output index selections and consecutive-layer subsets*** (Theorem 3). This flexibility is not only used internally for our layer-wise slope-bound refinement, but also serves as a versatile foundation for targeted downstream certification tasks. As one representative example, we leverage the certificate to perform reachability analysis by computing a certified over-approximation of the reachable output region (Theorem 6). \
***After the contribution list.*** Note that beyond standard robustness certification, the flexibility to select arbitrary indices and layer subsets enables specific certified analyses in modular pipelines. For instance, arbitrary index selection allows for coordinate-level sensitivity bounds, which are crucial when outputs represent distinct physical quantities in control systems (e.g., Liu, 2022) or physical surrogates (e.g., NN surrogates for power flow physics Pan et al., 2019; Zhao et al., 2020). Similarly, consecutive-layer selection supports the verification of intermediate representations in architectures like early-exit networks (Teerapittayanon et al., 2016), encoder-decoder structure ((Hinton \& Salakhutdinov (2006)), or multi-task models (Misra et al., 2016). Additionally, our reachability analysis (Theorem 6) provides certified output set over-approximations, explicitly illustrating the practical benefit of reduced conservatism for safety verification.

**(B) Add an example on robustness in Sec. 4.3**
To further demonstrate practical utility, we add a new robustness certification example at the end of Sec. 4.3. Due to space limits, we report key statistics in the main text, while the detailed sample-wise table is included in Appendix A.3. \
***Additional Content in Sec. 4.3*** To further demonstrate the utility of tight and strict Lipschitz estimates, we report robustness certificates on the regularized MNIST model using the standard Lipschitz-margin certificate for multi-class logits (Tsuzuku et al., NeurIPS 2018). This Lipschitz-based robustness certificate provides a lower bound on the $\ell_2$ perturbation magnitude required to eliminate the logit margin between the predicted class and its closest competitor, thereby guaranteeing label invariance within the certified radius. We sweep the same radii $\epsilon=1/2,1/4,\dots,1/256$ and use the same test points as in our local Lipschitz evaluation. For each test point $x$, we compute a certified local Lipschitz bound $L(x,\epsilon)$ that is valid within the analyzed radius $\epsilon$, compute the logit margin $m(x)=f_{\hat y}(x)-\max_{j\neq \hat y} f_j(x)$, and form the candidate radius $r_{est}(x,\epsilon)=m(x)/(\sqrt{2}L(x,\epsilon))$. Since a Lipschitz bound certified on radius $\epsilon$ is also valid for any smaller radius, we use the valid certificate $r_{cert}(x,\epsilon)=\min(r_{est}(x,\epsilon),\epsilon)$, which guarantees that any perturbation with $\lVert\delta\rVert_2<r_{cert}(x,\epsilon)$ cannot change the predicted label. We then report the least conservative certified radius across the swept scales via
$
r(x)=\max_{\epsilon \in 1/2,1/4,\dots,1/256} r_{cert}(x,\epsilon).$
Detailed results, along with comparisons against the certified radius obtained from the global trivial bound $L_{triv}=236.4329$, are reported in Appendix A.3. The mean certified radius produced by our local Lipschitz estimates is $6.14$ times larger than the mean certified radius obtained from $L_{triv}$, indicating a substantial reduction in conservatism.

**(C) Add a discussion of conservatism in CF.**
We add a pointer in the main text after introducing the ECLipsE-Gen series. \
***Appendix A.4.*** In ECLipsE-Gen-Local-CF, at each stage $i$, after obtaining refined per-neuron slope bounds $[\boldsymbol{\alpha}^i,\boldsymbol{\beta}^i]$ from local propagation, we replace them with a valid superset $[\boldsymbol{\alpha}^{i,adj},\boldsymbol{\beta}^{i,adj}]$ that satisfies $\boldsymbol{\alpha}^{i,adj}\odot\boldsymbol{\beta}^{i,adj}=0$ to enable a closed-form update, $i\in Z_{N-1}$. Under $\Lambda_i=\lambda_i I$, the term removed from $\tilde X_{i-1}$ in Eq. (14) is $\Delta^i_{slack} =\lambda_i W_i^T D_{\boldsymbol{\alpha}^i}D_{\boldsymbol{\beta}^i}W_i.$
Therefore, the induced slack is small when the refined lower slope bounds are already close to zero ($D_{\boldsymbol{\alpha}^i}D_{\boldsymbol{\beta}^i}$ is small). The slack can be substantial when both $\boldsymbol{\alpha}^i$ and $\boldsymbol{\beta}^i$ are non-negligible and the scale or alignment of $W_i$ amplifies $W_i^\top D_{\boldsymbol{\alpha}^i}D_{\boldsymbol{\beta}^i}W_i$.

---

### Decision · Action_Editor_JzYx · 2026-02-12

**Recommendation:** Accept as is

**Additional Comments:**

NA

**Audience:**

Yes

**Audience Explanation:**

The paper addresses the problem of estimating Lipschitz constants to certify the robustness and safety of neural networks. Hence,  this work will certainly be of interest to TMLR audience.

**Claims And Evidence:**

Yes

**Claims Explanation:**

The claims are supported by theoretical and empirical evidence.